biophysics/biomaterials/biomimetics

guanine platelet, light reflection, light intensity change, structural colour

**Author for correspondence:**
Masakazu Iwasaka
e-mail: miwamasa@hiroshima-u.ac.jp

# Flashing spots on the dorsal trunk of hardyhead silverside fish

## Masakazu Iwasaka

Hiroshima University, Kagamiyama 1-4-2, Higashihiroshima, Hiroshima 739-8527, Japan

MI, 0000-0003-4213-7390

A large number of living creatures are able to use ambient light effectively in biological signalling. *Atherinomorus lacunosus*, a teleost fish has alignments of circular spots on its dorsal trunk. The spot consists of iridophores, whose diameters are approximately 7–10 µm. The iridophore contains guanine crystals with diameters of 1–3 µm. Here, it is found that more than one spot with a diameter of approximately 0.1 mm causes a rhythmic flashing of light when viewed under white light. The typical light flash has a pulse width of approximately one second. When a pulsed train of flashes appears, the flash repeats at a typical frequency of 0.5–1 Hz. The observed phenomenon is one example of the evidence for the existence of rapid colour changing teleost fish.

## 1. Introduction

Studies on cephalopods (squids, cuttlefish and octopus) [1–4] have revealed that a dynamic camouflage effect occurs in the cellular tissue existing in the skin. A similar mechanism for colour changes has also been discovered in the tissue structures of fish [5–12] and a reptile [13]. In addition, many species of insects have static structural colours [14–19]. In biological materials that are able to control light, guanine is useful for light reflection and optical interference in iridescent tissues [5–7]. Guanine-based photonic crystal structures have been found in many species of fish [5,6,10,20–23], scallop [24] and animal plankton [25,26].

It is often possible to observe many markings on the surface of a fish body. There is a long history of research on fish body colours concerning their purpose in life, such as countershading [27,28], spectral coloration [1–4,29,30] and silvering camouflage of fish [22]. The role of patterns on a fish body surface has been suggested to be an active communication tool, and nonlinear mechanism studies have ascribed the stripe pattern in a fish to a Turing pattern, which is generated by an interaction between chromatophores [31–34]. Chromatophores are groups of cells containing particles that control the colour based on the filtering

**Figure 1.** Experimental set-up for video recordings and fibre optic measurements on the dorsal trunk of hardyhead silverside fish with an array of circular spots. Scale bar is 1 mm. (*b*) Experimental diagram of the observation of fish in an aquarium with a microscope lens and light source. An anaesthetized fish was placed on a bed made of soft clay in a small aquarium containing seawater with aeration. (*c*) Real light measurement, a method for collecting light from the observed area through the microscope lens. Exchange of the CMOS camera with an optical fibre (inner diameter $\phi$5 mm) and the opposite exchange were carried out smoothly (within 10 s).

or reflection of light [35]. Among the chromatophores, iridophores are known to possess reflecting particles such as guanine platelets [31,35].

Concerning the guanine platelets of fish, we have found a light reflection change in the guanine platelets by the magnetically induced orientation [23,36]. We were able to control the orientation of the guanine platelets and observed distinct light flashes. If the guanine platelets in fish skin exist in a flexible condition, we can expect a rapid reflection of light by them. The aim of this study is to find a rapid light intensity change concerning reflecting particles such as guanine platelets in a fish body. In this study, I investigate the dynamics of light reflection for cells of the dorsal skin of the hardyhead silverside fish, *Atherinomorus lacunosus*.

# 2. Material and methods

## 2.1. Sample

Specimens of the hardyhead silverside fish, *A. lacunosus*, were caught in Okinawa, Japan, by using a net and then were transported to Hiroshima. The specimens were kept in an outdoor aquarium (26–28°C) in Okinawa and in an indoor aquarium (26–27°C) in Hiroshima. The height and weight of the used specimens were 98.9 mm ± 4.9 mm and 9.45 g ± 1.43 g, respectively. Nine specimens were used for this study. The sex of the used specimens was not determined in this study. Animal experiments in this study were carried out in accordance with the policies of Hiroshima University Animal Care and Use Committee (approval number:

F19–2, Hiroshima University). A photograph of the belt-like pattern in the dorsal trunk of *A. lacunosus* is shown in figure 1*a*. This belt consisted of a wave-like alignment of circular spots.

## 2.2. Observation of anaesthetized fish in aquarium

Figure 1*b* shows the observation system for the fish in an aquarium, which consisted of a microscope lens and a light source. The fish were anaesthetized by exposure to 0.1% 2-phenoxy-ethanol for 30 s to 1 min and then were placed inside of a small box (aquarium) containing seawater with aeration. Experiments on the anaesthetized fish were performed for 5 to 10 min. Before the anaesthesia wore off, the fish were moved back to the large aquarium.

## 2.3. Video image recording

The images of dorsal spots were obtained by using two types of a high-resolution microscope lens, NAVITAR 2.0 × 1-51473 (Navitar, Rochester, USA) (figure 1*b*) and Hirox MXB-2500REZ (Hirox, Tokyo Japan). The former lens (NAVITAR) was connected to a CMOS camera (HOZAN L-835, HOZAN, Osaka, Japan) by a C-mount adapter. The swimming fish were recorded by a video camera (FDR-AX45TIC, SONY, Tokyo, Japan) for the macroscopic presentation shown in figure 1*a*. The focal distance of the lens (NAVITAR) was 32 mm.

For the digital microscopy recordings using capturing software (Xploview, VIXEN, Saitama, Japan), the white balance was manually set to be 4500–4650, and this white balance parameter was fixed during the individual video recording.

## 2.4. Light source

As a light source for the presented data in this paper, a white LED light (LA-HDF158A, Hayashi Repic Co. Ltd, Tokyo, Japan) was used. A light guide connected to this light source was directed towards the back of the fish, as shown in figure 1*b*. The size of the diameter of the light guide was approximately 8 mm. This provided a scattered light and the light polarization was not additionally modulated. The incident light was provided from the side, as illustrated in figure 1*b*, and the scattered or reflected light from the sample was monitored.

## 2.5. Fibre optic measurements

### 2.5.1. Measurement of real light from dorsal spots and skin

As shown in figure 1*c*, the reflected light from the dorsal trunk of the fish was directly introduced to an optical fibre bundle made of quartz with a total diameter of 5 mm. One end of the optical fibre was connected to a cooled charge coupled device (CCD) spectrometer (UCP-2000, Unisoku Co. Ltd, Osaka, Japan). The light was collected in the CCD with an exposure time of 20 ms, sampling time of 40 ms and recorded time of up to 300 s. The spectrometer was equipped with a diffraction grating that simultaneously collected light ranging from 450 to 600 nm. For the purpose of showing light intensity change (light flash rate), the data of light intensity at three to five wavelengths (450, 500, 528, 574 and 600 nm) was plotted. The obtained data were given relative to a white standard by using a reflectance standard plate (SRS-99-010, Labsphere Inc., New Hampshire, USA) with the same experimental system mentioned above. The other end of the optical fibre bundle (5 mm inner diameter) was positioned at the focused position on the top of the C-mount adapter of the microscope lens (NAVITAR).

Prior to the fibre optic measurement, adjustment of the region of interest was carried out by using a CMOS camera (HOZAN). The exchange of the CMOS camera with the optical fibre was executed within 10 s, then the CCD-spectrometer started the light collection. No crucial data were missed during the spectrometer/camera exchange. After the light collection was finished, the opposite exchange was carried out smoothly, and any drifting of the region of interest (specimen's moving outside of the field of view) was monitored. Any data thought to have been collected while the region of interest drifted was not used for the analysis.

### 2.5.2. Measurements on region of interest on a computer LCD screen during playback of recorded video

To magnify reflecting areas sufficiently for recording, direct measurements on LCD screen were carried out by using the same fibre optic system except for the lens. First, the video playback was paused at the time when the region of interest appeared. After one of the ends of the optical fibre was placed close to the region and fixed with surgical tape (paper tape) on the LCD screen, light collection by a CCD

spectrophotometer was started, then video playback commenced. To assist the fixation of optical fibre, a stand was also used. The data obtained from LCD screen measurement were given relative to a white standard (SRS-99-010, Labsphere).

### 2.5.3. Additional image analysis

To check one of the fibre optic measurements (figure 3b), additional image analyses were performed using Image J (NIH) software. After the colour image was transformed into a grey scale image, the intensity relative to grey in the image was measured.

# 3. Results and discussion

Figure 2 shows two examples of the flashing spots on the dorsal trunk of hardyhead silverside fish. The major colour of the circular spot in figure 2a at 7.50 and 8.73 s was blue or green, but became yellow at 9.30 s. A similar phenomenon is shown in figure 2b, where many circular spots proceeded with the colour change process which was reversible. The spots seemed to cause a light reflection flash.

Next, the continuous flashing of the circular spots was investigated, as shown in figure 3. These data were obtained from two specimens that were different from the specimens used in figure 2. The flash frequency of a dorsal spot was analysed by using two types of fibre optic measurements. In the first specimen (figure 3a), the flashing of the real light was monitored for 300 s (figure 3a(i)). A distinct flash at three wavelengths (528, 574 and 600 nm) was obtained because the light source intensity at these wavelengths was higher than at others (450 and 500 nm). Magnification of the time course for the light at 528 nm (figure 3a(iii)) suggested that the roughly estimated frequency of this spot was 0.5–1 Hz.

In the first two seconds of electronic supplementary material, S_movieFIG3A, 26 sheets of static colour images were captured (25 frames per second) and transformed to grey scale images, then the intensity of grey was plotted versus time, as shown in figure 3b. The intensity time course exhibited approximately four peaks during the first two seconds. The inset images correspond to the intensity changes.

Figure 3c shows the light flashes in another specimen. Figure 3c(i) presents the long-term cycle of the light flash that occurred every fifty seconds, which was repeated at least three times. Magnification of the time course for the light at three wavelengths (528, 574 and 600 nm) (figure 3c(iii)) indicates that the dorsal spots caused a flash of light reflection at frequencies of 0.5–0.7 Hz. The data shown here were obtained only by real light collection through the microscope lens and optical fibre that was connected to the spectrophotometer. In addition, this example showed a slower frequency at 0.02 Hz.

A more detailed analysis at higher frequencies for the flash of dorsal spots was carried out. The analyses were carried out only using light collection from the LCD screen, as shown in figure 4. The intensity was the summation of the light intensity at five wavelengths (450, 500, 528 574, and 600 nm). The data were obtained from an additional two specimens. Both specimens showed flashes that exhibited changes in their intensity at frequencies of 3 to 4 Hz. However, the light intensity changes in a black cell also showed a similar frequency, 3–4 Hz. The analysed videos (see electronic supplementary material, S_movieFIG4A and 4B) exhibited a period with rapid flashing. The fish skin vibration at a frequency of 3–4 Hz may be involved with the flash of the dorsal spots.

Figure 5 shows an analysis of the reflecting particles, presumably guanine crystals in a dorsal spot. Figure 5a,b is high-resolution microscopy images of circular spots captured from a euthanized specimen. The reflecting circular spot was shown to exist on a black cell with dendrites. According to previous studies [31,35], this black cell is a melanophore that contains the black pigments melanin. In both images ((a) and (b)), there were many iridophores with a diameter of 7–10 μm, and these cells formed the observed circular spot. In addition, the magnified image (figure 5c,d) indicated that the iridophores had guanine crystal particles whose diameter was estimated to be 1–3 μm.

To analyse these particles, a piece of dorsal skin with scales was separated from a euthanized specimen, and the particles were extracted from the skin by sticking with a plastic spatula in water. The particles floating in water exhibited Brownian motion and their rotation caused the light flash pattern shown in figure 5e. Larger particles with a size of 2–3 μm exhibited a distinct light reflection change. The light reflection behaviour of particles floating in water resembled the light reflection of the guanine platelets of goldfish and Japanese anchovy [23,36]. Fibre optic measurements on a reflecting particle appearing in a video display provided a time course of the light intensity, as shown

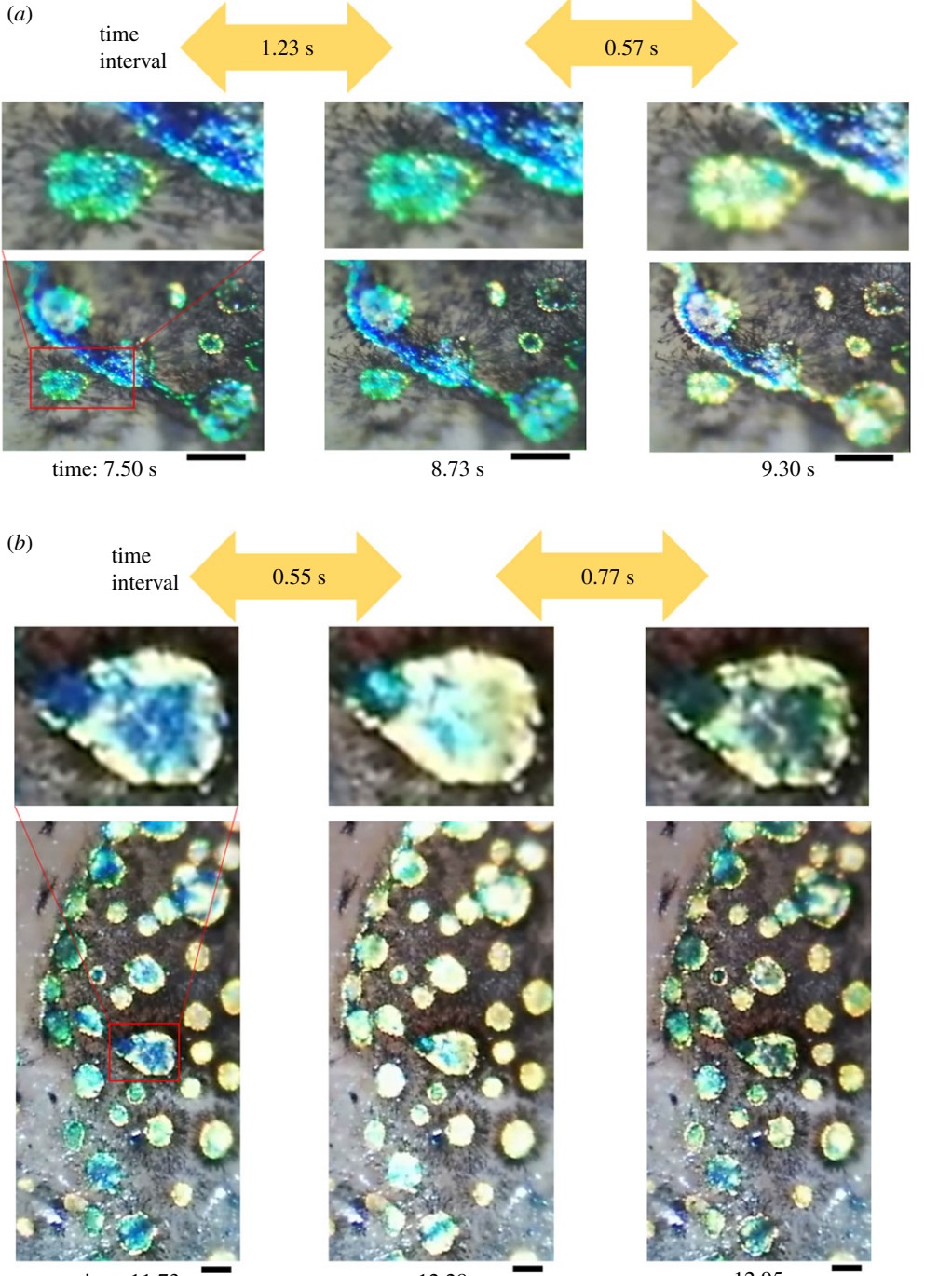

**Figure 2.** The observed phenomenon of the flash of light reflection in spots of the dorsal trunk of hardyhead silverside fish. (*a,b*) Two examples of the phenomenon. Time indicators denote passing time in the recorded video for each of the static images. The static images in (*a*) and (*b*) were captured from the recorded electronic supplementary material, videos S_movieFIG2A and 2B, respectively. Scale bar is 100 μm.

in figure 5*f*. The pulse width of the light flash generated by this floating particle was less than 1 s, i.e. a similar width to that observed from a dorsal spot. The frequency of the reflecting particles floating in water appeared random. The random reflection of the floating particles was the same as the previously reported behaviour of guanine crystal platelets floating in water [36].

The pioneer works on the rapid colour change of iridophore of teleost fish, *Pentapodus paradiseus* [11] and *Hoplolatilus chlupatyi* [12] showed a distinct colour change from blue to red. The hardyhead silverside fish, *A. lacunosus* did not show a change to red colour, but the fish could repeat the flashing faster.

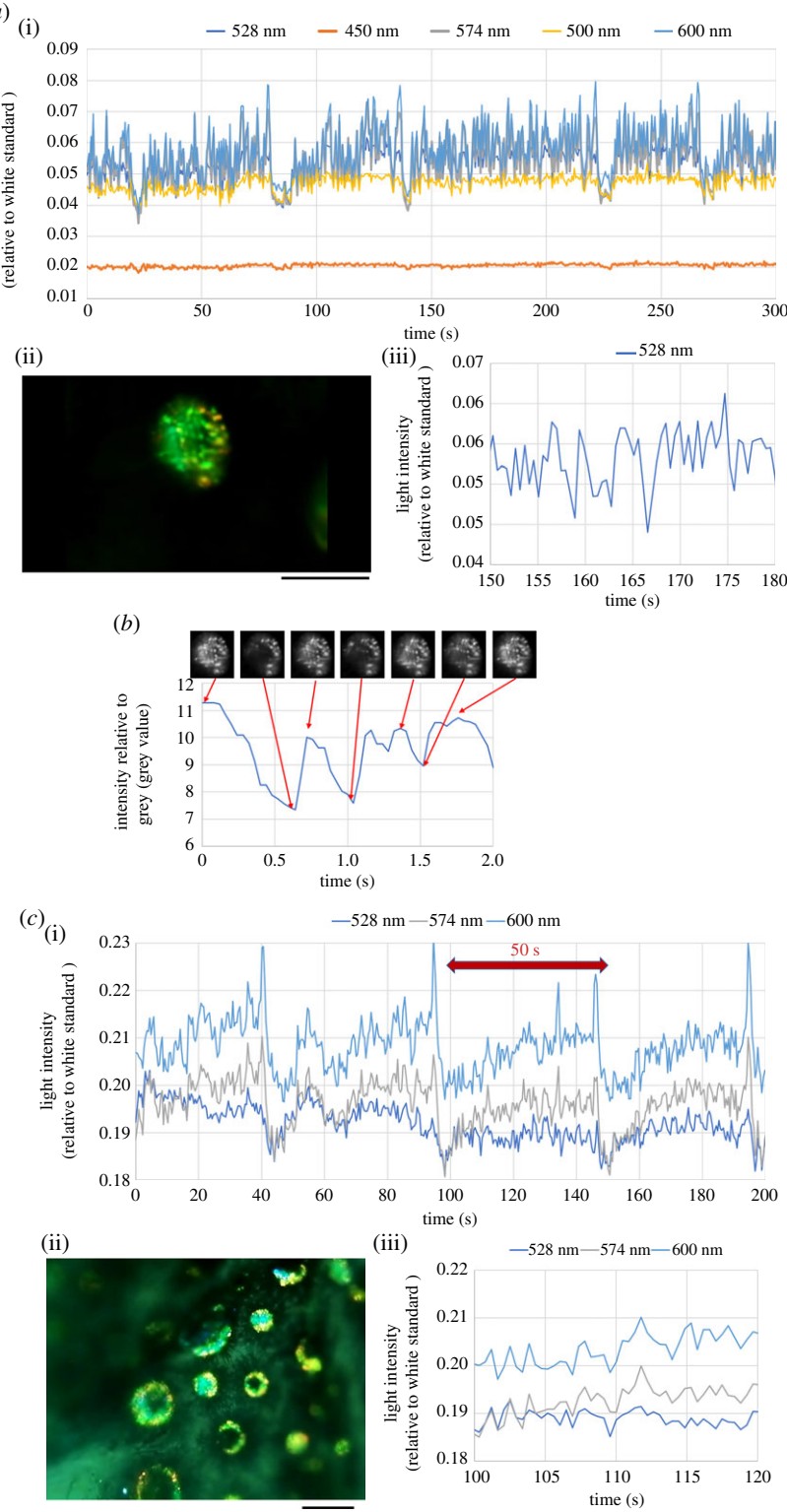

**Figure 3.** Analysis of flash frequency of the dorsal spot by using two types of fibre optic measurements, light collection through a microscope lens. (*a*) First specimen. (i) Time course of real light at five wavelengths through a microscope lens after light reflection from the body surface of the dorsal trunk. (ii) Static image in the beginning of electronic supplementary material, S_movieFIG3A. Scale bar is 100 µm. The spot existed in the area measured for the data (i) and (iii), which was determined separately from the video recording. (iii) Magnification of the time course at 528 nm shown in (i) from 150 to 180 s. (*b*) Image analysis using Image-J was carried out after the colour image was transformed to a grey scale image. (*c*) Second specimen. (i) Time course of real light at three wavelengths through a microscope lens after light reflection from the body surface of the dorsal trunk. (ii) Static image taken at the beginning of electronic supplementary material, S_movieFIG3C. Scale bar is 100 µm. The spot existed in the area measured for the data (i) and (iii), which was determined separately from the video recording. (iii) Magnification of the time course of (i) from 100 to 120 s.

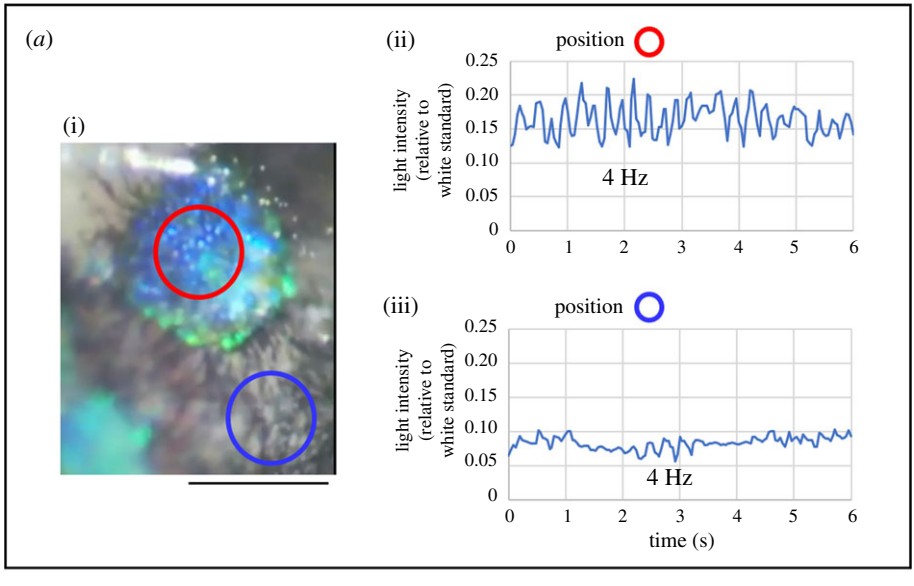

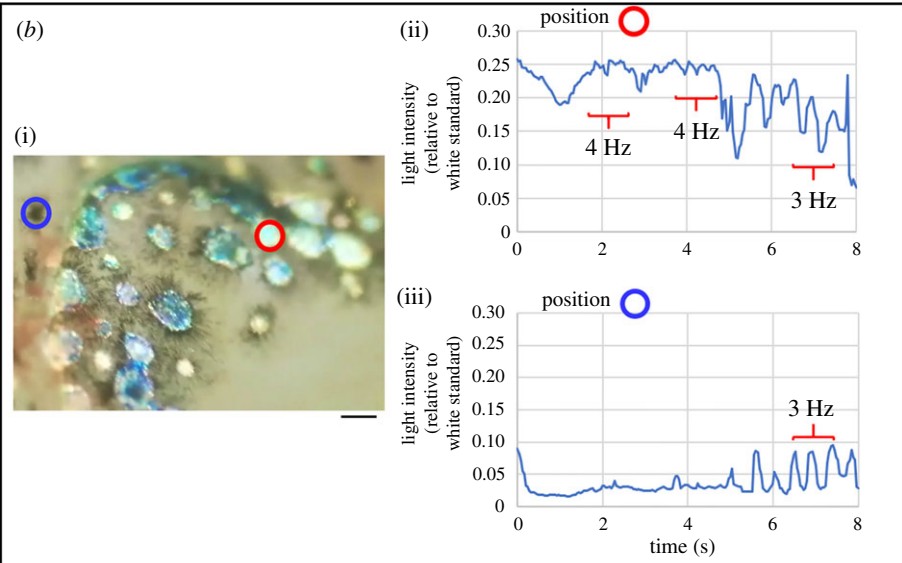

**Figure 4.** Analysis of flash frequency of the dorsal spot showing a fast flashing. (*a*), (*b*) Two examples of fibre optic measurements from a position on a computer LCD screen displaying the recoded video (see electronic supplementary material, S_movieFIG4A, 4B). Red and blue circles indicate the position on a dorsal spot and a black cell to which an end of an optical fibre ($\phi$5 mm) was positioned, respectively. Scale bar is 100 μm.

A mechanism controlling the frequency of flashes may exist inside the iridophores, spots or underlying skin. A detailed analysis of this will be performed in a future study.

# 4. Conclusion

It was found that circular spots on the dorsal trunk of hardyhead silverside fish, *A. lacunosus*, cause a light flash under illumination.

Fibre optic measurements revealed that the flash in the spot continues in a style of a pulsed train whose frequency varied from 0.02 to 4 Hz. Normally, an anaesthetized living fish exhibited flashes of the dorsal spots at frequencies of 0.5–1 Hz. Higher frequencies at 3–4 Hz were also observed from the vibrational motion of the skin.

The dorsal spot with a typical diameter size of 100 μm consisted of 30–50 iridophores whose sizes were 7–10 μm. Each of the cells contained guanine crystal particles with a size of 1–3 μm.

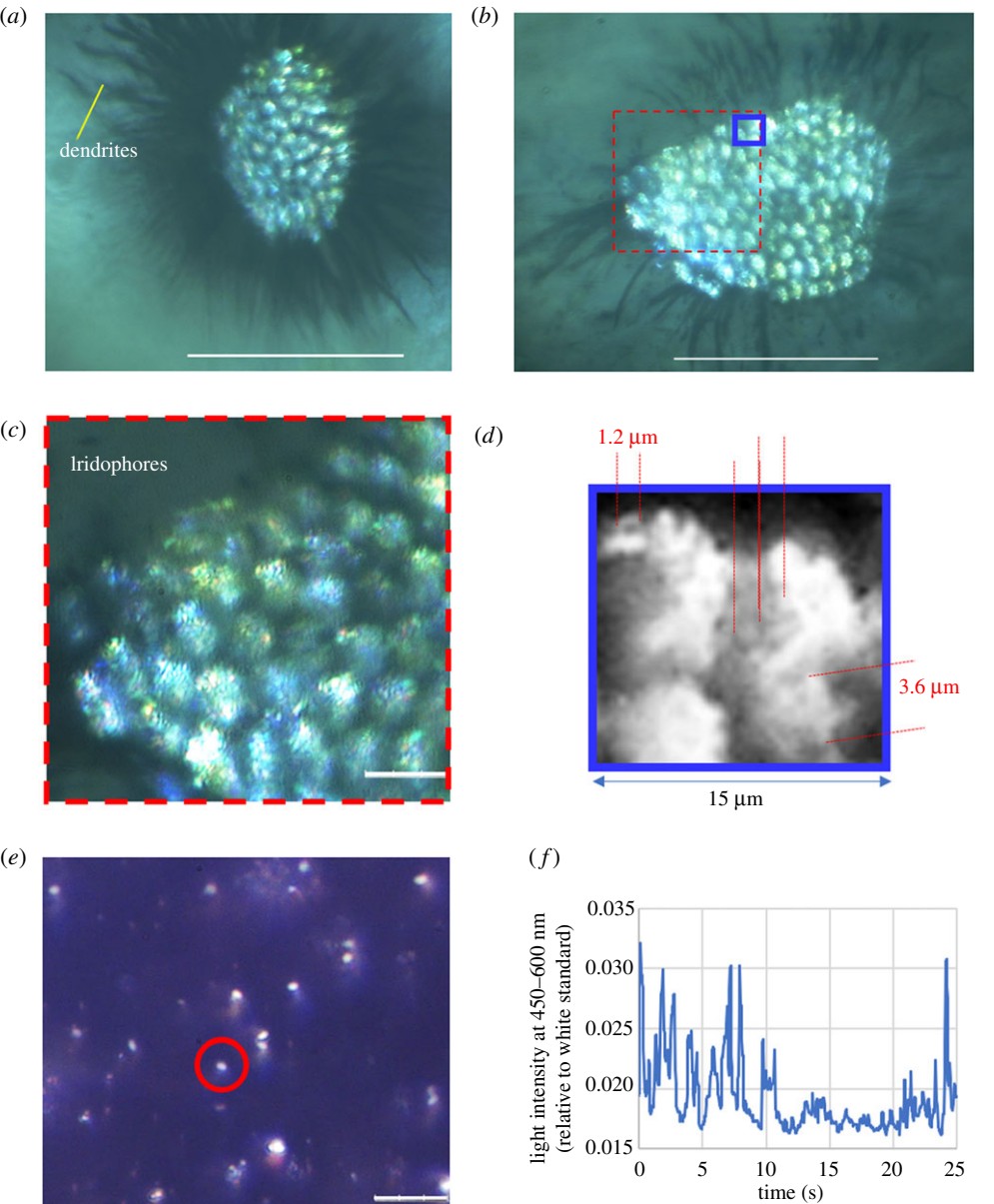

**Figure 5.** Analysis of reflecting particles in the iridophores of a dorsal spot. (*a*), (*b*) Reflecting iridophores on a black cell with dendrites. Scale bar is 100 µm. (*c*) Magnified image of the area marked by a red square shown in (*b*). Scale bar is 15 µm. (*d*) Magnified image of the area marked by a blue square shown in (*b*). (*e*) Reflecting particles extracted from the dorsal circular spots. The particles are floating in water. Scale bar is 15 µm. (*f*) Fibre optic measurement of flashing of the reflector particles floating in water. The measurement was carried out at a position on a computer LCD screen displaying the recoded video (see electronic supplementary material, S_movieFIG5). The end of an optical fibre (φ5 mm) was positioned at the point marked by a red circle shown in (*e*).

The results indicated that in the dorsal trunk of hardyhead silverside fish, there were iridophores frequently changing their light intensity under white light illumination by using guanine crystal particles.

Data accessibility. All data are available from the Dryad Digital Repository: https://doi.org/10.5061/dryad.9zw3r22bw [37].
Authors' contributions. M.I. performed the experimental design, experiments, measurements and analyses. All part of the manuscript and illustrations were prepared by M.I.
Competing interests. I declare I have no competing interests.
Funding. This work was supported by JST-CREST 'Advanced core technology for creation and practical utilization of innovative properties and functions based upon optics and photonics (grant no. JPMJCR16N1).'
Acknowledgements. The author thanks Edanz Group for editing a draft of this manuscript.

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
