## [Peer Review File · Royal Society Open Science]

Review History

RSOS-201578.R0 (Original submission)

Review form: Reviewer 1

Is the manuscript scientifically sound in its present form?

No

Are the interpretations and conclusions justified by the results?

Yes

Is the language acceptable?

No

Do you have any ethical concerns with this paper?

No

Have you any concerns about statistical analyses in this paper?

No

Reports © 2021 The Reviewers; Decision Letters © 2021 The Reviewers and Editors; Responses © 2021 The Reviewers, Editors and Authors. Published by the Royal Society under the terms of the Creative Commons Attribution License <http://creativecommons.org/licenses/by/4.0/>, which permits unrestricted use, provided the original author and source are credited

Recommendation?

Major revision is needed (please make suggestions in comments)

Comments to the Author(s)

See comments in attached word doc (Appendix A).

Review form: Reviewer 2**Is the manuscript scientifically sound in its present form?**

No

Are the interpretations and conclusions justified by the results?

No

Is the language acceptable?

Yes

Do you have any ethical concerns with this paper?

No

Have you any concerns about statistical analyses in this paper?

Yes

Recommendation?

Major revision is needed (please make suggestions in comments)

Comments to the Author(s)

Masakazu Iwasaka, Rapid active-reflection display manuscript

In this manuscript, the author presents an interesting structural colour-changing phenomenon in a fish species. Such colour changes are not uncommon amongst fishes, although the speed of this colour change is unusual. The author uses stereo microscopic imaging by attaching a camera to the microscope to image these colour changes. The author then analyzes the RGB contents of the images to quantify intensity changes of the reflective patterns. A magnetic field is applied to some guanine crystal reflectors in an experimental setup to show that magnetic stimuli can change reflectance.

Reading about this interesting fish species and its fascinating colour change is exciting. I have a series of comments for the author that I hope will improve the manuscript.

Overall, more detail is required in the materials and methods section. The entire manuscript could be made more succinct by reducing sections that do not directly relate to or benefit this paper. The flow of the manuscript would also be improved by ensuring that placement sentences and paragraphs is logical and that an argument is built and discussed fully. The figures need work, e.g., labelling areas of interest that the author would like the reader to look at. There is too much guess work at this point. Some images appear irrelevant, some of the plots do not seem to match the images that the data were (seemingly) derived from.

I will use the outside line numbers on my version of the pdf (there are two, numbered 1-60 and 5-30, I will use the 1-60 numbering).

Title: Put FISH in here, or TELEOST, so that this paper can more easily be searched. STRUCTURAL COLORATION could be in here too, or in the keywords. The word DISPLAY should be removed in my opinion. Display in animal behaviour means a signalling intent, which has to be identified as such.

Abstract, p1:

Line 32 coloring species ... poor wording, I suggest you re-word. For example: attractive not only to members of the same colorful species...

Line 35 detailed function [...] remains unknown ... This needs to be toned down. There is a lot of literature detailing functions and mechanisms of biological reflectors in a lot of species. There is always more work that can be done to elucidate further, which this paper does.

Line 35 After spp name, add teleost fish here, to make clear what kind of species you are describing here.

Line 39 rapid reflection changes in platelets There is no evidence for this from this paper. This is likely to be the case assuming that this teleost uses a similar system compared to other fish spp but it sounds like you have direct evidence for this, which is not the case. I would suggest taking this statement out of the abstract.

Line 43 passive illumination ... Illumination is an incorrect term. This is structural reflectance. Also, it is passive in terms of using incoming light to reflect back but it appears to be active in terms of control. This sentence is best refined along these lines.

Furthermore, the abstract is incomplete, e.g., magnetic field experiment not mentioned at all.

Introduction, manuscript (not pdf) pp 2-3:

Line 7 Rewrite beginning of first sentence [...] cephalopods (squid, cuttlefish and octopus) [...] The way it currently reads, you are saying that cephs, cuttle and octop are separate organisms.

Line 8 ... IN the skin, not near ... color changes, not coloring

Line 14 purpose in life ... the author lists several (including some of the older) papers on this topic that talk about the various purposes, including camouflage by countershading, but also various signalling examples. So, this paragraph could be brought to the point by being more specific about the functional aspect of what we do know.

The last two sentences (lines 15-17) in this first paragraph are out of context and stick out - they are better moved elsewhere or deleted.

Line 22 ascribed instead of imputed, which has a negative meaning

Sentence Lines 26-27 (cell to cell interactions ...) how do these studies relate to the author's study? Merely saying this here makes little sense in the context of this paper.

Line 29 The silvery shine on the side ... of which spp?

Lines 33-36 Sentence that starts with Detailed analyses of the fish ... again, how does this referenced study relate to the author's study?

Line 37 Most cases of coloring ... first, it needs to be fish body colors, not coloring on a fish body Then, these types of strong statements require a reference.

It has to be said that this is an area about which we do not know very much. Most people do not have the time or money to do the sorts of studies that are required to answer this question, i.e., watching animals for hours, days or weeks at a time, and for the few studies that are available (all older literature from the earlier part of the 1900s) it turns out that what we have assumed are

static patterns actually change. It just takes the fish hours, even several days to do so. Examples of flounders, skates, and others.

Paragraph, lines 44-48 ... This paragraph makes no sense in the context of this paper. Which spp is the author referring to? At the end of this paragraph the author jumps back to talking about guanine crystals.

Last two paragraphs of the introduction, lines 49-54 on p2 and lines 3-9 on p3. ... These last two paragraphs can be turned into a couple of sentences that succinctly state the objectives of the paper. The results of the paper should be removed here.

Other important aspects: what are iridopores, what are chromatophores, references, etc.

Materials and methods, p3

This section is incomplete and needs some detailed attention.

Line 15 Specimens [...] were obtained ... using fishing rods, boat, seine nets, bought in a store? more details please.

How many fish?

What size?

Males, females?

Kept at aquarium, temperature, water supply, animal care details....?

Line 21 The iridophore spots ... Euthanised fish? Living specimen? If living specimen, describe handling procedure, incl revival etc. If euthanized, was the whole fish used or just a dissected part of it?

Line 23 Coolpix camera ... did you white balance manually? Was exposure manual? If you had auto exposure on, I have a hard time seeing that you can analyze the RGB pixels for intensity changes between images taken at different times. More info, please. A spectrometer would have been the way to go with this...

What is your sample size? How many images? Still images? From video? It seems that it would be hard to capture still images from these iridescent flashes that you suggested are as fast as <1sec.

How was the pixel analysis done? Did you have an area of interest that was the same for each image? How many pixels? As I mentioned above, unless your camera setting was in manual mode, the RGB content of each of your images would be have been adjusted by the camera and the images would not be useful for RGB analysis.

Line 28 Sentence: The light illumination ... light was monitored 50-80 degrees ...

Lighting is probably the most critical part when working with structural reflectors because changing the angle of the light, or the spread of the focusing beam can give a reflector the appearance of changing colour, even when in nature, there is no colour change. I do not doubt that this species truly is capable of active colour change but a more detailed explanation on methods is necessary here to avoid confusion. A 50 to 80 degree angle is a very big difference that could cause reflectors to appear to turn on an off, just by moving the light.

More details about the light source are necessary (Point? Parallel? What kind of beam?), where was it attached, how were angles controlled?

Line 34 Radiance ... ? This would require very careful measurements with specific photometric equipment. This does not seem to be the case, unless I'm missing something. You looked at overall intensity, it seems to me. Also, the terms radiance, intensity irradiation and some others, are used interchangeably, which has to be fixed throughout the manuscript.

Sentence lines 37-38 When green color of an image [...]

What does this mean? This method needs more explanation.

Normally, a spectrometer would be used for this type of work but not everyone has access to this kind of equipment and I appreciate that sometimes you have to make do with what you have available. But explain it sufficiently, so that it makes sense and a reader can fully trust your analysis.

Line 41 A realistic model [...] was formed

This sounds like 3D printing but the following sentences suggest you dissected a piece of fish tissue out and placed the material on this acrylic plate?

If this is the case, SAMPLE would be a better word, not model.

Line 42 delete in the plane. You can shorten this ... acrylic plate (20mm x 20mm, 1mm thick)

Lines 42-45 Turn this sentence around...

A well was drilled into the center of this plate... Guanine platelets were suspended inside this well...

Line 46 why sea bream? Why not the spp you worked on?

Results and Discussion, pp4-11

Does the living animal show these colour flashes? It is unclear from the materials and methods whether the author worked with living animals or euthanized/ dissected specimens.

Do females and males show the same colour flashes?

P4, line 11 where is the dendrite? this is interesting because it would point future research towards physiological studies. It needs to be pointed out in the image.

Figure 1 ... the two higher mag yellow boxes look identical. Are both needed? I would delete one.

P5 figure 2A what are the surrounding black structures? What do the two images show? At different times? Different angles?

Figure 2B same as for A

Also, there is not a dramatic difference between the center and the right-hand image. I would suggest getting rid of one of them.

Fig2C replace irradiation with incident light.

Figure 2 Caption

Light illumination: what is the exact angle? how did you ensure that the incident light direction did not change during imagery, was the preparation secured, etc?

Scale bar, 100 mm.... that must be a typo

Micro particles if you want to start using the materials terminology and start talking about particles, I would suggest you introduce this someplace. Otherwise, stick to reflecting plates, guanine plates, ...

Fig2C What is this Fig. is based on? Unless you have TEM images, this kind of theoretical drawing has little value here. You also do not sufficiently explain the point of this drawing. The left drawing suggests if plates are angled so that light goes through a series of internal reflective processes, instead of being reflected upwards, an observer above sees nothing. Right drawing

shows the assemblage for a visible reflectance when viewed from above. This needs to be stated. But, again, have you verified that these plates behave like "mirrors"?

P6 First sentence ... light irradiation ... replace with incident light.
Same line ... along the optical axis of the lens ... Please explain this: This would mean illumination came from the inside of the microscope for this to be along the optical axis. It does not sound like this is what you did.

Line 10 micro particles... as before

Lines 13-15 Two sentences: Behind the iridophore, a pseudopod..... This suggests
It is unclear to me what the author is trying to say.

Lines 25-27 A tunable.... Remove the two sentences here and focus on your results

Paragraph lines 27-37 is this in a euthanized fish? Anesthetized fish? This is important to know. Are these flashes artifacts or a visual signal that these fish put out? Do they do this while actively swimming?

Two sentences in lines 38-45 are awkward, perhaps rephrasing them would help clarify what the author is trying to say.

P7 Panel Fig 3A ... What is the difference between these 11 images? Different time scale? They all look the same to me. How were the data in the plot underneath derived? Average of some number of pixels? This chart does not visually jive with the images above. Get rid of 0 and 12, since there are no corresponding sequences above.

Fig. 3B Three sequences are shown in the plot, but only two images are shown. Again, to me, there is not an appreciable difference between the iridescent spots here, looking at the image at the top and the one at the bottom. Spot iv looks a little different perhaps, and there is a tiny change in v_i , which does not show up in your corresponding plot. The other three spots do not look very different at all.

Line 43 brilliance ... The author uses different terminology throughout.
Intensity relative to grey may be better?

P8 Line12 Should not be strongly related to blood flow [...] Innervation by the neural network [...]
This conclusion is based on what? More explanation is needed here.

Line 15 Brilliance again. See above

Line 19-20 [...] should exist inside the parietal skin
Unless there is evidence, this may need to be stated more as a question than a statement.

Figure 4
Delete separately in line 45
Line 48, summation in RGB colors ... this needs to be explained. Meta analysis ... this needs explanation as well.

Figure 5
5C needs more explanation to be useful. It is unclear whether one scenario explains lack of reflectance and the other scenario explains reflectance under magnetic field stimulation?

The magnetic field experiment needs more explanation, including more methods. Why was this done in the first place? What was the purpose? Why not use the spp that you studied? Why a bream?

P11 Line 20 Reference for lanternfish paper, please.

P11

This entire page, in my opinion, needs to be re-structured, made more succinct and clear. The author talks about applying a magnetic field to bream (why bream, what about *A. lacunosus*?) guanine crystals and observing reflective changes. This moves away from the actual biology of this animal, which is fine. It moves into a potential realm of application, although it remains a mystery to me how applying changes in magnetic field direction (how about strength?) would be a realistic application potential. The author provides limited insight into what the magnetic field experiments add to this study.

The flow of this entire page needs to be improved. Sentences and paragraphs jump back and forth with little connection.

Supporting information. P15-18

Line 36 Radiance... terminology... with your method, you cannot talk about radiance here.

Line 39 radiance [...] disappeared ... Again, terminology re radiance. Also, reflectance did not disappear, it would have been reflected back downwards. But yes, it is not transmitted, which is the point you are trying to make.

Line 43 Particles, crystals, terminology is used interchangeably throughout the manuscript. Delete BIOGENIC

Figure S1

How were these angles controlled? What exactly were these angles? Estimating them from the drawing is not possible.

What was the light source? Point light or scattered? Please make sure this information is in the materials section.

This figure needs to be explained a lot more to be useful.

The images all more or less look the same to me, there is not a huge difference, perhaps a slight decrease in reflectance between a,b,c compared to d,e but other than that, it is unclear why a,b and c are here, what the arrows show, and why d and e are here.

Figure S2

This idea of blood flow needs more careful explanation. The idea itself makes some sense to me but the supporting evidence shown here does not convince me. The images shown all look the same to me. Perhaps I'm missing something but the author is also not pointing me in a particular direction. More detailed explanation is necessary.

Review form: Reviewer 3

Is the manuscript scientifically sound in its present form?

Yes

Are the interpretations and conclusions justified by the results?

Yes

Is the language acceptable?

Yes

Do you have any ethical concerns with this paper?

No

Have you any concerns about statistical analyses in this paper?

No

Recommendation?

Accept with minor revision (please list in comments)

Comments to the Author(s)

The author describes a flickering reflective pattern in a spot in the skin of a fish. The authors describe the flickering and suggest that it is actively controlled by the fish, who continuously rearranges guanine crystals in the spot to change the reflective patterns. They experimentally support this conjecture by showing that guanine crystals in solution can be arranged by magnetic fields. This is an interesting observation, and should set the stage for more mechanistic investigations into the phenomenon. The science is sound, but I had to read it a few times before I fully understood the story. The authors should clarify it a bit more by, for example, using the common name of the fish as well as the scientific name and outlining the motivation of the study more clearly. Otherwise I just have a few comments and suggestions.

P2L18: why are they considered important and interesting?

P2L49: "light switching panel" is not clear. Do you mean the lights are turning off and on, or flashing, or what?

P2L51: How do you know it's active?

P4L14: green and blue colors are typically also structural colors. Do you know how these colors are made?

P8L13: Is this just normal movement of the scales? Do other fish do this?

P10L53: Neat experiment!

Decision letter (RSOS-201578.R0)

Dear Professor Iwasaka,

The Editors assigned to your paper RSOS-201578 "Rapid active-reflection display on dorsal stripes of *Atherinomorus lacunosus*" have now received comments from reviewers and would like you to revise the paper in accordance with the reviewer comments and any comments from the Editors. Please note this decision does not guarantee eventual acceptance.

Please submit your revised manuscript and required files (see below) no later than 21 days from today's (ie 09-Nov-2020) date. Note: the ScholarOne system will 'lock' if submission of the revision is attempted 21 or more days after the deadline. If you do not think you will be able to meet this deadline please contact the editorial office immediately.

on behalf of Dr Silvia Vignolini (Associate Editor) and Pietro Cicuta (Subject Editor)
openscience@royalsociety.org

Reviewer comments to Author:

Reviewer: 1
Comments to the Author(s)

See comments in attached word doc.

Reviewer: 2
Comments to the Author(s)

Masakazu Iwasaka, Rapid active-reflection display manuscript

In this manuscript, the author presents an interesting structural colour-changing phenomenon in a fish species. Such colour changes are not uncommon amongst fishes, although the speed of this colour change is unusual. The author uses stereo microscopic imaging by attaching a camera to the microscope to image these colour changes. The author then analyzes the RGB contents of the images to quantify intensity changes of the reflective patterns. A magnetic field is applied to some guanine crystal reflectors in an experimental setup to show that magnetic stimuli can change reflectance.

Reading about this interesting fish species and its fascinating colour change is exciting. I have a series of comments for the author that I hope will improve the manuscript.

Overall, more detail is required in the materials and methods section. The entire manuscript could be made more succinct by reducing sections that do not directly relate to or benefit this paper. The flow of the manuscript would also be improved by ensuring that placement sentences and paragraphs is logical and that an argument is built and discussed fully. The figures need work, e.g., labelling areas of interest that the author would like the reader to look at. There is too much guess work at this point. Some images appear irrelevant, some of the plots do not seem to match the images that the data were (seemingly) derived from.

I will use the outside line numbers on my version of the pdf (there are two, numbered 1-60 and 5-30, I will use the 1-60 numbering).

Title: Put FISH in here, or TELEOST, so that this paper can more easily be searched. STRUCTURAL COLORATION could be in here too, or in the keywords. The word DISPLAY should be removed in my opinion. Display in animal behaviour means a signalling intent, which has to be identified as such.

Abstract, p1:

Line 32 coloring species ... poor wording, I suggest you re-word. For example: attractive not only to members of the same colorful species...

Line 35 detailed function [...] remains unknown ... This needs to be toned down. There is a lot of literature detailing functions and mechanisms of biological reflectors in a lot of species. There is always more work that can be done to elucidate further, which this paper does.

Line 35 After spp name, add teleost fish here, to make clear what kind of species you are describing here.

Line 39 rapid reflection changes in platelets There is no evidence for this from this paper. This is likely to be the case assuming that this teleost uses a similar system compared to other fish spp but it sounds like you have direct evidence for this, which is not the case. I would suggest taking this statement out of the abstract.

Line 43 passive illumination ... Illumination is an incorrect term. This is structural reflectance. Also, it is passive in terms of using incoming light to reflect back but it appears to be active in terms of control. This sentence is best refined along these lines.

Furthermore, the abstract is incomplete, e.g., magnetic field experiment not mentioned at all.

Introduction, manuscript (not pdf) pp 2-3:

Line 7 Rewrite beginning of first sentence [...] cephalopods (squid, cuttlefish and octopus) [...] The way it currently reads, you are saying that cephs, cuttle and octop are separate organisms.

Line 8 ... IN the skin, not near ... color changes, not coloring

Line 14 purpose in life ... the author lists several (including some of the older) papers on this topic that talk about the various purposes, including camouflage by countershading, but also various signalling examples. So, this paragraph could be brought to the point by being more specific about the functional aspect of what we do know.

The last two sentences (lines 15-17) in this first paragraph are out of context and stick out - they are better moved elsewhere or deleted.

Line 22 ascribed instead of imputed, which has a negative meaning

Sentence Lines 26-27 (cell to cell interactions ...) how do these studies relate to the author's study? Merely saying this here makes little sense in the context of this paper.

Line 29 The silvery shine on the side ... of which spp?

Lines 33-36 Sentence that starts with Detailed analyses of the fish ... again, how does this referenced study relate to the author's study?

Line 37 Most cases of coloring ... first, it needs to be fish body colors, not coloring on a fish body Then, these types of strong statements require a reference.

It has to be said that this is an area about which we do not know very much. Most people do not have the time or money to do the sorts of studies that are required to answer this question, i.e., watching animals for hours, days or weeks at a time, and for the few studies that are available (all older literature from the earlier part of the 1900s) it turns out that what we have assumed are static patterns actually change. It just takes the fish hours, even several days to do so. Examples of flounders, skates, and others.

Paragraph, lines 44-48 ... This paragraph makes no sense in the context of this paper. Which spp is the author referring to? At the end of this paragraph the author jumps back to talking about guanine crystals.

Last two paragraphs of the introduction, lines 49-54 on p2 and lines 3-9 on p3. ... These last two paragraphs can be turned into a couple of sentences that succinctly state the objectives of the paper. The results of the paper should be removed here.

Other important aspects: what are iridopores, what are chromatophores, references, etc.

Materials and methods, p3

This section is incomplete and needs some detailed attention.

Line 15 Specimens [...] were obtained ... using fishing rods, boat, seine nets, bought in a store? more details please.

How many fish?

What size?

Males, females?

Kept at aquarium, temperature, water supply, animal care details....?

Line 21 The iridophore spots ... Euthanised fish? Living specimen? If living specimen, describe handling procedure, incl revival etc. If euthanized, was the whole fish used or just a dissected part of it?

Line 23 Coolpix camera ... did you white balance manually? Was exposure manual? If you had auto exposure on, I have a hard time seeing that you can analyze the RGB pixels for intensity changes between images taken at different times. More info, please. A spectrometer would have been the way to go with this...

What is your sample size? How many images? Still images? From video? It seems that it would be hard to capture still images from these iridescent flashes that you suggested are as fast as <1sec.

How was the pixel analysis done? Did you have an area of interest that was the same for each image? How many pixels? As I mentioned above, unless your camera setting was in manual mode, the RGB content of each of your images would be have been adjusted by the camera and the images would not be useful for RGB analysis.

Line 28 Sentence: The light illumination ... light was monitored 50-80 degrees ...

Lighting is probably the most critical part when working with structural reflectors because changing the angle of the light, or the spread of the focusing beam can give a reflector the appearance of changing colour, even when in nature, there is no colour change. I do not doubt that this species truly is capable of active colour change but a more detailed explanation on

methods is necessary here to avoid confusion. A 50 to 80 degree angle is a very big difference that could cause reflectors to appear to turn on an off, just by moving the light.

More details about the light source are necessary (Point? Parallel? What kind of beam?), where was it attached, how were angles controlled?

Line 34 Radiance ... ? This would require very careful measurements with specific photometric equipment. This does not seem to be the case, unless I'm missing something. You looked at overall intensity, it seems to me. Also, the terms radiance, intensity irradiation and some others, are used interchangeably, which has to be fixed throughout the manuscript.

Sentence lines 37-38 When green color of an image [...]

What does this mean? This method needs more explanation.

Normally, a spectrometer would be used for this type of work but not everyone has access to this kind of equipment and I appreciate that sometimes you have to make do with what you have available. But explain it sufficiently, so that it makes sense and a reader can fully trust your analysis.

Line 41 A realistic model [...] was formed

This sounds like 3D printing but the following sentences suggest you dissected a piece of fish tissue out and placed the material on this acrylic plate?

If this is the case, SAMPLE would be a better word, not model.

Line 42 delete in the plane. You can shorten this ... acrylic plate (20mm x 20mm, 1mm thick)

Lines 42-45 Turn this sentence around...

A well was drilled into the center of this plate... Guanine platelets were suspended inside this well...

Line 46 why sea bream? Why not the spp you worked on?

Results and Discussion, pp4-11

Does the living animal show these colour flashes? It is unclear from the materials and methods whether the author worked with living animals or euthanized/ dissected specimens.

Do females and males show the same colour flashes?

P4, line 11 where is the dendrite? this is interesting because it would point future research towards physiological studies. It needs to be pointed out in the image.

Figure 1 ... the two higher mag yellow boxes look identical. Are both needed? I would delete one.

P5 figure 2A what are the surrounding black structures? What do the two images show? At different times? Different angles?

Figure 2B same as for A

Also, there is not a dramatic difference between the center and the right-hand image. I would suggest getting rid of one of them.

Fig2C replace irradiation with incident light.

Figure 2 Caption

Light illumination: what is the exact angle? how did you ensure that the incident light direction did not change during imagery, was the preparation secured, etc?

Scale bar, 100 mm... that must be a typo

Micro particles if you want to start using the materials terminology and start talking about particles, I would suggest you introduce this someplace. Otherwise, stick to reflecting plates, guanine plates, ...

Fig2C What is this Fig. is based on? Unless you have TEM images, this kind of theoretical drawing has little value here. You also do not sufficiently explain the point of this drawing. The left drawing suggests if plates are angled so that light goes through a series of internal reflective processes, instead of being reflected upwards, an observer above sees nothing. Right drawing shows the assemblage for a visible reflectance when viewed from above. This needs to be stated. But, again, have you verified that these plates behave like "mirrors"?

P6 First sentence ... light irradiation ... replace with incident light.

Same line ... along the optical axis of the lens ... Please explain this: This would mean illumination came from the inside of the microscope for this to be along the optical axis. It does not sound like this is what you did.

Line 10 micro particles... as before

Lines 13-15 Two sentences: Behind the iridophore, a pseudopod..... This suggests It is unclear to me what the author is trying to say.

Lines 25-27 A tunable.... Remove the two sentences here and focus on your results

Paragraph lines 27-37 is this in a euthanized fish? Anesthetized fish? This is important to know. Are these flashes artifacts or a visual signal that these fish put out? Do they do this while actively swimming?

Two sentences in lines 38-45 are awkward, perhaps rephrasing them would help clarify what the author is trying to say.

P7 Panel Fig 3A ... What is the difference between these 11 images? Different time scale? They all look the same to me. How were the data in the plot underneath derived? Average of some number of pixels? This chart does not visually jive with the images above. Get rid of 0 and 12, since there are no corresponding sequences above.

Fig. 3B Three sequences are shown in the plot, but only two images are shown.

Again, to me, there is not an appreciable difference between the iridescent spots here, looking at the image at the top and the one at the bottom. Spot iv looks a little different perhaps, and there is a tiny change in vi, which does not show up in your corresponding plot. The other three spots do not look very different at all.

Line 43 brilliance ... The author uses different terminology throughout.

Intensity relative to grey may be better?

P8 Line12 Should not be strongly related to blood flow [...] Innervation by the neural network [...]

This conclusion is based on what? More explanation is needed here.

Line 15 Brilliance again. See above

Line 19-20 [...] should exist inside the parietal skin
Unless there is evidence, this may need to be stated more as a question than a statement.

Figure 4

Delete separately in line 45

Line 48, summation in RGB colors ... this needs to be explained. Meta analysis ... this needs explanation as well.

Figure 5

5C needs more explanation to be useful. It is unclear whether one scenario explains lack of reflectance and the other scenario explains reflectance under magnetic field stimulation?

The magnetic field experiment needs more explanation, including more methods. Why was this done in the first place? What was the purpose? Why not use the spp that you studied? Why a bream?

P11 Line 20 Reference for lanternfish paper, please.

P11

This entire page, in my opinion, needs to be re-structured, made more succinct and clear. The author talks about applying a magnetic field to bream (why bream, what about *A. lacunosus*?) guanine crystals and observing reflective changes. This moves away from the actual biology of this animal, which is fine. It moves into a potential realm of application, although it remains a mystery to me how applying changes in magnetic field direction (how about strength?) would be a realistic application potential. The author provides limited insight into what the magnetic field experiments add to this study.

The flow of this entire page needs to be improved. Sentences and paragraphs jump back and forth with little connection.

Supporting information. P15-18

Line 36 Radiance... terminology... with your method, you cannot talk about radiance here.

Line 39 radiance [...] disappeared ... Again, terminology re radiance. Also, reflectance did not disappear, it would have been reflected back downwards. But yes, it is not transmitted, which is the point you are trying to make.

Line 43 Particles, crystals, terminology is used interchangeably throughout the manuscript.
Delete BIOGENIC

Figure S1

How were these angles controlled? What exactly were these angles? Estimating them from the drawing is not possible.

What was the light source? Point light or scattered? Please make sure this information is in the materials section.

This figure needs to be explained a lot more to be useful.

The images all more or less look the same to me, there is not a huge difference, perhaps a slight decrease in reflectance between a,b,c compared to d,e but other than that, it is unclear why a,b and c are here, what the arrows show, and why d and e are here.

Figure S2

This idea of blood flow needs more careful explanation. The idea itself makes some sense to me but the supporting evidence shown here does not convince me. The images shown all look the

same to me. Perhaps I'm missing something but the author is also not pointing me in a particular direction. More detailed explanation is necessary.

Reviewer: 3

Comments to the Author(s)

The author describes a flickering reflective pattern in a spot in the skin of a fish. The authors describe the flickering and suggest that it is actively controlled by the fish, who continuously rearranges guanine crystals in the spot to change the reflective patterns. They experimentally support this conjecture by showing that guanine crystals in solution can be arranged by magnetic fields. This is an interesting observation, and should set the stage for more mechanistic investigations into the phenomenon. The science is sound, but I had to read it a few times before I fully understood the story. The authors should clarify it a bit more by, for example, using the common name of the fish as well as the scientific name and outlining the motivation of the study more clearly.

Otherwise I just have a few comments and suggestions.

P2L18: why are they considered important and interesting?

P2L49: "light switching panel" is not clear. Do you mean the lights are turning off and on, or flashing, or what?

P2L51: How do you know it's active?

P4L14: green and blue colors are typically also structural colors. Do you know how these colors are made?

P8L13: Is this just normal movement of the scales? Do other fish do this?

P10L53: Neat experiment!

===PREPARING YOUR MANUSCRIPT===

If you have been asked to revise the written English in your submission as a condition of publication, you must do so, and you are expected to provide evidence that you have received

language editing support. The journal would prefer that you use a professional language editing service and provide a certificate of editing, but a signed letter from a colleague who is a native speaker of English is acceptable. Note the journal has arranged a number of discounts for authors using professional language editing services (<https://royalsociety.org/journals/authors/benefits/language-editing/>).

===PREPARING YOUR REVISION IN SCHOLARONE===

-- If you have uploaded ESM files, please ensure you follow the guidance at <https://royalsociety.org/journals/authors/author-guidelines/#supplementary-material> to

include a suitable title and informative caption. An example of appropriate titling and captioning may be found at https://figshare.com/articles/Table_S2_from_Is_there_a_trade-off_between_peak_performance_and_performance_breadth_across_temperatures_for_aerobic_scope_in_teleost_fishes_/3843624.

Author's Response to Decision Letter for (RSOS-201578.R0)

See Appendices B & C.

RSOS-201578.R1 (Revision)

Review form: Reviewer 1

Is the manuscript scientifically sound in its present form?

Yes

Are the interpretations and conclusions justified by the results?

Yes

Is the language acceptable?

Yes

Do you have any ethical concerns with this paper?

No

Have you any concerns about statistical analyses in this paper?

No

Recommendation?

Accept as is

Comments to the Author(s)

The paper has been significantly improved upon revision and is an important contribution to the field of structural coloration. One minor comment is that the author has removed any mention of 'iridophores', perhaps due to a misunderstanding of one the reviewers comments. In the revised ms iridophore has been substituted for 'circular cell'. Given their reflective properties and location in the fish skin (and the well-known fish chromatophore system), then I think it would be perfectly acceptable to term the circular cells 'iridophores' or 'iridophore cells' throughout the paper.

On a similar note, the author has also removed most mention of guanine crystals in the paper in exchange for 'particles'. I appreciate there is no direct spectroscopic or diffraction data proving the particles are guanine in this paper, but, given everything we know, then it would be very fair

to assume they are indeed guanine. I think it would be entirely reasonable for the author to discuss the optical phenomenon in terms of guanine crystal movement, rather than just 'particles'. One option would be that in the first occurrence in the results section then the term 'particles' be followed with the parenthesis "(presumably guanine crystals (refs))". Thereafter I think the terms particles/platelets/crystals or even guanine crystals could be used interchangeably.

Review form: Reviewer 2

Is the manuscript scientifically sound in its present form?

No

Are the interpretations and conclusions justified by the results?

No

Is the language acceptable?

Yes

Do you have any ethical concerns with this paper?

No

Have you any concerns about statistical analyses in this paper?

No

Recommendation?

Accept with minor revision (please list in comments)

Comments to the Author(s)

Iwasaka manuscript, RSoc Open Science

I reviewed an earlier version of this manuscript and agreed to re-review it. The author has made substantial changes to the manuscript and has addressed most of the comments that were raised earlier. However, in revising the manuscript, its figures, removing data and adding new data to this paper, a few new concerns have come up for me. Most of these may merely require revising the manuscript again, although I do have some substantial concerns that may need the author to take out datasets or think of ways to collect these data differently. More details below.

Title- I would replace PART with TRUNK Dorsal part could be head as well. Trunk is the back of a fish.

Abstract line 35: I suggested this before... I would stay away from the term DISPLAY because it is a very specific form of SIGNAL. Instead, I would rewrite this sentence to read... "... use ambient light effectively in biological signalling."

Next sentence: whether they are attractive or not, even to members of the same species, depends on the circumstances and the type of signal. There are certainly deterring visual signals out there. I would delete the sentence. It doesn't add anything to the abstract and nothing is lost by taking it out.

Line 38. After this sentence (ends in PART), move the two last sentences here.

Line 38: more than one spot.

Line 42: what is a flash train?

Last two sentence... move up

Finish abstract with a conclusion and/or why this study matters. I currently find this a weak and uninformative abstract.

Introduction

Line 8: insects do not have dynamic colour change for camouflage. At least the refs 11-16 cannot be used to that effect. An appearance of changing colour can be affected by movement but this is not the mechanism that the author is trying to introduce here, i.e., dynamic colour changes that originate from a structural/physiological etc. change in the skin.

Line 16: and other examples, e.g., spectral colouration, rather than just silvery. Look at some of the papers from NJ Marshall's lab.

Line 25: The author said that they removed the magnetic field part of the study. It is still mentioned here.

Line 28: The aim of this study.... Has the author truly accomplished to confirm that rapid light intensity changes relate to the optomechanics...? I do not see any more information on that anywhere in the manuscript (and also see my later comment on plates in suspension). To truly answer this question the author would have had to employ much higher res imaging techniques than were used, including also techniques that study the structural component in intact cells.

Line 31: Sentence that starts with "It is interesting..." delete that sentence unless it can be made stronger. Just because something is interesting isn't enough justification.

Materials and methods

Overall comment... Is the formatting 2.5, 2.5.1 etc strictly necessary? It reads more like a progress report or a thesis than a scientific paper...? Maybe this is just my opinion.

Lines 42-44: combine these two sentences into one. Temperatures right after mentioning the tanks.

Light source, point 2.4. line 21. SIZE ... OUTLET Does the author mean diameter? Outlet might not be best wording...maybe replace by focused beam or something.

Section on fiber optic measurements

Which wavelengths? How were they measured? Filters? More detail, please. HOWEVER, I would like to add that I do not completely agree with the author about the use of the spectrophotometer in obtaining spectral data. More details below.

Line 42... data were not was

Section 2.5.2 Measurements on LCD screen. Why were these done? I do not follow the reason. A computer LCD screen is far from replicating the reflective properties of the real animal. LCD colours are based on RGB content. Why would you collect spectral data from an LCD screen? Because of intensity changes? If so, all mention of spectra needs to be removed. All data need to be normalized for intensity, calibrated against a white standard, etc.

Fig. 1

b, top and side views? Make this obvious.

c, why do we see the tank set up twice if the point of this fig is that the camera and the spectrometer were used interchangeably during an experiment? I would suggest to just write at the top of the camera box that this location was for camera and spec.

Fig. 1d. This newly added method makes no sense to me. I am guessing that what the author is trying to do is collect data on spectral changes but because the spots are too small to capture with the setup the author employed, they used an LCD screen to obtain these data. For reasons mentioned above, this is not possible. No spectral data are transmitted through an LCD screen. A microscope should have been used here and the fiber optic cable should have been attached to that. There are plenty of papers that document how others have accomplished collecting spectral reflectance data from small reflecting structures. And if the point was to collect data on flash rates, then this needs to be stated clearly and any mention of spectral data needs to be removed.

Caption of Fig. 1 last 4 lines. Not replacement... they were exchanged to be able to record video and spectra. Ideally, to be able to do both, one would use a scope with multiple mounts that are either functional at the same time (because of beam splitters) or one switches back and forth between the ports, rather than having to unscrew the camera, place the spec, unscrew that again and put the camera back in place. The author may want to be transparent about this and state that there was nothing lost in terms of data acquisition.

Results and discussion section, lines 29-35: LCD vs wavelength... see my earlier comment.

Line 37 Author writes SHOULD BE but this needs to be WAS?

Lines 37-44: why does the author call this a possible artifact? Could this be the underlying mechanism? Or is the author suggesting the possibility that there really are no dynamic changes at all but what we see are merely the result of a change in viewing angle that gives this the appearance of colour change when in actual fact, nothing changes at all? Multilayer reflectors do that... they change colour depending on the direction from which they are viewed. This is without any active changes inside the reflector.

Line 53: I am more comfortable about the author using an LCD screen to get a better idea about flash rate although those data were already obtained using the video camera. So, I'm not sure why we are seeing different ways of doing the same thing.

Line 59/60 and following page 3-5. Sentences on blood flow. I would remove this altogether. Maybe you have more evidence down the line worth a follow up study but this comes without any evidence, so I would suggest removing the sentences.

Line 22, p7 of manuscript. Section on floating particles. If reflecting plates are placed in suspension, they will do what the author describes because they are moving around relative to the light source. I do not understand why this is important to mention here? Perhaps a little more detail is needed to make this point clear. Is the author using this as evidence that this fish species' reflecting system is based on an array of ordered platelets? I would like to point out that even collagen, and many other cellular structures, will reflect light if teased apart and placed in suspension. It's merely a difference in refractive index that will cause this. Others (especially the older literature from the 1960s and 1970s, namely Denton and others) have used this technique to determine refractive indices of reflective plates but, in my opinion, just placing reflectors inside a liquid does not give us much of anything.

Conclusion

Line 34: EXPOSURE... this sounds like the light is causing the flashes, i.e., is there a photosensitive aspect to this? of not, I would merely state "under white light illumination"
General comment about the conclusion... Is there anything else? I think this paper needs a general discussion and subsequent conclusions about why this study is important, how it relates to other studies on fish, cephalopods. See more thoughts at the end of my review.

Fig. 3

Why arbitrary units? Were these not calibrated to give you an intensity relative to a white standard? Sometimes these arbitrary units were a log scale apart...? Data could/should be normalized?

This fig needs a lot more work to be clear. The caption is very confusing an all over the place. The order of the panels inside this figure are also confusing and given that the author is trying to report data from two specimens I would expect to see the same order of panels for each specimen inside this figure.

Fig. 4

As with Fig. 3... arbitrary units that are all over the place. The layout of the figure is also very confusing. images and panels need to be aligned, labelled, etc.

Fig. 5.

Label dendrites.

A general comment of Figures.

The author is presenting data on a species of fish that shows reflecting flashes. This is the story of this paper. In my opinion, there are way too many Figs to make this point. Given the simplicity of what the author is reporting, the Figures are over-complicating the story, adding data that do not add to the story (e.g., recording spectral data from an LCD screen), even adding the spectrophotometer data on flash rate when the author was already able to obtain these data using the video camera. I would suggest condensing the Figs so that there is one figure showing images, not several, and one Fig that shows flash rates, not several. Also, once the author has revised the method for collecting spectral data, one Fig. that shows these data, not several.

FINAL COMMENT:

I continue to like this story, and I do not need convincing that these types of studies are interesting and necessary. However, not everyone agrees on this and I do think a focus on a bigger picture is necessary. There are many examples in the existing literature that report colour changes in fish. How is this study different? Why does a reader need to read this paper? What does the study add to what we know? What are the shortcomings, what are the improvements that could be made? Where does this study lead us in the future? These are only a small number of thoughts that could guide the author in revising the discussion and conclusion sections. I do not suggest adding lengthy sentences because I do like the conciseness of the current manuscript. After all, this is a "brief communication"- type paper and it needs to remain brief. But it is currently lacking important information.

Review form: Reviewer 3

Is the manuscript scientifically sound in its present form?

Yes

Are the interpretations and conclusions justified by the results?

Yes

Is the language acceptable?

Yes

Do you have any ethical concerns with this paper?

No

Have you any concerns about statistical analyses in this paper?

No

Recommendation?

Accept with minor revision (please list in comments)

Comments to the Author(s)

The author has addressed my suggestions well. I just have a few wording suggestions, and questions.

P21L14: "...a long history of.."

P21L26: "...platelets, and observed.."

P21L43: What is an open aquarium? Is it outside?

P22L41: "Any data thought to have been collected.."

P22L50: ..and fixed with.."

P22L53: A stand was used for what purpose?
P25L51: "...spots was carried out.."
P25L57: time->period
P26L7: "...spots captured.."
P26L9: literatures->studies
P26L10: "...contains the black pigment melanin"
P26L16L "...a piece of dorsal skin.."
P26L28: "...appeared random.."

Decision letter (RSOS-201578.R1)

Dear Professor Iwasaka

On behalf of the Editors, we are pleased to inform you that your Manuscript RSOS-201578.R1 "Flashing spots on the dorsal part of hardyhead silverside fish" has been accepted for publication in Royal Society Open Science subject to minor revision in accordance with the referees' reports. Please find the referees' comments along with any feedback from the Editors below my signature.

Please submit your revised manuscript and required files (see below) no later than 7 days from today's (ie 25-Jan-2021) date. Note: the ScholarOne system will 'lock' if submission of the revision is attempted 7 or more days after the deadline. If you do not think you will be able to meet this deadline please contact the editorial office immediately.

on behalf of Dr Silvia Vignolini (Associate Editor) and Pietro Cicuta (Subject Editor)
openscience@royalsociety.org

Reviewer comments to Author:

Reviewer: 1

Comments to the Author(s)

The paper has been significantly improved upon revision and is an important contribution to the field of structural coloration. One minor comment is that the author has removed any mention of 'iridophores', perhaps due to a misunderstanding of one the reviewers comments. In the revised ms iridophore has been substituted for 'circular cell'. Given their reflective properties and location in the fish skin (and the well-known fish chromatophore system), then I think it would be perfectly acceptable to term the circular cells 'iridophores' or 'iridophore cells' throughout the paper.

On a similar note, the author has also removed most mention of guanine crystals in the paper in exchange for 'particles'. I appreciate there is no direct spectroscopic or diffraction data proving the particles are guanine in this paper, but, given everything we know, then it would be very fair to assume they are indeed guanine. I think it would be entirely reasonable for the author to discuss the optical phenomenon in terms of guanine crystal movement, rather than just 'particles'. One option would be that in the first occurrence in the results section then the term 'particles' be followed with the parenthesis "(presumably guanine crystals (refs))". Thereafter I think the terms particles/platelets/crystals or even guanine crystals could be used interchangeably.

Reviewer: 2

Comments to the Author(s)

Iwasaka manuscript, RSoc Open Science

I reviewed an earlier version of this manuscript and agreed to re-review it. The author has made substantial changes to the manuscript and has addressed most of the comments that were raised earlier. However, in revising the manuscript, its figures, removing data and adding new data to this paper, a few new concerns have come up for me. Most of these may merely require revising the manuscript again, although I do have some substantial concerns that may need the author to take out datasets or think of ways to collect these data differently. More details below.

Title- I would replace PART with TRUNK Dorsal part could be head as well. Trunk is the back of a fish.

Abstract line 35: I suggested this before... I would stay away from the term DISPLAY because it is a very specific form of SIGNAL. Instead, I would rewrite this sentence to read... "... use ambient light effectively in biological signalling."

Next sentence: whether they are attractive or not, even to members of the same species, depends on the circumstances and the type of signal. There are certainly deterring visual signals out there. I would delete the sentence. It doesn't add anything to the abstract and nothing is lost by taking it out.

Line 38. After this sentence (ends in PART), move the two last sentences here.

Line 38: more than one spot.

Line 42: what is a flash train?

Last two sentence... move up

Finish abstract with a conclusion and/or why this study matters. I currently find this a weak and uninformative abstract.

Introduction

Line 8: insects do not have dynamic colour change for camouflage. At least the refs 11-16 cannot be used to that effect. An appearance of changing colour can be affected by movement but this is not the mechanism that the author is trying to introduce here, i.e., dynamic colour changes that originate from a structural/physiological etc. change in the skin.

Line 16: and other examples, e.g., spectral colouration, rather than just silvery. Look at some of the papers from NJ Marshall's lab.

Line 25: The author said that they removed the magnetic field part of the study. It is still mentioned here.

Line 28: The aim of this study.... Has the author truly accomplished to confirm that rapid light intensity changes relate to the optomechanics...? I do not see any more information on that anywhere in the manuscript (and also see my later comment on plates in suspension). To truly answer this question the author would have had to employ much higher res imaging techniques than were used, including also techniques that study the structural component in intact cells.

Line 31: Sentence that starts with "It is interesting..." delete that sentence unless it can be made stronger. Just because something is interesting isn't enough justification.

Materials and methods

Overall comment... Is the formatting 2.5, 2.5.1 etc strictly necessary? It reads more like a progress report or a thesis than a scientific paper...? Maybe this is just my opinion.

Lines 42-44: combine these two sentences into one. Temperatures right after mentioning the tanks.

Light source, point 2.4. line 21. SIZE ... OUTLET Does the author mean diameter? Outlet might not be best wording...maybe replace by focused beam or something.

Section on fiber optic measurements

Which wavelengths? How were they measured? Filters? More detail, please. HOWEVER, I would like to add that I do not completely agree with the author about the use of the spectrophotometer in obtaining spectral data. More details below.

Line 42... data were not was

Section 2.5.2 Measurements on LCD screen. Why were these done? I do not follow the reason. A computer LCD screen is far from replicating the reflective properties of the real animal. LCD colours are based on RGB content. Why would you collect spectral data from an LCD screen?

Because of intensity changes? If so, all mention of spectra needs to be removed. All data need to be normalized for intensity, calibrated against a white standard, etc.

Fig. 1

b, top and side views? Make this obvious.

c, why do we see the tank set up twice if the point of this fig is that the camera and the spectrometer were used interchangeably during an experiment? I would suggest to just write at the top of the camera box that this location was for camera and spec.

Fig. 1d. This newly added method makes no sense to me. I am guessing that what the author is trying to do is collect data on spectral changes but because the spots are too small to capture with the setup the author employed, they used an LCD screen to obtain these data. For reasons mentioned above, this is not possible. No spectral data are transmitted through an LCD screen. A microscope should have been used here and the fiber optic cable should have been attached to that. There are plenty of papers that document how others have accomplished collecting spectral reflectance data from small reflecting structures. And if the point was to collect data on flash rates, then this needs to be stated clearly and any mention of spectral data needs to be removed.

Caption of Fig. 1 last 4 lines. Not replacement... they were exchanged to be able to record video and spectra. Ideally, to be able to do both, one would use a scope with multiple mounts that are either functional at the same time (because of beam splitters) or one switches back and forth

between the ports, rather than having to unscrew the camera, place the spec, unscrew that again and put the camera back in place. The author may want to be transparent about this and state that there was nothing lost in terms of data acquisition.

Results and discussion section, lines 29-35: LCD vs wavelength... see my earlier comment.

Line 37 Author writes SHOULD BE but this needs to be WAS?

Lines 37-44: why does the author call this a possible artifact? Could this be the underlying mechanism? Or is the author suggesting the possibility that there really are no dynamic changes at all but what we see are merely the result of a change in viewing angle that gives this the appearance of colour change when in actual fact, nothing changes at all? Multilayer reflectors do that... they change colour depending on the direction from which they are viewed. This is without any active changes inside the reflector.

Line 53: I am more comfortable about the author using an LCD screen to get a better idea about flash rate although those data were already obtained using the video camera. So, I'm not sure why we are seeing different ways of doing the same thing.

Line 59/60 and following page 3-5. Sentences on blood flow. I would remove this altogether.

Maybe you have more evidence down the line worth a follow up study but this comes without any evidence, so I would suggest removing the sentences.

Line 22, p7 of manuscript. Section on floating particles. If reflecting plates are placed in suspension, they will do what the author describes because they are moving around relative to the light source. I do not understand why this is important to mention here? Perhaps a little more detail is needed to make this point clear. Is the author using this as evidence that this fish species' reflecting system is based on an array of ordered platelets? I would like to point out that even collagen, and many other cellular structures, will reflect light if teased apart and placed in suspension. It's merely a difference in refractive index that will cause this. Others (especially the older literature from the 1960s and 1970s, namely Denton and others) have used this technique to determine refractive indices of reflective plates but, in my opinion, just placing reflectors inside a liquid does not give us much of anything.

Conclusion

Line 34: EXPOSURE... this sounds like the light is causing the flashes, i.e., is there a photosensitive aspect to this? of not, I would merely state "under white light illumination"

General comment about the conclusion... Is there anything else? I think this paper needs a general discussion and subsequent conclusions about why this study is important, how it relates to other studies on fish, cephalopods. See more thoughts at the end of my review.

Fig. 3

Why arbitrary units? Were these not calibrated to give you an intensity relative to a white standard? Sometimes these arbitrary units were a log scale apart...? Data could/should be normalized?

This fig needs a lot more work to be clear. The caption is very confusing and all over the place. The order of the panels inside this figure are also confusing and given that the author is trying to report data from two specimens I would expect to see the same order of panels for each specimen inside this figure.

Fig. 4

As with Fig. 3... arbitrary units that are all over the place. The layout of the figure is also very confusing. images and panels need to be aligned, labelled, etc.

Fig. 5.

Label dendrites.

A general comment of Figures.

The author is presenting data on a species of fish that shows reflecting flashes. This is the story of this paper. In my opinion, there are way too many Figs to make this point. Given the simplicity of what the author is reporting, the Figures are over-complicating the story, adding data that do not add to the story (e.g., recording spectral data from an LCD screen), even adding the spectrophotometer data on flash rate when the author was already able to obtain these data using the video camera. I would suggest condensing the Figs so that there is one figure showing images, not several, and one Fig that shows flash rates, not several. Also, once the author has revised the method for collecting spectral data, one Fig. that shows these data, not several.

FINAL COMMENT:

I continue to like this story, and I do not need convincing that these types of studies are interesting and necessary. However, not everyone agrees on this and I do think a focus on a bigger picture is necessary. There are many examples in the existing literature that report colour changes in fish. How is this study different? Why does a reader need to read this paper? What does the study add to what we know? What are the shortcomings, what are the improvements that could be made? Where does this study lead us in the future? These are only a small number of thoughts that could guide the author in revising the discussion and conclusion sections. I do not suggest adding lengthy sentences because I do like the conciseness of the current manuscript. After all, this is a "brief communication"- type paper and it needs to remain brief. But it is currently lacking important information.

Reviewer: 3

Comments to the Author(s)

The author has addressed my suggestions well. I just have a few wording suggestions, and questions.

P21L14: "...a long history of.."

P21L26: "...platelets, and observed.."

P21L43: What is an open aquarium? Is it outside?

P22L41: "Any data thought to have been collected.."

P22L50: ..and fixed with.."

P22L53: A stand was used for what purpose?

P25L51: "...spots was carried out.."

P25L57:time->period

P26L7:"..spots captured.."

P26L9: literatures->studies

P26L10: "...contains the black pigment melanin"

P26L16L "...a piece of dorsal skin.."

P26L28:"..appeared random.."

===PREPARING YOUR MANUSCRIPT===

===PREPARING YOUR REVISION IN SCHOLARONE===

Author's Response to Decision Letter for (RSOS-201578.R1)

See Appendices D & E.

RSOS-201578.R2 (Revision)

Review form: Reviewer 2

Is the manuscript scientifically sound in its present form?

Yes

Are the interpretations and conclusions justified by the results?

Yes

Is the language acceptable?

Yes

Do you have any ethical concerns with this paper?

No

Have you any concerns about statistical analyses in this paper?

No

Recommendation?

Accept with minor revision (please list in comments)

Comments to the Author(s)

Abstract

Line 37 of iridophores, whose (comma)

Line 39 delete WHICH SHOWS LIGHT FLASHING because you mention it in the following sentence.

Line 43 UNDER ILLUMINATION... I would replace with WHEN VIEWED UNDER WHITE LIGHT

Last sentence: I would revise this sentence more along the lines of why the study is important and what we can learn from this study...

Incidentally: A paper by Mathger et al 2003 J Exp Biol 206: 3607-3613 reported fast reflective changes (continuous flashing) in a tropical teleost, so the phenomenon you are reporting is not completely unique hardyheads. However, your study adds to a growing body of literature reporting colour changes in an animal group that was (and still is) largely considered to have static colouration. I would finish the sentence with something along these lines.

Introduction

Line 6 The author claims colour changes in a mammal and uses a chameleon paper as a reference? (Reptile...)

Materials and methods

Section on Fiber optic measurements

Line 32 Rewrite sentence along these lines: The spectrometer was equipped with a diffraction grating that simultaneously collected light ranging from 450 nm to 650 nm.

Line 38 This comment is regarding the arbitrary units (I commented on this previously). As a seasoned spectrometer user myself, I suspect that the author's "arbitrary" units are relative to a white standard. Assuming that the set-up was calibrated before each measurement and specimens were not moved, magnification not changed, total reflective area inside field of view remained steady, these measurements can be presented on a scale from 0-1, or 0-100%, whichever the author prefers. The units are therefore not arbitrary. Of course, you would not normalize all of these data. You would not want to cancel out the intensity changes you are trying to demonstrate, but I would suggest getting away from using "arbitrary".

Line 45 WITHOUT SCREWING.... Maybe find a better term here... Or just take it out. All that matters is that it was a quick equipment change, and that you didn't miss anything. Next sentence: Maybe re-write sentence? E.g., No crucial data were missed during spectrometer/camera exchange.

Line 46 Exchange, rather than REPLACEMENT

Line 47 Drifting? Explain, please. Specimen moved outside of field of view etc.

Line 49 Re-write perhaps? ... thought to have been collected while the region of interest drifted ...?

Important information missing here: Calibration... what kind of reflectance standard was used? Diameter of fiber optic cable?

Section on measurements off LCD screen

Line 54 Make clearer WHY this was done... To magnify reflecting areas sufficiently for recording...

The first two sentences could also be combined, they are a little bit repetitive.

Last sentence: Did you not use a white standard before? I would reword this and state that data WERE (not WAS) given RELATIVE TO a white standard (not normalized)

Fig. 1C I would say EXCHANGE, rather than replacement. No need for both terms

Fig. 4 Why is the start time in (a) 25 sec but the x-axis starts at zero? In (b), start time is 0 sec and the x-axis starts at zero too? Just a minor comment but it gets a reader confused. In reality, you can probably get rid of the two start times above the images.

Now the y-axis is normalized? How come? This is inconsistent compared to Fig. 3. See my comments above about the arbitrary units. These are presumably relative to a white standard.

Fig. 5 Maybe make all images the same size and align them? Fig looks a tad careless.

Final comment: The author has improved the manuscript tremendously. I only have one last suggestion, which I raised previously and still feel needs to be addressed. How does this study compare to others in the field? How is it different? Is the flashing faster, slower, spans wider wavelength, etc., than what other studies have reported? As it stands, this is a very brief report that lacks a bigger picture.

Decision letter (RSOS-201578.R2)

Dear Professor Iwasaka

On behalf of the Editors, we are pleased to inform you that your Manuscript RSOS-201578.R2 "Flashing spots on the dorsal trunk of hardyhead silverside fish" has been accepted for publication in Royal Society Open Science subject to minor revision in accordance with the referees' reports. Please find the referees' comments along with any feedback from the Editors below my signature.

Please submit your revised manuscript and required files (see below) no later than 7 days from today's (ie 11-Mar-2021) date. Note: the ScholarOne system will 'lock' if submission of the revision is attempted 7 or more days after the deadline. If you do not think you will be able to meet this deadline please contact the editorial office immediately.

Best regards,
Lianne Parkhouse
Editorial Coordinator

on behalf of Professor Pietro Cicuta (Subject Editor)
 openscience@royalsociety.org

Associate Editor Comments to Author:

A number of suggestions are included by the referee that will add value to your work. In particular, please be sure to add a few sentences, or a short paragraph, that more clearly situates your work within the wider field, as suggested by the reviewer - this will help readers understand the context in which your research exists.

Reviewer comments to Author:

Reviewer: 2
 Comments to the Author(s)

Abstract

Line 37 of iridophores, whose (comma)

Line 39 delete WHICH SHOWS LIGHT FLASHING because you mention it in the following sentence.

Line 43 UNDER ILLUMINATION... I would replace with WHEN VIEWED UNDER WHITE LIGHT

Last sentence: I would revise this sentence more along the lines of why the study is important and what we can learn from this study...

Incidentally: A paper by Mathger et al 2003 J Exp Biol 206: 3607-3613 reported fast reflective changes (continuous flashing) in a tropical teleost, so the phenomenon you are reporting is not completely unique hardyheads. However, your study adds to a growing body of literature reporting colour changes in an animal group that was (and still is) largely considered to have static colouration. I would finish the sentence with something along these lines.

Introduction

Line 6 The author claims colour changes in a mammal and uses a chameleon paper as a reference? (Reptile...)

Materials and methods

Section on Fiber optic measurements

Line 32 Rewrite sentence along these lines: The spectrometer was equipped with a diffraction grating that simultaneously collected light ranging from 450 nm to 650 nm.

Line 38 This comment is regarding the arbitrary units (I commented on this previously). As a seasoned spectrometer user myself, I suspect that the author's "arbitrary" units are relative to a white standard. Assuming that the set-up was calibrated before each measurement and specimens were not moved, magnification not changed, total reflective area inside field of view remained steady, these measurements can be presented on a scale from 0-1, or 0-100%, whichever the author prefers. The units are therefore not arbitrary. Of course, you would not normalize all of these data. You would not want to cancel out the intensity changes you are trying to demonstrate, but I would suggest getting away from using "arbitrary".

Line 45 WITHOUT SCREWING.... Maybe find a better term here... Or just take it out. All that matters is that it was a quick equipment change, and that you didn't miss anything. Next sentence: Maybe re-write sentence? E.g., No crucial data were missed during spectrometer/camera exchange.

Line 46 Exchange, rather than REPLACEMENT

Line 47 Drifting? Explain, please. Specimen moved outside of field of view etc.

Line 49 Re-write perhaps? ... thought to have been collected while the region of interest drifted ...?

Important information missing here: Calibration... what kind of reflectance standard was used?
Diameter of fiber optic cable?

Section on measurements off LCD screen

Line 54 Make clearer WHY this was done... To magnify reflecting areas sufficiently for recording...

The first two sentences could also be combined, they are a little bit repetitive.

Last sentence: Did you not use a white standard before? I would reword this and state that data WERE (not WAS) given RELATIVE TO a white standard (not normalized)

Fig. 1C I would say EXCHANGE, rather than replacement. No need for both terms

Fig. 4 Why is the start time in (a) 25 sec but the x-axis starts at zero? In (b), start time is 0 sec and the x-axis starts at zero too? Just a minor comment but it gets a reader confused. In reality, you can probably get rid of the two start times above the images.

Now the y-axis is normalized? How come? This is inconsistent compared to Fig. 3. See my comments above about the arbitrary units. These are presumably relative to a white standard.

Fig. 5 Maybe make all images the same size and align them? Fig looks a tad careless.

Final comment: The author has improved the manuscript tremendously. I only have one last suggestion, which I raised previously and still feel needs to be addressed. How does this study compare to others in the field? How is it different? Is the flashing faster, slower, spans wider wavelength, etc., than what other studies have reported? As it stands, this is a very brief report that lacks a bigger picture.

===PREPARING YOUR MANUSCRIPT===

===PREPARING YOUR REVISION IN SCHOLARONE===

Author's Response to Decision Letter for (RSOS-201578.R2)

See Appendix F.

Decision letter (RSOS-201578.R3)

Dear Professor Iwasaka,

I am pleased to inform you that your manuscript entitled "Flashing spots on the dorsal trunk of hardyhead silverside fish" is now accepted for publication in Royal Society Open Science.

on behalf of Professor Pietro Cicuta (Subject Editor)
openscience@royalsociety.org

Appendix A

Iwasaka reports an intriguing phenomenon of 'iridophore blinking' in the skin of the Hardyhead silverside fish. This behavior is attributed to periodic and aperiodic fluctuations in the angle of intracellular guanine platelets. The observation itself is extremely interesting and, as far as I am aware, is the first report of 'blinking' behavior in guanine-containing iridophores. However, the paper is poorly written, lacking appropriate referencing and often confusing. It lacks information needed to support the conclusions or that would clarify the results.

There is an important underlying phenomenon at the core of this paper, but it needs an overhaul in order to justify the conclusions and clearly articulate the message to reader.

Comments:

Pg2, paragraph 5: references required to support (a) counter-shading and (b) silvering camouflage of fish.

Pg2, paragraph 6:

I do not understand this paragraph. Is this meant to be a general statement about the presence of photophores in fish and the role of guanine crystals within them? If so, consider revising to:

"Some species of fish contain light-emitting cells (photophores) in the ventral side of the body. The photophores.....The role of reflective crystals such as guanine within these photophores needs further clarification".

Pg3, paragraph 1: lacks referencing.

Pg5, Fig. 2: There needs to be some more information here to explain what experiment is being performed. What is meant by 'light irradiation'? Presumably the iridophores are being exposed to light the whole time they are being imaged, so does this mean an additional, more powerful light source is added along, or perpendicular with the optical axes? Then comparison is made with the behavior of the iridophores under ambient conditions? Is the left-hand image in Fig. 2a showing the behavior ambient light conditions? If so, why is the 'light' arrow directed towards this and not the right-hand image? In Fig. 2b what do the blue arrows signify? Light on? Time since exposure?

Pg6, line 25/26: lacks referencing.

Pg6, paragraph 4: The first half of the paragraph reads as a discussion or conclusion and not as results. Either it should be removed to the end of the paper, or to the end of this paragraph. The confusion of "this study found for the first time..." comes before we have even been introduced to the blinking behaviour "Fig. 3 shows....".

Pg6: final paragraph needs revision. I cannot understand it.

Pg7: Fig. 3: The blinking behavior in the images is not obvious at all with the naked eye (at least in comparison to the spectacular behavior shown in MovieS2,3. Can the images be made larger to show the blinking more clearly? Would the blinking be more obvious if the images showed a longer time range (e.g. over several seconds?). In the plots of brilliance changes, how do we know these changes are significant and not just noise? Are these changes normalized to something reasonable to establish the significance of the changes? For example, on the lower bound, what would the grey scale value of a melanophore be? On the upper bound, what is the grey scale value of a perfectly-oriented iridophore?

Fig.S2: needs to be explained in much more detail. I do not understand what is being proposed here about the blood stream. What do the arrows mean in B? No explanation is given at all in the caption. Is there a time difference between all the images? If so, what is it?

In movie S2 and S3, are these real-time movies? What is the time axes? The relevant experimental section for movies is absent!

Pg. 7/8. Figs 3,4. The images need to be made larger.

Pg. 11, line 10-14: this is not true. The blue coloring does not result from changes in the platelet spacing and angle. The blue color is a product of the platelet spacings and angles, and changes in the reflected color may be induced by changes in these 2 parameters. What is meant by pre-existing red color?

Pg. 12: no concluding section

Appendix B

Dear Reviewers and Editors,

I hope you are keeping well at this difficult time. I appreciate your support of my manuscript.

I have revised the following manuscript according to the three Reviewers' comments.

Manuscript ID: RSOS-201578

Former title of the manuscript: Rapid active-reflection display on dorsal stripes of *Atherinomorus lacunosus*.

Revised title: Flashing spots on the dorsal part of hardyhead silverside fish

Author: Masakazu Iwasaka

Thank you very much for reading my manuscript and spending time on providing a review. According to the very important comments from all reviewers, I have revised the paragraph explaining the motivation of this study. The revised manuscript simply focuses on the dramatic phenomenon of the flashing in dorsal spots of hardyhead silverside fish. A statement about the motivation is now included in the Introduction, the experimental methods have been expanded, and the results are explained using statements that describe the obtained evidence.

I hope this revision suitably responds to the comments provided in the initial review.

Sincerely

Masakazu Iwasaka, Prof.
Research Institute for Nanodevice and Bio Sysrems
Hiroshima University
1-4-2 Kagamiyama, Higashihiroshima
739-8527 Hiroshima, Japan
miwamasa@hiroshima-u.ac.jp

Responses to Reviewer comments:

Responses to Reviewer 1

General comment: Iwasaka reports an intriguing phenomenon of ‘iridophore blinking’ in the skin of the Hardyhead silverside fish. This behavior is attributed to periodic and aperiodic fluctuations in the angle of intracellular guanine platelets. The observation itself is extremely interesting and, as far as I am aware, is the first report of ‘blinking’ behavior in guanine-containing iridophores. However, the paper is poorly written, lacking appropriate referencing and often confusing. It lacks information needed to support the conclusions or that would clarify the results. There is an important underlying phenomenon at the core of this paper, but it needs an overhaul in order to justify the conclusions and clearly articulate the message to reader.

Response: I appreciate the very important comments from the reviewer. The manuscript has been revised on the basis of the reviewer’s comments. All of the figures were revised to include additional experimental data. The observed phenomenon seemed to arise from ‘guanine-containing iridophores’. In the revised manuscript, to more accurately describe the evidence obtained, ‘iridophore’ was changed to ‘circular cell’. At the end of Results and Discussion, it is suggested that the ‘circular cell’ with a size of 7–10 μm may be an iridophore.

Based on the same reasoning, ‘reflecting particle’ is used instead of ‘guanine’. The conclusions now only include statements that describe the obtained evidence.

Comment 1) :Pg2, paragraph 5: references required to support (a) counter-shading and (b) silvering camouflage of fish.

Response: I added references concerning countershading and silvering camouflage of fish in the following sentences.

“There is long history of research on fish body colors concerning their purpose, such as counter-shading [25, 26] and silvering camouflage of fish [20].

20.Jordan TM, Partridge JC, Roberts NW. 2012 Non-polarizing broadband multilayer re-reflectors in fish. *Nature Photon.* 260, 759-763.(doi: 10.1038/nphoton.2012.260)

25.Thayer, A, 1896 The law which underlies protective coloration. *Auk* 13, 124–129.

26.Rowland, HM, 2009 From Abbott Thayer to the present day: what have we learned about the function of countershading? *Philos. Trans. R. Soc. B-Biol. Sci.* 364, 519–527. (<https://doi.org/10.1098/rstb.2008.0261>).”

Comment 2) :Pg2, paragraph 6:

I do not understand this paragraph. Is this meant to be a general statement about the presence of photophores in fish and the role of guanine crystals within them? If so, consider revising to:

“Some species of fish contain light-emitting cells (photophores) in the ventral side of the body. The photophores.....The role of reflective crystals such as guanine within these photophores needs further clarification”.

Response: I deleted this sentence because this was confusing.

Comment 3) : Pg3, paragraph 1: lacks referencing.

Response: I deleted the paragraph because the data provided in this manuscript does not show evidence for neural activity.

Comment 4) : Pg5, Fig. 2: There needs to be some more information here to explain what experiment is being performed. What is meant by ‘light irradiation’? Presumably the iridophores are being exposed to light the whole time they are being imaged, so does this mean an additional, more powerful light source is added along, or perpendicular with the optical axes? Then comparison is made with the behavior of the iridophores under ambient conditions? Is the left-hand image in Fig. 2a showing the behavior ambient light conditions? If so, why is the ‘light’ arrow directed towards this and not the right-hand image? In Fig. 2b what do the blue arrows signify? Light on? Time since exposure?

Response: More information about the experimental methods have been included in the revised manuscript. I apologize for the old Fig. 2 missing the time indication. As shown in the revised Figure 1, one light source was used. The incident light came from the side of the microscope (or direction of the observation). The scattered light from the skin of the fish was accepted into the microscope lens. As shown in the revised Figure 2, the same intensity and the same direction of the incident light were maintained. Under these conditions, the circular spots independently changed their light intensity. The new data in the revised Fig. 2 correspond to the questions raised regarding the old Fig. 2. Two supporting videos are available for “real-time playback of the recorded video.”

Comment 5) : Pg6, line 25/26: lacks referencing.

Response: The paragraphs were revised. I deleted the sentences referring to insufficient evidence from my data.

Comment 6) : Pg6, paragraph 4: The first half of the paragraph reads as a discussion or conclusion and not as results. Either it should be removed to the end of the paper, or to the end of this paragraph. The conclusion of “this study found for the first time...” comes before we have even been introduced to the blinking behaviour “Fig. 3 shows....”.

Response: According to reviewer’s comment, the sentences were removed and the Results and discussion section was rearranged.

Comment 7) : Pg6: final paragraph needs revision. I cannot understand it.

Response: The confusing sentences were removed from the text.

Comment 8) : Pg7: Fig. 3: The blinking behavior in the images is not obvious at all with the naked eye (at least in comparison to the spectacular behavior shown in MovieS2,3. Can the images be made larger to show the blinking more clearly? Would the blinking be more obvious if the images showed a longer time range (e.g. over several seconds?). In the plots of brilliance changes, how do we know these changes are significant and not just noise? Are these changes normalized to something reasonable to establish the significance of the changes? For example, on the lower bound, what would the grey scale value of a melanophore be? On the upper bound, what is the grey scale value of a perfectly-oriented iridophore?

Response: The old Fig. 3 was replaced with the revised Figure 2, which shows two examples of the phenomenon showing light intensity changes in the spots consisting of circular cells (which may be

iridophores). Fig. 2 shows expanded images and I believe the changes in the images are distinct enough for the naked eye. Based on another reviewer's comments, analyses of the light intensity were improved. Fiber optic measurements directly collecting light from the fish skin were carried out, as shown in the revised Figure 3. In addition, the light intensity changes in the region of interest on the LCD screen were also measured using the same measurement system. According to the reviewer's comments, a comparison of the light intensity at the circular cells (iridophores) with that at a black cell (melanophore) was performed. A discussion on the significance of the light intensity changes focused on the frequencies of the light reflection flash.

The author would like to use the terms 'flash' or 'flashing' instead of 'blinking' because the data also show an increase of one shot of light.

Comment 9) : Fig.S2: needs to be explained in much more detail. I do not understand what is being proposed here about the blood stream. What do the arrows mean in B? No explanation is given at all in the caption. Is there a time difference between all the images? If so, what is it?

Response: The arrows in Fig. S2B refer to the direction stream observed at the edge of the spots area. I apologize for the unclear images. Fig. S2 was removed in the revised manuscript because clear data showing the effect of blood flow on the light flash were not obtained in the additional experiments performed in this revision process. Simply, a sentence referring to the conjecture about the blood flow effect was included in the text as follows.

“In addition to the skin motion, other influencing factors, such as blood flow, may be conjectured. In the present study, most of observed dorsal spots existed apart from blood vessels, and no correlation between the flash and blood flow was detected.”

Comment 10) : In movie S2 and S3, are these real-time movies? What is the time axes? The relevant experimental section for movies is absent!

Response: Yes, the movies are shown in real time, that is, the timing is the same as real time. No adjustment of the speed was carried out. In the revised manuscript, Fig. 2–5 are based on six movies (videos) for each of the presented figures. All of these movies can be viewed in real time. The relevant information for the movies has been added in the revised supporting information.

Comment 11) : Pg. 7/8. Figs 3,4. The images need to be made larger.

Response: In the revised manuscript, all of the images were magnified or a magnified subset image was also included.

Comment 12) : Pg. 11, line 10-14: this is not true. The blue coloring does not result from changes in the platelet spacing and angle. The blue color is a product of the platelet spacings and angles, and changes in the reflected color may be induced by changes in these 2 parameters. What is meant by pre-existing red color?

Response: Thank you for the appropriate corrections. In this revision, I revised the statement about the mechanism of the structural color.

I deleted the term 'pre-existing red color' in the revision. In the first submission, I was trying to say the green color of the spots in hardyhead silverside fish was a product of blue and another color such as red. The newly added images shown in the revised Figure 2 suggests, in my opinion, that the green

color may be the product of blue (made of guanine platelets) and yellow (possibly made of guanine platelets or other particles). However, this conjecture is not supported by enough evidence at this stage. Therefore, I did not make a statement about this point in the revised manuscript.

Comment 13) : Pg. 12: no concluding section

Response: According to the suggestion, a concluding section was added.

Responses to Reviewer 2

General comment: In this manuscript, the author presents an interesting structural colour-changing phenomenon in a fish species. Such colour changes are not uncommon amongst fishes, although the speed of this colour change is unusual. The author uses stereo microscopic imaging by attaching a camera to the microscope to image these colour changes. The author then analyzes the RGB contents of the images to quantify intensity changes of the reflective patterns. A magnetic field is applied to some guanine crystal reflectors in an experimental setup to show that magnetic stimuli can change reflectance.

Reading about this interesting fish species and its fascinating colour change is exciting. I have a series of comments for the author that I hope will improve the manuscript.

Overall, more detail is required in the materials and methods section. The entire manuscript could be made more succinct by reducing sections that do not directly relate to or benefit this paper. The flow of the manuscript would also be improved by ensuring that placement sentences and paragraphs is logical and that an argument is built and discussed fully. The figures need work, e.g., labelling areas of interest that the author would like the reader to look at. There is too much guess work at this point. Some images appear irrelevant, some of the plots do not seem to match the images that the data were (seemingly) derived from.

I will use the outside line numbers on my version of the pdf (there are two, numbered 1-60 and 5-30, I will use the 1-60 numbering).

Response: I appreciate the very important comments from the reviewer. Thank you very much for your advice regarding the spectrophotometry of the light flash. I carried out additional experiments on this in November after receiving the reviewer's comments, and obtained valuable fiber optic measurement data.

The manuscript was revised based on the comments from the reviewer. The Methods section was extended to provide sufficient information about the measurements, handling, etc. The Results and discussion section was entirely replaced with statements based on the available evidence.

Comment 1) :Title: Put FISH in here, or TELEOST, so that this paper can more easily be searched. STRUCTURAL COLORATION could be in here too, or in the keywords. The word DISPLAY should be removed in my opinion. Display in animal behaviour means a signalling intent, which has to be identified as such.

Response: According to the kind comments, the title was revised to "Flashing spots on the dorsal part of hardyhead silverside fish."

Comment 2) :Abstract, p1:

Line 32 coloring species ... poor wording, I suggest you re-word. For example: attractive not only to members of the same colorful species...

Response: According to the comment, the sentence was revised to "...not only to members of the same colorful species but also to humans."

Comment 3) :Line 35 detailed function [...] remains unknown ... This needs to be toned down. There is a lot of literature detailing functions and mechanisms of biological reflectors in a lot of species. There is always more work that can be done to elucidate further, which this paper does.

Response: Thank you for the comment. This sentence was removed from text.

Comment 4) :Line 35 After spp name, add teleost fish here, to make clear what kind of species you are describing here.

Response: According to the comment, the sentence was revised as "*Atherinomorus lacunosus*, a teleost fish, has alignments of..."

Comment 5) :Line 39 rapid reflection changes in platelets There is no evidence for this from this paper. This is likely to be the case assuming that this teleost uses a similar system compared to other fish spp but it sounds like you have direct evidence for this, which is not the case. I would suggest taking this statement out of the abstract.

Response: According to the comment, the sentence was deleted.

Comment 6) :Line 43 passive illumination ... Illumination is an incorrect term. This is structural reflectance. Also, it is passive in terms of using incoming light to reflect back but it appears to be active in terms of control. This sentence is best refined along these lines.

Response: Thank you for the comments on this critical point of the phenomenon. It is hard to explain the detailed mechanism at present. Therefore, this sentence was removed.

Comment 7) :Furthermore, the abstract is incomplete, e.g., magnetic field experiment not mentioned at all.

Response: The revised manuscript did require the magnetic field experiments to obtain the conclusion. Therefore, I think a comment on the magnetic field experiments is not required in the Abstract.

Comment 8) :Introduction, manuscript (not pdf) pp 2-3:

Line 7 Rewrite beginning of first sentence [...] cephalopods (squid, cuttlefish and octopus) [...]
The way it currently reads, you are saying that cephs, cuttle and octop are separate organisms.

Response: According to the comment, the sentence was revised.

Comment 9) :Line 8 ... IN the skin, not near ... color changes, not coloring

Response: According to the comment, the sentence was revised.

Comment 10) :Line 14 purpose in life ... the author lists several (including some of the older) papers

on this topic that talk about the various purposes, including camouflage by countershading, but also various signalling examples. So, this paragraph could be brought to the point by being more specific about the functional aspect of what we do know.

Response: According to the comment, the sentence was revised by adding suitable references.

Comment 11) :The last two sentences (lines 15-17) in this first paragraph are out of context and stick out - they are better moved elsewhere or deleted.

Response: According to the comment, the sentence was deleted.

Comment 12) :Line 22 ascribed instead of imputed, which has a negative meaning

Response: Thank you for the kind advice. This word was revised.

Comment 13) :Sentence Lines 26-27 (cell to cell interactions ...) how do these studies relate to the author's study? Merely saying this here makes little sense in the context of this paper.

Response: As claimed by the reviewer, that topic is out of the scope of this study. Therefore, the sentence was deleted.

Comment 14) :Line 29 The silvery shine on the side ... of which spp?

Response: I was considering the Atlantic herring. This comment was revised and re-written in the sentence referring to countershading.

Comment 15) :Lines 33-36 Sentence that starts with Detailed analyses of the fish ... again, how does this referenced study relate to the author's study?

Response: According to the comment, the sentence was deleted because it is not related.

Comment 16) :Line 37 Most cases of coloring ... first, it needs to be fish body colors, not coloring on a fish body

Then, these types of strong statements require a reference.

It has to be said that this is an area about which we do not know very much. Most people do not have the time or money to do the sorts of studies that are required to answer this question, i.e., watching animals for hours, days or weeks at a time, and for the few studies that are available (all older literature from the earlier part of the 1900s) it turns out that what we have assumed are static patterns actually change. It just takes the fish hours, even several days to do so. Examples of flounders, skates, and others.

Response: I recognized that the comments were already mentioned at the beginning of the Introduction. Dynamic camouflage and fish body color change were referenced there. Therefore, I have removed the sentences from Line 37.

Comment 17) :Paragraph, lines 44-48 ... This paragraph makes no sense in the context of this paper. Which spp is the author referring to? At the end of this paragraph the author jumps back to talking about guanine crystals.

Response: As the reviewer mentioned, the sentence is not strongly related to this study. The paragraph was deleted.

Comment 18): Last two paragraphs of the introduction, lines 49-54 on p2 and lines 3-9 on p3. ... These last two paragraphs can be turned into a couple of sentences that succinctly state the objectives of the paper. The results of the paper should be removed here.

Response: I have reconsidered the motivation of the study mentioned in the Introduction and reached the same conclusion as the Reviewer. The two paragraphs were deleted and replaced with new sentences showing the original purpose of this study, as follows.

“Concerning the guanine platelets of fish, we have observed a light reflection change in the guanine platelets of fish upon exposure to a magnetic field. [21,32] We were able to control the orientation of the guanine platelets and observed distinct light flashes. If the guanine platelets in fish skin exist in a flexible condition, we can expect a rapid reflection of light by them. The aim of this study is to confirm if a rapid light intensity change can be observed that relates to the opto-mechanical behavior of reflecting particles such as guanine platelets in a fish body. It is interesting to explore various kinds of fish from the viewpoint of discovering new optical properties in a fish body. In this study, I investigate the dynamics of light reflection for cells of the dorsal skin of the hardyhead silverside fish, *Atherinomorus lacunosus*.”

Comment 19): Other important aspects: what are iridopores, what are chromatophores, references, etc.

Response: Referencing was carefully checked to provide some basic information about these cells.

Comment 20): Materials and methods, p3

This section is incomplete and needs some detailed attention.

Line 15 Specimens [...] were obtained ... using fishing rods, boat, seine nets, bought in a store? more details please.

How many fish?

What size?

Males, females?

Kept at aquarium, temperature, water supply, animal care details....?

Response: According to the comment, an explanation about the specimens was added as follows.

“2.1 Sample

Specimens of the hardyhead silverside fish, *Atherinomorus lacunosus*, were caught in Okinawa, Japan, by using a net, and then were transported to Hiroshima. The specimens were kept in an open aquarium (Okinawa) and in an indoor aquarium (Hiroshima). The temperature of the seawater in the aquarium was 26–28 °C (open aquarium) and 26–27 °C (indoor aquarium). The height and weight of the used specimens were 98.9 mm +/- 4.9 mm and 9.45 g +/- 1.43 g, respectively. Nine specimens were used for this study. The sex of the used specimens was not determined in this study. Animal experiments in this study were carried out in accordance with the policies of Hiroshima University Animal Care and Use Committee (approval number: F19-2, Hiroshima University). A photograph of the belt-like pattern in the dorsal part of *A. lacunosus* is shown in Figure 1 (a). This belt consisted of a wave-like alignment of circular spots.”

Comment 21): Line 21 The iridophore spots ... Euthanised fish? Living specimen? If living specimen, describe handling procedure, incl revival etc. If euthanized, was the whole fish used or just a dissected part of it?

Response: The observation was carried out in anesthetized fish in an aquarium. Detailed methods are shown in the revised Fig. 1, and the handling procedure is described as follows.

Figure 1(b) shows the observation system for the fish in an aquarium, which consisted of a microscope lens and a light source. The fish were anesthetized by exposure to 0.1% 2-phenoxy-ethanol for 30 s to 1 minute, and then were placed inside of a small box (aquarium) containing sea water with aeration. Experiments on the anesthetized fish were performed for 5 to 10 minutes. Before the anesthesia wore off, the fish were moved back to the large aquarium.

Comment 22): Line 23 Coolpix camera ... did you white balance manually? Was exposure manual? If you had auto exposure on, I have a hard time seeing that you can analyze the RGB pixels for intensity changes between images taken at different times. More info, please. A spectrometer would have been the way to go with this...

What is your sample size? How many images? Still images? From video? It seems that it would be hard to capture still images from these iridescent flashes that you suggested are as fast as <1sec.

How was the pixel analysis done? Did you have an area of interest that was the same for each image? How many pixels? As I mentioned above, unless your camera setting was in manual mode, the RGB content of each of your images would be have been adjusted by the camera and the images would not be useful for RGB analysis.

Response: The image data were obtained from the video recordings with a frame rate of 29.97 fps. The white balance was set manually. The description of the methods was revised and expanded as follows.

“The images of dorsal spots were obtained by using two types of high resolution microscope lens, NAVITAR 2.0x1-51473 (Navitar, Rochester, USA) (Figure 1(b)) and Hirox MXB-2500REZ (Hirox, Tokyo Japan). The former lens (NAVITAR) was connected to a CMOS camera (HOZAN L-835, HOZAN, Osaka, Japan) by a C-mount adapter. The swimming fish were recorded by a video camera (FDR-AX45TIC, SONY, Tokyo, Japan) for the macroscopic presentation shown in Figure 1(a). The focal distance of the lens (NAVITAR) was 32 mm.

For the digital microscopy recordings using capturing software (Xplovview, VIXEN, Tokyo, Japan), the white balance was manually set to be 4500–4650, and this white balance parameter was fixed during the individual video recording.”

Comment 23): Line 28 Sentence: The light illumination ... light was monitored 50-80 degrees ... Lighting is probably the most critical part when working with structural reflectors because changing the angle of the light, or the spread of the focusing beam can give a reflector the appearance of changing colour, even when in nature, there is no colour change. I do not doubt that this species truly is capable of active colour change but a more detailed explanation on methods is necessary here to avoid confusion. A 50 to 80 degree angle is a very big difference that could cause reflectors to appear to turn on an off, just by moving the light.

Response: I have revised the explanation about the incident angle. As shown in the new Fig. 1 (b), the incident angle on the surface of the seawater in the aquarium was roughly set as 50 degrees. The skin

of the fish was inclined ~10 degrees depending on each experiment. The incident light was dispersed when it reached the skin, and the light was scattered on the skin. The conditions for the incident light setting were not precisely controlled. The actual angle is shown in new Fig. 1 (b). The direction of incident light was tilted by 30 degrees in the horizontal plane, as shown in the upper panel of Fig. 1 (b). It is true that, with this setting, the used microscope system provided the video image of the light intensity change in spots with a size of 0.1 mm.

Comment 24): More details about the light source are necessary (Point? Parallel? What kind of beam?), where was it attached, how were angles controlled?

Response: I appreciate these comments. A section explaining the light source was added in the revised manuscript, as follows. The light guide shown in Fig. 1(b) was made of flexible aluminous hose and it was easy to set the appropriate angle.

2.4 Light source

As a light source for the presented data in this paper, a white LED light (LA-HDF158A, Hayashi Repic Co. Ltd., Tokyo, Japan) was used. A light guide connected to this light source was directed towards the back of the fish, as shown in Fig. 1(b). The size of the outlet of the light guide was approximately 8 mm. This provided a scattered light and the light polarization was not additionally modulated. The incident light was provided from the side, as illustrated in Fig. 1(b), and the scattered or reflected light from the sample was monitored.

Comment 25): Line 34 Radiance ... ? This would require very careful measurements with specific photometric equipment. This does not seem to be the case, unless I'm missing something. You looked at overall intensity, it seems to me. Also, the terms radiance, intensity irradiation and some others, are used interchangeably, which has to be fixed throughout the manuscript.

Response: Thank you for pointing out this very important experimental matter. I have a fiber optics measurement system. By using this system, additional experiments were carried out and the data are presented in the revised Figs. 3–5. The term was fixed as 'light intensity' throughout the manuscript.

Comment 26): Sentence lines 37-38 When green color of an image [...]

What does this mean? This method needs more explanation.

Normally, a spectrometer would be used for this type of work but not everyone has access to this kind of equipment and I appreciate that sometimes you have to make do with what you have available. But explain it sufficiently, so that it makes sense and a reader can fully trust your analysis.

Response: The sentence was deleted because the revised manuscript was improved by containing data obtained from fiber optics measurements. The following sentences were added to text.

“2.5 Fiber optic measurements

2.5.1 Measurement of real light from dorsal spots and skin

As shown in Fig. 1(c), the reflected light from the dorsal part of the fish was directly introduced to an optical fiber bundle made of quartz with a total diameter of 10 mm. One end of the optical fiber was connected to a cooled CCD-type spectrophotometer (UCP-2000, Unisoku Co. Ltd. Osaka, Japan). The light was collected in the CCD device with an exposure time of 20 ms, sampling time of 40 ms, and recorded time of up to 300 s. The other end of the optical fiber was positioned at the focused position on the top of the C-mount adapter of the microscope lens (Navitar).

Prior to the fiber optic measurement, adjustment of the region of interest was carried out by using a CMOS camera (Hozan). Replacement of the CMOS camera with the optical fiber was executed within 10 seconds, then the spectrophotometer started the light collection. After the light collection was finished, the opposite replacement was carried out smoothly, and any drifting of the region of interest was monitored. Any data that were conjectured to have been collected where there was a drifting of the region of interest was not used for the analysis.

2.5.2 Measurements on region of interest on a computer LCD screen during playback of recorded video

By using the same fiber optic system except for the lens, measurements on the region of interest on a computer display were carried out. First, the video playback was paused at the time when the region of interest appeared. After one of the ends of the optical fiber was placed close to the region and fixed by using surgical tape (paper tape) on the LCD screen, light collection by a CCD spectrophotometer was started, then video playback commenced. To assist the optical fiber, a stand was also utilized.”

Comment 27): Line 41 A realistic model [...] was formed

This sounds like 3D printing but the following sentences suggest you dissected a piece of fish tissue out and placed the material on this acrylic plate?

If this is the case, SAMPLE would be a better word, not model.

Response: I deleted mentioning the topic of biomimicry from this manuscript. Therefore, the related sentence was removed.

Comment 28): Line 42 delete in the plane. You can shorten this ... acrylic plate (20mm x 20mm, 1mm thick)

Response: I deleted mentioning the topic of biomimicry from this manuscript. Therefore, the related sentence was removed.

Comment 29): Lines 42-45 Turn this sentence around...

A well was drilled into the center of this plate... Guanine platelets were suspended inside this well...

Response: I deleted mentioning the topic of biomimicry from this manuscript. Therefore, the related sentence was removed.

Comment 30):

Line 46 why sea bream? Why not the spp you worked on?

Response: I carried out an additional experiment using reflecting particles that were extracted from the dorsal skin of a euthanized fish. Without carrying out the magnetic field experiments, an observation of the light reflection of the particles floating in water was analyzed, as shown in the revised Fig. 5.

Comment 31): Results and Discussion, pp4-11

Does the living animal show these colour flashes? It is unclear from the materials and methods whether the author worked with living animals or euthanized/dissected specimens.

Do females and males show the same colour flashes?

Response: Yes, the living fish showed flashes under anesthetized condition. I am planning to construct

an observation system to capture the image of swimming fish in the future. As described in the Methods section, the sex of the fish was not determined in this study. However, all of the observed fish showed similar flashes. I am planning to explore any differences in the flashes between females and males.

Comment 32): P4, line 11 where is the dendrite? this is interesting because it would point future research towards physiological studies. It needs to be pointed out in the image.

Response: I suggest that a melanophore-like cell exists beneath the circular spot, as shown in the revised Figure 5. In Fig. 5, the black cell shows the dendrites.

Comment 33): Figure 1 ... the two higher mag yellow boxes look identical. Are both needed? I would delete one.

Response: The old figure 1 was revised, and a revised Fig. 1(a) showing a macroscopic view of the fish's back is presented according to the reviewer's advice.

Comment 34): P5 figure 2A what are the surrounding black structures? What do the two images show? At different times? Different angles?

Response: Same as the response to Comment 32, I propose that the black cell is a melanophore. The old Figure 2 was replaced with newer images. The differences of the two images in the old Fig. 2A is the time.

Comment 35): Figure 2B same as for A

Also, there is not a dramatic difference between the center and the right-hand image. I would suggest getting rid of one of them.

Response: Concerning the presentation of images, I improved Figure 2 by using clearer images and focused on the notable differences. Please refer to the new Figure 2.

Comment 36): Fig2C replace irradiation with incident light.

Response: Throughout the revised manuscript, 'irradiation' was replaced with 'incident light'.

Comment 37): Figure 2 Caption

Light illumination: what is the exact angle? how did you ensure that the incident light direction did not change during imagery, was the preparation secured, etc?

Scale bar, 100 mm.... that must be a typo

Micro particles if you want to start using the materials terminology and start talking about particles, I would suggest you introduce this someplace. Otherwise, stick to reflecting plates, guanine plates, ...

Response: The light angle is shown in the revised Figure 1 (b). As explained in the response to Comment 24, a light guide made of flexible aluminous hose was used, and the light guide did not cause any drift during the measurement for up to hour.

Thank you for the correction of the scale bar.

In the revised manuscript, I discuss the microparticles that may be contained in the circular cell with a diameter of 7–10 μm , as shown in the new Figure 5. As advised by the reviewer, the reflecting microparticles were extracted by scraping the dorsal dissected sample from a euthanized specimen. A

high resolution microscopy (Hirox microscopy) image suggested the existence of reflecting microparticles with a size of 1–3 μm . Please refer to the image in the revised Figure 5 and the attached supporting video.

Comment 38): Fig2C What is this Fig. is based on? Unless you have TEM images, this kind of theoretical drawing has little value here. You also do not sufficiently explain the point of this drawing. The left drawing suggests if plates are angled so that light goes through a series of internal reflective processes, instead of being reflected upwards, an observer above sees nothing. Right drawing shows the assemblage for a visible reflectance when viewed from above. This needs to be stated. But, again, have you verified that these plates behave like "mirrors"?

Response: I have removed this ambiguous illustration from the manuscript. As mentioned by Reviewer, I cannot determine the detailed structure inside the cell without TEM.

Concerning the question, “have you verified that these plates behave like "mirrors"?”, I carried out an additional experiment using the reflecting particles from the dorsal spot area, and made an aqueous suspension of the floating particles. The revised Figure 5 shows that the particles floating in water behaved like tiny mirrors. I also refer to our previous study showing a similar mirror-like behavior of guanine platelets.

Comment 39): P6 First sentence ... light irradiation ... replace with incident light.

Same line ... along the optical axis of the lens ... Please explain this: This would mean illumination came from the inside of the microscope for this to be along the optical axis. It does not sound like this is what you did.

Response: Throughout the revised manuscript, ‘irradiation’ was replaced with ‘incident light’. The confusing sentence “... along the optical axis of the lens ...” was deleted.

Comment 40): Line 10 micro particles... as before

Response: The sentence was revised and deleted according to the reviewer’s comment.

Comment 41): Lines 13-15 Two sentences: Behind the iridophore, a pseudopod..... This suggests It is unclear to me what the author is trying to say.

Response: The sentence was deleted.

Comment 42): Lines 25-27 A tunable.... Remove the two sentences here and focus on your results

Response: The sentence was revised and this work’s original result was enhanced.

Comment 43): Paragraph lines 27-37 is this in a euthanized fish? Anesthetized fish? This is important to know. Are these flashes artifacts or a visual signal that these fish put out? Do they do this while actively swimming?

Response: The observed specimens were anesthetized. The sample for the reflecting particle analysis was a dorsal dissected sample from a euthanized specimen. There was an artifact of fish skin motion, as explained by Figures 3 and 4. I made a distinction between them by analyzing the frequency of the light flashes.

As I responded to **Comment 31**, the present study only presents the data of living fish showing flashes

under anesthetized condition. I am planning to construct an observation system to capture the image of swimming fish in the future. It may possible to video them, but at present, I have not yet succeeded in taking a clear video.

Comment 44):Two sentences in lines 38-45 are awkward, perhaps rephrasing them would help clarify what the author is trying to say.

Response: The sentence was deleted and the Results and discussion section was completely revised.

Comment 45):P7 Panel Fig 3A ... What is the difference between these 11 images? Different time scale? They all look the same to me. How were the data in the plot underneath derived? Average of some number of pixels? This chart does not visually jive with the images above. Get rid of 0 and 12, since there are no corresponding sequences above.

Response: The data shown in the old Fig. 3 was replaced with clearer data as shown in the revised Fig. 2. The sentences relating to the analysis of this old Fig. 3 were deleted.

Comment 46):Fig. 3B Three sequences are shown in the plot, but only two images are shown.

Again, to me, there is not an appreciable difference between the iridescent spots here, looking at the image at the top and the one at the bottom. Spot iv looks a little different perhaps, and there is a tiny change in v_i , which does not show up in your corresponding plot. The other three spots do not look very different at all.

Response: Same as the previous response to Comment 45, the sentences relating to the analysis of the old Fig. 3 were deleted.

Comment 47):Line 43 brilliance ... The author uses different terminology throughout.

Intensity relative to grey may be better?

Response: Thank you very much for the reviewer's kind advice. I used the recommended term 'Intensity relative to gray' in the revised Figure 3(a)(4).

Comment 48):P8 Line12 Should not be strongly related to blood flow [...] Innervation by the neural network [...]

This conclusion is based on what? More explanation is needed here.

Response: I changed my opinion in the revised manuscript. The sentence was removed in the revised manuscript because clear data showing the effect of blood flow on the light flash was not obtained from the additional experiments conducted in this revision process. A sentence regarding the conjecture about the blood flow effect was inserted in the text as follows.

"In addition to the skin motion, other influencing factors, such as blood flow, may be conjectured. In the present study, most of observed dorsal spots existed apart from blood vessels, and no correlation between the flash and blood flow was detected."

Comment 49):Line 15 Brilliance again. See above

Response: Thank you again. I have corrected this terminology in the revision.

Comment 50):Line 19-20 [...] should exist inside the parietal skin

Unless there is evidence, this may need to be stated more as a question than a statement.

Response: In the revised manuscript, this kind of sentence has been stated as a conjecture or hypothesis.

Comment 51):Figure 4

Delete separately in line 45

Line 48, summation in RGB colors ... this needs to be explained. Meta analysis ... this needs explanation as well.

Response: The old Figure 4 was deleted and replaced with the revised Figure 3, which was obtained by employing fiber optics measurements. The analysis shown in the new Figure 3(a)(4) used the 'Intensity of gray' measurement from the gray scale images which were converted from the color images. In the case of this image analysis, the video images (29.77 fps) were separated (or sampled) at 25 frames per second. I carried out the frame-by-frame manual analysis for 51 frames (about 2 sec video).

Comment 52):Figure 5

5C needs more explanation to be useful. It is unclear whether one scenario explains lack of reflectance and the other scenario explains reflectance under magnetic field stimulation?

Response: The magnetic field experiment became unnecessary for the story of the revised manuscript. Therefore, the old Figure 5 was deleted. I appreciate the reviewer for checking this figure.

Comment 53):The magnetic field experiment needs more explanation, including more methods. Why was this done in the first place? What was the purpose? Why not use the spp that you studied? Why a bream?

Response: As explained before, the magnetic field experiment became unnecessary for the purpose of the revised manuscript. The reason why I used bream was because of the volume of guanine platelets available from a specimen. I am planning in the next stage of experiments to collect a sufficient number of reflecting particles from the dorsal skin of hardyhead silverside fish for the purpose of comparing the light reflecting properties with those from the guanine platelets of other fish.

Comment 54):P11 Line 20 Reference for lanternfish paper, please.

Response: The sentence was removed in the revision. A paper on lanternfish was reported by my collaborator in guanine platelet research.

“Paitio J, Yano D, Muneyama E, Takei S, Hironori Asada H, Iwasaka M, Oba Y, 2020 Reflector of the body photophore in lanternfish is mechanistically tuned to project the biochemical emission in photocytes for counterillumination, *Journal Biochemical and Biophysical Research Communications* 521, 821-826”

Comment 55):P11

This entire page, in my opinion, needs to be re-structured, made more succinct and clear. The author talks about applying a magnetic field to bream (why bream, what about *A. lacunosus*?) guanine crystals and observing reflective changes. This moves away from the actual biology of this animal, which is fine. It moves into a potential realm of application, although it remains a mystery to me how applying

changes in magnetic field direction (how about strength?) would be a realistic application potential. The author provides limited insight into what the magnetic field experiments add to this study. The flow of this entire page needs to be improved. Sentences and paragraphs jump back and forth with little connection.

Response: As explained before, the magnetic field experiment became unnecessary for the purpose of the revised manuscript.

The entire Results and discussion section was re-structured according to the Reviewer's comments.

Comment 56):Supporting information. P15-18

Line 36 Radiance... terminology... with your method, you cannot talk about radiance here.

Response: I changed the term 'radiance' to 'light intensity' in the revised manuscript. In addition, all of the old Supporting figures were removed in the revision. The data are now shown in the main text. Seven supporting videos and their captions are now presented in the Supporting information.

Comment 57):Line 39 radiance [...] disappeared ... Again, terminology re radiance. Also, reflectance did not disappear, it would have been reflected back downwards. But yes, it is not transmitted, which is the point you are trying to make.

Response: Thank you for the comment. I revised the terminology in text. Figure S2 was removed.

Comment 58):Line 43 Particles, crystals, terminology is used interchangeably throughout the manuscript. Delete BIOGENIC

Response: I revised the terminology and deleted 'biogenic'.

Comment 59): Figure S1

How were these angles controlled? What exactly were these angles? Estimating them from the drawing is not possible.

What was the light source? Point light or scattered? Please make sure this information is in the materials section.

This figure needs to be explained a lot more to be useful.

The images all more or less look the same to me, there is not a huge difference, perhaps a slight decrease in reflectance between a,b,c compared to d,e but other than that, it is unclear why a,b and c are here, what the arrows show, and why d and e are here.

Response: This figure showed a rough measurement. This figure became unnecessary in the revision. Therefore, Figure S1 was deleted.

Comment 60):Figure S2

This idea of blood flow needs more careful explanation. The idea itself makes some sense to me but the supporting evidence shown here does not convince me. The images shown all look the same to me. Perhaps I'm missing something but the author is also not pointing me in a particular direction. More detailed explanation is necessary.

Response: Figure S2 was deleted because clear data showing the effect of blood flow on the light flash was not obtained from the additional experiments conducted during this revision process. A sentence referring to the conjecture about the blood flow effect was inserted in the main text as follows.

“In addition to the skin motion, other influencing factors, such as blood flow, may be conjectured. In the present study, most of observed dorsal spots existed apart from blood vessels, and no correlation between the flash and blood flow was detected.”

Responses to Reviewer 3

General comment:

The author describes a flickering reflective pattern in a spot in the skin of a fish. The authors describe the flickering and suggest that it is actively controlled by the fish, who continuously rearranges guanine crystals in the spot to change the reflective patterns. They experimentally support this conjecture by showing that guanine crystals in solution can be arranged by magnetic fields. This is an interesting observation, and should set the stage for more mechanistic investigations into the phenomenon. The science is sound, but I had to read it a few times before I fully understood the story. The authors should clarify it a bit more by, for example, using the common name of the fish as well as the scientific name and outlining the motivation of the study more clearly.

Response: Thank you very much for reading the manuscript and spending time on this review. I revised the motivation of this study by focusing on the dramatic phenomenon of flickering (or flashing) in dorsal spots of the fish whose common name is hardyhead silverside fish. I appreciate the reviewer’s interest in the observation under magnetic fields. However, I thought the data with magnetic fields should be provided by using the same species of fish, and I felt it requires further work to establish suitable experimental methods. Therefore, I replaced the topic of the magnetic field experiments with the fiber optics measurements for the detailed properties of the light flash in spots.

In the revised manuscript, a statement about the motivation was explained in the Introduction, the experimental methods were expanded, and the results were explained with statements based on the obtained evidence. I hope this revision clearly responds to your comments.

Comment 1):

Otherwise I just have a few comments and suggestions.

P2L18: why are they considered important and interesting?

Response: I appreciate your question. In the revised manuscript, the sentence was beyond the scope of the introduction. Therefore, the sentence was deleted.

Comment 2):

P2L49: “light switching panel” is not clear. Do you mean the lights are turning off and on, or flashing, or what?

Response: I intended to say ‘flashing’. The sentence referring to the “light switching panel” was deleted owing to the major revision of the paragraphs in the Introduction.

Comment 3):

P2L51: How do you know it’s active?

Response: The dorsal skin of the anesthetized fish was observed in real time and the real time image was recorded in a video (movie). In this observation, it was found that the reflecting spots actively caused flashes.

Comment 4):

P4L14: green and blue colors are typically also structural colors. Do you know how these colors are made?

Response: There remains conjectures about the origin of these colors. If the blue color is a structural color, perhaps reflecting platelets made of guanine are candidates for the material providing multiple reflections. The green color may result from multiple stacking of the guanine platelets, but I propose we need more evidence to make a statement on the actual mechanism.

Comment 5):

P8L13: Is this just normal movement of the scales? Do other fish do this?

Response: In the revision, I checked the vibrational frequency of the skin, and compared that with the flash frequency in spots. The skin including the scales of a living fish exhibited a movement at a frequency of 2 Hz or higher. In contrast, typical frequencies of the flash in the spot were 0.5–1 Hz. The results indicated that spots exhibited a flash independent of the skin/scale movement. In the supporting movie for the new Figure 2 of the revised manuscript (**S_movieFIG2B**), a movement of a scale at frequency of approximately 2 Hz can be seen on the right-hand side, but we can recognize that the spots are flashing individually at other frequencies.

I am hoping to explore other fish that exhibit a similar flashing, but I have not yet discovered any at present.

Comment 6):

P10L53: Neat experiment!

Response: I appreciate the comment. In the revised manuscript, the data of the magnetic field experiments were removed because the topic was beyond the scope of the revised manuscript.

I appreciate all of the important comments from the Reviewers. Thank you so much for spending your time to review the manuscript.

Appendix C

Flashing spots on the dorsal part of hardyhead silverside fish

~~Rapid active-reflection display on dorsal stripes of *Atherinomorus lacunosus*~~

Masakazu Iwasaka

Hiroshima University, Kagamiyama 1-4-2, Higashihiroshima, Hiroshima 739-8527, Japan

Correspondence to: Masakazu Iwasaka, miwamasa@hiroshima-u.ac.jp

A large number of living creatures are able to use environmental light effectively as a biological display. The biological structural colors are very attractive not only to members of the same colorful species within the coloring species but also to humans. However, the detailed function of bio-reflectors, which constitute the structural color with respect to communication, remains unknown. *Atherinomorus lacunosus*, a teleost fish has alignments of circular iridophore spots on its dorsal part. Here it is found that a spot with a diameter of approximately 0.1 mm causes a rhythmic flashing blinking of light under exposure to an incident light. —a side light exposure. owing to rapid reflection changes in platelets existing inside the spot. The speed of the intensity change of light at a frequency of approximately 1 Hz is proposed to be controlled by the nervous system of *A. lacunosus*. This kind of passive illumination may contribute to the development of a new optical device with low energy consumption. The typical light flash has a pulse width of approximately one second. When a flash train appears, the flash repeats at a typical frequency of 0.5–1 Hz. The spot consists of circular cells whose diameters are approximately 7–10 μm . The circular cell contains reflecting particles with diameters of 1–3 μm , which show light flashing.

1. Introduction

Studies on cephalopods, (squids, cuttlefish [1-3] and octopus) [1-4] have revealed that a dynamic camouflage effect occurs in the cellular tissue existing near in the skin. A similar mechanism for coloring changes has also been discovered in the tissue structures of fish, [5-10] insects [11-16] and a mammal [17]. In biological materials that are able to control light, guanine is useful for light reflection and optical interference in iridescent tissues. [5-7] Guanine-based photonic crystal structures have been found in many species of fish, [5,6,10,18-21] scallop [22] and animal plankton. [23,24]

It is often possible to observe many markings on the surface of a fish body. There is long history of research on fish body colors concerning their purpose in the life, such as countershading [25, 26] and silvering camouflage of fish [20]. The role of patterns on a fish body surface has been suggested to be an active communication tool, and nonlinear mechanism studies have ascribed the stripe pattern in a fish to a Turing pattern, which is generated by an interaction between chromatophores. [27-30] Chromatophores are groups of cells containing particles that control the color based on the filtering or reflection of light. [31] Among the chromatophores, iridophores are known to possess reflecting particles such as guanine platelets. [27,31]

Concerning the guanine platelets of fish, we have observed a light reflection change in the guanine platelets of fish upon exposure to a magnetic field. [21,32] We were able to control the orientation of the guanine platelets and observed distinct light flashes. If the guanine platelets in fish skin exist in a flexible condition, we can expect a rapid reflection of light by them. The aim of this study is to confirm if a rapid light intensity change can be observed that relates to the opto-mechanical behavior of reflecting particles such as guanine platelets in a fish body. It is interesting to explore various kinds of fish from the viewpoint of discovering new optical properties in a fish body. In this study, I investigate the dynamics of light reflection for cells of the dorsal skin of the hardyhead silverside fish, *Atherinomorus lacunosus*.

although their purpose in the life of the fish remains to be clarified. For example, the side part and dorsal part of a fish are frequently colored in silver and black, respectively. The color is generated by several kinds of chromatophores. [25]

It has also been indicated that the color patterning by several kinds of chromatophores containing a pigment is concerned with both hormonal and neural activity. [26] Additionally, the role of patterns on a fish body surface has been suggested to be an active communication tool, while nonlinear mechanism studies have imputed the stripe pattern in a fish to a Turing pattern, which was generated by an interaction between the chromatophores. [27,28,29,30]

Developmental pathways, possibly controlled by the interactions between the same and different kinds of chromatophores, are considered to be an interesting and important topic. The effects of cell to cell interactions on the color pattern formation in zebrafish have been investigated in recent studies. [31-33]

The silvery shine of the side and ventral parts is mainly controlled by iridophores containing guanine crystals. Over the dorsal region that is mainly covered by black melanophores, some species, such as Japanese anchovy, have an additional layer that looks partially blue to purple. We conjecture that the additional color modification is carried out by light interference in guanine crystal layers. Detailed analyses of the fish coloring pattern have been performed on the skin of zebrafish, in which a variety of patterns, stripes and spots have been inspected from the viewpoints of developmental and ethological features. [34]

Most cases of coloring on a fish body are static, that is, non-dynamic coloring. Black coloring of the dorsal part of a fish is explained by a counter shading effect, which helps the fish become less visible to a predator approaching from above. In contrast, the ventral part

covered with shiny iridophores can result in a matching of the light intensity between the fish body and water for a predator looking from below. As the guanine crystals that result in light reflection are fixed behind the scales, the skin tissue itself performs a passive reflection. Control of the light reflection can be carried out by the swimming motion of the fish.—

A species of fish has a light emitting tissue—a photophore—in the ventral or side part of the body, and the photophores that are distributed over the surface of the ventral to side parts exhibit an active matching of the light intensity, known as counter illumination. The role of reflector particles, such as guanine crystals, still requires further clarification.—

The present study focuses on how the reflecting spots behave like a light switching panel on the dorsal skin of *Atherinomorus lacunosus*. *A. lacunosus* is one of the genera belonging to silversides, order *Atheriniformes*. For the first time, an active and rhythmic light reflection on fish skin is discovered. The reflecting particles existing inside the spot exhibit a rapid change of brilliance at approximately 1 Hz for one-directional light irradiation.—

Previous studies have reported that some species of squid show a dynamic structural color change. It has been suggested that the color change is controlled by the neural system of the squid. However, the time to obtain a sufficient color change is more than several seconds. Studies on neon tetra have also provided data about its dynamic color control in chromatophores on a similar time scale. In the case of a chameleon, the color change speed is of the same order. Here, we present the quickest change of color or light intensity in a species of silversides.—

2. Materials and Methods

2.1 Sample

Specimens of the hardyhead silverside fish, *Atherinomorus lacunosus*, were caught in Okinawa, Japan, by using a net, and then were transported to Hiroshima. The specimens were kept in an open aquarium (Okinawa) and in an indoor aquarium (Hiroshima). The temperature of the seawater in the aquarium was 26–28 °C (open aquarium) and 26–27 °C (indoor aquarium). The height and weight of the used specimens were 98.9 mm +/- 4.9 mm and 9.45 g +/- 1.43 g, respectively. Nine specimens were used for this study. The sex of the used specimens was not determined in this study. Animal experiments in this study were carried out in accordance with the policies of Hiroshima University Animal Care and Use Committee (approval number: F19-2, Hiroshima University). A photograph of the belt-like pattern in the dorsal part of *A. lacunosus* is shown in Figure 1 (a). This belt consisted of a wave-like alignment of circular spots.

2.2 Observation of anesthetized fish in and aquarium

Figure 1(b) shows the observation system for the fish in an aquarium, which consisted of a microscope lens and a light source. The fish were anesthetized by exposure to 0.1% 2-phenoxy-ethanol for 30 s to 1 minute, and then were placed inside of a small box (aquarium) containing sea water with aeration. Experiments on the anesthetized fish were performed for 5 to 10 minutes. Before the anesthesia wore off, the fish were moved back to the large aquarium.

2.3 Video image recording

The images of dorsal spots were obtained by using two types of high resolution microscope lens, NAVITAR 2.0x1-51473 (Navitar, Rochester, USA) (Figure 1(b)) and Hirox MXB-2500REZ (Hirox, Tokyo Japan). The former lens (NAVITAR) was connected to a CMOS camera (HOZAN L-835, HOZAN, Osaka, Japan) by a C-mount adapter. The swimming fish were recorded by a video camera (FDR-AX45TIC, SONY, Tokyo, Japan) for the macroscopic presentation shown in Figure 1(a). The focal distance of the lens (NAVITAR) was 32 mm.

For the digital microscopy recordings using capturing software (Xplovview, VIXEN, Tokyo, Japan), the white balance was manually set to be 4500–4650, and this white balance parameter was fixed during the individual video recording.

2.4 Light source

As a light source for the presented data in this paper, a white LED light (LA-HDF158A, Hayashi Repic Co. Ltd., Tokyo, Japan) was used. A light guide connected to this light source was directed towards the back of the fish, as shown in Fig. 1(b). The size of the outlet of the light guide was approximately 8 mm. This provided a scattered light and the light polarization was not additionally modulated. The incident light was provided from the side, as illustrated in Fig. 1(b), and the scattered or reflected light from the sample was monitored.

2.5 Fiber optic measurements

2.5.1 Measurement of real light from dorsal spots and skin

As shown in Fig. 1(c), the reflected light from the dorsal part of the fish was directly introduced to an optical fiber bundle made of quartz with a total diameter of 10 mm. One end of the optical fiber was connected to a cooled CCD-type spectrophotometer (UCP-2000, Unisoku Co. Ltd. Osaka, Japan). The light was collected in the CCD device with an exposure time of 20 ms, sampling time of 40 ms, and recorded time of up to 300 s. The other end of the optical fiber was positioned at the focused position on the top of the C-mount adapter of the microscope lens (Navitar).

Prior to the fiber optic measurement, adjustment of the region of interest was carried out by using a CMOS camera (Hozan). Replacement of the CMOS camera with the optical fiber was executed within 10 seconds, then the spectrophotometer started the light collection. After the light collection was finished, the opposite replacement was carried out smoothly, and any drifting of the region of interest was monitored. Any data that were conjectured to have been collected where there was a drifting of the region of interest was not used for the analysis.

2.5.2 Measurements on region of interest on a computer LCD screen during playback of recorded video

By using the same fiber optic system except for the lens, measurements on the region of interest on a computer display were carried out. First, the video playback was paused at the time when the region of interest appeared. After one of the ends of the optical fiber was placed close to the region and fixed by using surgical tape (paper tape) on the LCD screen, light collection by a CCD spectrophotometer was started, then video playback commenced. To assist the optical fiber, a stand was also utilized.

2.6 Additional image analysis

To check one of the fiber optic measurements (Fig. 3(a)(4)), additional image analyses were performed using Image J (NIH) software. After the color image was transformed to a gray scale image, the intensity relative to gray in the image was measured.

2.1 Sample and image analysis

~~Specimens of *Atherinomorus lacunosus* were obtained in Okinawa. Animal experiments in this study were carried out in accordance with the policies of Hiroshima University Animal Care and Use Committee (approval number: F19-2, Hiroshima University).~~

2.2 Image analysis

~~The iridophore spots on the body surface were analyzed by using digital image processing units. Low resolution images of the fish body surface were taken by using a digital camera~~

(Coolpix AW130, NIKON co. Tokyo, Japan) and a digital microscope (PC-230, Vixen, Tokyo, Japan). For the high resolution images with an amplification of $\times 230$, the same digital microscope was used. The images of fish and its body surface were obtained under exposure to white LED light (MLEP-B070W1LRD, MCEP-CW8-2M, Moritex, Tokyo, Japan) whose light polarization was not additionally modulated. The light illumination was provided from the side and the scattered or reflected light from the sample was monitored. The incident angle of the irradiation was 50–80 degrees. Additionally, by changing the direction of the irradiation on the plane of the observed area, images with the highest and lowest contrast were captured. The observation of the sample was carried out under a wet condition, where the surface of the sample was soaked in water containing anesthetic solution (2-phenoxy-ethanol).

Image analyses were performed using Image J (NIH) software. Radiance of the sampled image was measured and quantified by the gray scale value, which was produced from the intensities of red (R), green (G) and blue (B) pixels. When green color of an image was suitable for evaluation, the gray scale value of B was employed.

2.3 Mimicry of iridophore spots

A realistic model of the iridophore spot was formed on a transparent acrylic plate whose size was 20 mm \times 20 mm in the plane and 1 mm in thickness. An aqueous suspension containing guanine platelets was facilitated in a cylindrical well, which was fabricated in the plate. The well, approximately 1.2 mm in diameter, was bored by an end mill of a router. Guanine platelets of sea bream, *Sparidae* were suspended in water and were poured into the well, and the microplatelets were floating in the water during the observation with and without magnetic field exposure.

The fabricated acrylic plate was placed on the gap space between magnetic poles of a resistive electromagnet (WS15-40-5K-MS, Hayama Inc., Koriyama, Japan). The gap length in the magnetic field exposure space with a maximum field of 500 mT was 70 mm. Applying the magnetic fields, brilliance changes in the formed spot were recorded by a digital microscope system (RH-2000 with MXB-2500REZ (lens), Hirox co., Tokyo, Japan) under one-directional light irradiation from the side of the plate. LED light (LA-HDF158A, Hayashi-Repic, co. ltd, Tokyo, Japan) was used as the light source.

Figure 1. Experimental set-up for video recordings and fiber optic measurements on the dorsal part of hardyhead silverside fish. (a) Image of the dorsal part with an array of circular spots. Scale bar is 1 mm. (b) Experimental diagram of the observation of fish in an aquarium with a microscope lens and light source. An anesthetized fish was placed on a bed made of soft clay in a small aquarium containing sea water with aeration. (c) Real light measurement, a method

for collecting light from the observed area through the microscope lens. Replacement of the CMOS camera with an optical fiber ($\phi 10$ mm) and the opposite replacement were carried out smoothly (within 10 seconds). (d) Collection of light from the region of interest on a LCD screen to which an end of optical fiber was affixed.

3. Results and Discussion

A photograph of the belt-like pattern in the dorsal part of *A. lacunosus* is shown in Figure 1. This belt consisted of a wave-like alignment of chromatophores. Magnifications of the image (Fig. 1 (b)) show circular spots that were aligned along the edge of a scale. The line of circular spots, which was composed of connecting spots, extended from the ventral to dorsal sides. This iridophore-like spot overlapped with a black melanophore which had a dendrite. In Fig. 1(a) and (b), silver and green long bands were observed on the side of the body of the fish. In addition to the structural color, green, blue and yellow colors were observed in the dorsal belt and also in the parietal skin.

Figure 1. Reflecting spot arrays on the dorsal part of *Atherinomorus lacunosus*. (a) Top view of the dorsal part showing belts of spots aligned along the edges of a scale. Bar is 10 mm. (b) Magnified images of the belts. Bar is 0.5 mm.

Figure 2 shows two examples of the flashing spots on the dorsal part of hardyhead silverside fish. The major color of the circular spot in Fig. 2(a) at 7.50 s and 8.73 s was green, but became yellow at 9.30 s. A similar phenomenon is shown in Fig. 2(b), where many circular spots proceeded with the color change process which was reversible. The spots seemed to cause a light reflection flash.

Figure 2. The observed phenomenon of the flash of light reflection in spots of the dorsal part of hardyhead silverside fish. (a) and (b) Two examples of the phenomenon. Time indicators denote passing time in the recorded video for each of the static images. The static images in (a) and (b) were captured from the recorded videos S_movieFIG2A and 2B, respectively. Scale bar is 100 μ m.

Figure 2. Reflection enhancement in an iridescent spot under side light illumination. (a) Change of brilliance in the fringe of the circular spot. Scale bar is 100 μm . (b) Expanded images of the spot containing micro-particles. Upper small cuts show four particles that resemble platelets, probably causing an inclined surface. Scale bar is 5 μm . (c) Illustrations of the iridophore. The reflector platelet, assumed to be made of guanine, is controlling the direction of light reflection.

Exposing this iridophore-like spot to light irradiation along the optical axis of the lens showed no remarkable reflection change. However, irradiating light from a side direction exhibited a dynamic brilliance change, as shown in Figure 2. In the image shown in Fig. 2(a), the fringe of the circular spot suddenly became brilliant when side illumination was supplied from the right side. Inside the spot, the green and yellow particles became distinct.

A magnification of the spot is shown in Figure 2(b), where it was obvious that active light reflection of this fish was achieved by micro-particles. The size of the circular reflecting particles was 5–10 micrometers in diameter, which is a similar size as the guanine crystal platelets from a fish, for example, sea bream. Behind the iridophore, a pseudopod was observed within the penetration of the used lens. This suggested that the circular shape of the reflecting particle was the broad surface of a platelet rather than the cross-section of a fiber elongated to the vision.

The brilliance became dark when the light irradiation was terminated. This proved that the brilliance of the spot arose from light reflection and not light emission such as bioluminescence. Additionally, the changes in brilliance and colors when the angle of incident light was varied exactly resembled those of accumulated guanine platelets from other fishes (Supporting Information Fig. S1, Movie S1). A proposed model for the inside structure of the spot is shown in Fig. 4(c). A platelet or a stack of platelets were aligned in a spot. When a trigger for changing the incline of the platelets by any mechanical force occurred, the reflector platelets of the peripheral part oriented themselves to enhance the brilliance.

A tunable color change in biological tissue has been reported previously in cuttlefish and neon tetra. These studies focused on the change in thickness and angle of the layers, which control the optical interference that dominates its structural color. The newly discovered feature in spots of *A. lacunosus* suggests the possibility of a rapid response in reflection only. This study found for the first time that the spots caused a rapid blinking of the reflected light. Figure 3(a) shows an example of the reflection blinking in connected spots formed a line. The analysis on a time course of the green light reflection intensity revealed the occurrence of synchronized light blinking by three spots in the line. This blinking occasionally repeated with a rhythm like respiration. Another type of arrangement of spots, scattered spots, is shown in Figure 3(b).

There was no distinct synchronization of individual blinking. In the scattered spots area, the light reflection changes of the spots occurred separately. —

The origin of blinking is ascertained in the following discussion on a careful test of the correlation between influencing factors. First, less of an effect of the skin motion on the reflection changes in the spot exists. Although a discontinuity of the refractive index of the scale and water can change the incident angle of light when the scales slide owing to skin motion, no correlation was found between the spot reflection and the sliding. It was possible to conjecture that a stream of blood with a discontinuity of the refractive index in the flow caused additional light reflection dynamics (Supporting Fig. S2, Movie S2). —

Figure 3. Blinking iridescent spots with synchronization (a) and desynchronization (b). (a) Synchronized blinking in three spots (I–III) in a group of connected spots forming a line. Length of time between two sequences is approximately 0.5–1.0 s. Brilliance (gray scale value, g.s.v.) of green color only was calculated. (b) Desynchronized blinking in three spots (IV–VII) in a group of scattered spots. Scale bars are 100 μm . —

Figure 3. Analysis of flash frequency of the dorsal spot by using two types of fiber optic measurements, light collection through a microscope lens and from a LCD screen.

(a) First specimen. (1) Time course of real light at five wavelengths through a microscope lens after light reflection from the body surface of the dorsal part. (2) Magnification of the time course at 528 nm shown in (1) from 150 s to 180 s. (3) Static image in the beginning of

S_movieFIG3A, 4B. Scale bar is 100 μm . The spot existed in the area measured for the data (1)–(2), which was determined separately from the video recording. (5)–(7) Analyses on light collection from the LCD screen. The value of the intensity was a summation of the light intensity at five wavelengths (450 nm, 500 nm, 528 nm, 574 nm, 600 nm). (4) Time course of intensity of gray which was measured from the spot shown in (3). Image analysis using Image-J was carried out after the color image was transformed to a gray scale image. (5–7) Data obtained with a cooled CCD spectrometer. (5–6) Time course of light intensity in the whole area of the video image. (7) Time course of light collection in the blue circle shown in (3), where an edge of a neighboring spot appears.

(b) Second specimen. (1) Time course of real light at three wavelengths through a microscope lens after light reflection from the body surface of the dorsal part. (2) Static image taken at the beginning of S_movieFIG3B. Scale bar is 100 μm . The spot existed in the area measured for the data (1)–(2), which was determined separately from the video recording. (3) Magnification of the time course of (1) from 100 s to 120 s.

Next, the continuous flashing of the circular spots was investigated, as shown in Figure 3. These data were obtained from two specimens that were different from the specimens used in Fig. 2. The flash frequency of a dorsal spot was analyzed by using two types of fiber optic measurements. In the first specimen (Fig. 3(a)), the flashing of the real light was monitored for 300 s (Fig. 3(a)(1)). A distinct flash at three wavelengths (528 nm, 574 nm, and 600 nm) was obtained because the light source intensity at these wavelengths was higher than at others (450 nm and 500 nm). Magnification of the time course for the light at 528 nm (Fig. 3(a)(2)) suggested that the roughly estimated frequency of this spot was 0.5–1 Hz.

In the first two seconds of S_movieFIG3A, 26 sheets of static color images were captured (25 frame per second) and transformed to gray scale images, then the intensity of gray was plotted versus time, as shown in Fig. 3(a)(4). The intensity time course exhibited approximately four peaks during the first two seconds. The inset images correspond to the intensity changes.

By using fiber optic light collection from a LCD screen, additional analysis could be carried out (Fig. 3(a) (5-7)). The red circle marked on Fig. 3(a)(3) was the position measured by the optical fiber. Fig. 3(a)(5) shows the change in light intensity from the beginning to 30s in the playback of S_movieFIG3A. Fig. 3(a)(6) shows a magnification of the first two seconds. The intensity was the summation of the light intensity at five wavelengths (450 nm, 500 nm, 528 nm, 574 nm, and 600 nm). The roughly estimated frequency of this spot was 0.3–1 Hz, which was similar to the results of the real light shown in Fig. 3(a) (1-2).

As a possible artifact, the effect of skin movement should be tested. In the blue circle marked in Fig. 3(a)(3), an edge of a neighboring spot appeared that showed a vibrational movement during video playback. The change of light intensity arising from this movement was analyzed on the LCD screen by a fiber optic measurement (Fig. 3(a)(7)). The result indicated the existence of a change in light intensity with a frequency of around 2 Hz that resulted from skin motion, and this corresponded to the results of the image-only analysis (Fig. 3(a)(4)).

Another specimen (Fig. 3(b)) indicated that the dorsal spots caused a flash of light reflection at frequencies of 0.5–0.7 Hz. The data shown here were obtained only by real light collection through the microscope lens and optical fiber that was connected to the spectrophotometer. In addition, this example showed a slower frequency at 0.02 Hz. Fig. 3(b)(1) presents the long-term cycle of the light flash that occurred every fifty seconds, which was repeated at least three times.

Figure 4. Analysis of flash frequency of the dorsal spot showing a fast flashing. (a), (b) Two examples of fiber optic measurements from a position on a computer LCD screen displaying the recorded video (S_movieFIG4A, 4B). Red and blue circles indicate the position on a dorsal spot and a black cell to which an end of an optical fiber ($\phi 10$ mm) was positioned, respectively. Start time means the time in the recorded video when the spectrometer started its measurement on the LCD screen. Scale bar is 100 μ m.

A more detailed analysis at higher frequencies for the flash of dorsal spots is needed. The analyses were carried out only using light collection from the LCD screen, as shown in Figure 4. The data were obtained from an additional two specimens. Both specimens showed flashes that exhibited changes in their intensity at frequencies of 3 Hz to 4 Hz. However, the light intensity changes in a black cell also showed a similar frequency, 3 Hz ~ 4 Hz. The analyzed videos (S_movieFIG4A and 4B) exhibited a time with rapid flashing. The fish skin vibration at a frequency of 3–4 Hz may be involved with the flash of the dorsal spots. In addition to the skin motion, other influencing factors, such as blood flow, may be conjectured. In the present study, most of observed dorsal spots existed apart from blood vessels, and no correlation between the flash and blood flow was detected.

The blinking spots were also found in the skin of the head where no edges of the scales existed. The circular spots in the parietal skin exhibited the most active reflection blinking

(Supporting Movie S3). Figure 4 shows spots on the skin of the parietal region covering the brain. These spots appeared to be disconnected on the surface of the skin. Time course analyses of the brilliance in three colors (red, green and blue) were carried out on four areas involving one or several spots. Most of the spots exhibited a light reflection reaching the maximum intensity in 0.3–0.6 s, which was returned to a lower intensity and then continued to oscillate.

The oscillation of light reflection should not be strongly related to blood flow dynamics beneath the skin. Innervation by the neural network is proposed to be a factor instead. Figure 4(b) shows time courses of the brilliance change of all spots existing in an area including the spots that are shown in Figure 4(a). The spots exhibiting a remarkable light reflection had a synchronized phase. Two spots separated by a distance of up to 0.5 mm exhibited synchronization of the periodic change at 1 Hz. Nerve fibers controlling the light reflection of these individual spots should exist inside the parietal skin.—

Figure 4. Blinking spots separately located on the skin of the parietal region above the brain of *A. lacunosus*. (a) Image sequence of four groups (I–IV) of blinking spots (spot). Scale bar is 200 μm. (b) Time course of change in brilliance of the spots in the images shown in (a). (gray scale value, summation in RGB colors). (c) Image sequence of a part of the area shown in (a). (d) Meta-analysis on blinking and non-blinking spots in the area shown in (c). Scale bar is 200 μm.—

Figure 5. Experimental simulation of the blinking spot by using an aqueous solution containing guanine platelets of sea bream, *Sparidae* localizing in a hole formed in an acrylic plate. (a) Floating guanine platelets in a well containing water. Side light illumination is provided from the bottom. (b) Under exposure of a magnetic field at 500 mT. A, B; Scale bar is 1 mm. (c) Mechanism of the magnetically induced incline change in guanine platelet and the light reflection modulation.—

It was apparent that the reflector particles in the spot had light reflection anisotropy resembling that of guanine crystal platelets existing in fish skin and choroidal tapetum. The typical size of a fish guanine platelet was a few micrometers to several tens of micrometers in length and around 100 nm in thickness. Therefore, it is possible to mimic the reflecting spot of *A. lacunosus* by using guanine platelets extracted from a skin of fish, such as goldfish.—

Figure 5 shows a simulation of the spot formed in a hole in an acrylic plate. The aqueous solution containing guanine platelets of sea bream was localized in the hole. A demonstration of the light reflection change in the mimicked spot was carried out by applying a magnetic field at 500 mT. The mechanism of the magnetic orientation of guanine platelets owing to its diamagnetic anisotropy provided the inclination of the guanine-based reflector platelets floating in water. As a result of restraining the rotational axis of a platelet by applying the magnetic fields orthogonal to the incident light, an enhancement in the platelet that was similar to the brilliance changes shown in Fig. 2 was achieved.—

One of the characteristic colors that is produced by periodically aligned guanine platelets is blue. The coloring can result from changes in the relative distance and angle between platelets. The enhancement of a green color (Fig. 2) is proposed to be a mixture of pre-existing red color and additionally enhanced blue color. The mechanism for controlling the brilliance in spots of the dorsal belt of *A. lacunosus* within one second should be determined by the intracellular mechanical parts controlling the spacing and tilting angle of the reflector platelets. It was proposed that actin protein is one of the candidates that controls active color tuning in an iridophore [26]. It is questionable if actin fibers are able to perform a motion within one second inside the cell. Past studies have suggested the motion of tunable coloring on longer time spans. As a mechanical transducer tilts the platelet incline, liquid pressure actuation may be involved. A recent study found the effect of osmotic pressure on blue coloring in a photophore of luminous lanternfish.—

Compared with previous findings, the repeated blinking in a reflecting spot of *A. lacunosus* resulting in a faster light intensity change with a frequency of ~ 1 Hz suggested the direct government of an iridophore by the nervous system. The speed of the color change in squid should be influenced by both body motion and neural signals controlling their iridophore. [3] It seems that the speed of reflection intensity changes in *A. lacunosus* was faster. Additionally, different spatial resolutions and key materials of the reflector unit were used for each species. A protein named reflectin, which exists inside an iridophore of a squid, is broadly distributed in the tissue, while the reflector platelets of *A. lacunosus* are compactly packed in the circular spot with a diameter of 0.1 mm. This may suggest that the use of this spot contributes to a short distance communication between fishes whose body length is ~ 100 mm.

There is a possibility that the light reflection involves excited light from an unknown fluorescence molecule. Some literature has reported the optical properties of a guanine crystal of fish from the viewpoint of fluorescence. A complete explanation of the optics in a guanine platelet of fish is not possible at present; however, the rapid light intensity change in the spot reflection requires some mechanical apparatus that is driven by a neural signal. The aim of this apparatus should be the adjustment of light intensity and color balance for the efficient display of the back of *A. lacunosus*. The belt of spots resembles an illumination board for inter-species communication, which remains for bio-ethological research to clarify the meaning of forming a dynamic pattern of light spots on a two-dimensional display by reflecting environmental light. The analysis of the biological optical tool can provide us with an opportunity to make a mechanical actuator of micrometer sized optical parts embedded in a wet biological tissue.

Circular spot alignments in dorsal and parietal skin of *A. lacunosus* realize a blinking that resembles an apparent movement. The biological tissue constituted of reflecting micro-particles is actuated with a frequency of 1 Hz or higher.

Figure 5 shows an analysis of the reflecting particles in a dorsal spot. Figure 5 (a) and (b) are high resolution microscopy images of circular spots that were captured from a euthanized specimen. The reflecting circular spot was shown to exist on a black cell with dendrites. According to previous literatures [27,31], this black cell is a melanophore that contains black pigments, so-called melanin. In both images ((a) and (b)), there were many circular cells with a diameter of 7–10 μm , and these cells formed the observed circular spot. These circular cells were considered to be iridophores containing light reflecting platelets. In addition, the magnified image (Figure 5 (c, d)) indicated that the small circular cells had smaller particles whose diameter was estimated to be 1–3 μm .

To analyze these smaller particles, a piece dorsal skin with scales was separated from a euthanized specimen, and the particles were extracted from the skin by sticking with a plastic spatula in water. The particles floating in water exhibited Brownian motion and their rotation caused the light flash pattern shown in Figure 5 (e). The size of the particles was 1–3 μm . Larger particles with a size of 2–3 μm exhibited a distinct light reflection change. The light reflection behavior of particles floating in water resembled the light reflection of guanine platelets of goldfish and Japanese anchovy [21,32]. Fiber optic measurements on a reflecting particle appearing in a video display provided a time course of the light intensity, as shown in Figure 5 (f). The pulse width of the light flash generated by this floating particle was less than one second, i.e., a similar width to that observed from a dorsal spot. The frequency of the reflecting particles floating in water looked random. A mechanism controlling the frequency of flashes may exist inside the circular cells, spots or underlying skin. A detailed analysis of this will be performed in a future study.

Figure 5. Analysis of reflecting particles in the circular cells of a dorsal spot. (a), (b) Reflecting circular cells on a black cell with dendrites. Scale bar is 100 μm . (c) Magnified image of the area marked by a red square shown in (b). Scale bar is 15 μm . (d) Magnified image of the area marked by a blue square shown in (b). (e) Reflecting particles extracted from the dorsal circular spots. The particles are floating in water. Scale bar is 15 μm . (f) Fiber optic measurement of flashing of the reflector particles floating in water. The measurement was carried out at a position on a computer LCD screen displaying the recorded video (S_movieFIG5). The end of an optical fiber ($\phi 10$ mm) was positioned at the point marked by a red circle shown in (e).

4. Conclusion

It was found that circular spots on the dorsal part of hardyhead silverside fish, *Atherinomorus lacunosus*, cause a light flash under exposure to incident light.

Fiber optic measurements revealed that the flash in the spot continues in a style of a pulsed train whose frequency varied from 0.02 Hz to 4 Hz. Normally, an anesthetized living fish exhibited flashes of the dorsal spots at frequencies of 0.5–1 Hz. Higher frequencies at 3–4 Hz were also observed from the vibrational motion of the skin.

The dorsal spot with a typical diameter size of 100 μm consisted of 30–50 circular cells whose sizes were 7–10 μm . Each of the cells contained light reflecting particles with a size of 1–3 μm .

Data accessibility. All data are available from the Dryad Digital Repository: <https://doi.org/10.5061/dryad.9zw3r22bw> [35–33].
https://datadryad.org/stash/share/eC4kHyKuhPlwt_wYUp3-QzJtFXwikoVXJGloGFhwKA

Authors' contributions. M.I. performed the experimental design, experiments, measurements and analyses. All part of manuscript and illustrations were prepared by M. I.

Competing interests. Author declares no competing interests.

Funding. This work was supported by JST-CREST “Advanced core technology for creation and practical utilization of innovative properties and functions based upon optics and photonics (Grant number: JPMJCR16N1).”

Acknowledgements. The author thanks Edanz Group for editing a draft of this manuscript.

References

1. Kreit E, Mäthger LM, Hanlon RT, Dennis PB, Naik RR, Forsythe E, Heikenfeld J. 2012 Biological versus electronic adaptive coloration: how can one inform the other? *J. R. Soc Interface* **10**, 0601. (doi: 10.1098/rsif.2012.0601)
2. Allen JJ, Mäthger LM, Barbosa A, Buresch KC, Sogin E, Schwartz J, Chubb C, Hanlon RT. 2010 Cuttlefish dynamic camouflage: responses to substrate choice and integration of multiple visual cues. *Proc. R. Soc. B* **277**, 1031–1039. (doi: 10.1098/rspb.2009.1694)
3. Wardill TJ, Gonzalez-Bellido PT, Crook RJ, Hanlon RT. 2012 Neural control of tuneable skin iridescence in squid. *Proc. R. Soc. B* **279**, 4243–4252. (doi: 10.1098/rspb.2012.1374)
4. Mäthger LM, Bell G, Kuzirian AM, Allen JJ, Hanlon RT. 2012 How does the blue-ringed octopus (*Hapalochlaena lunulata*) flash its blue rings? *J. Exp. Biol.* **215**, 3752–3757. (doi: 10.1242/jeb.076869)
5. Denton EJ. 1970 Review Lecture: On the organization of reflecting surfaces in some marine animals. *Philos. Trans. R. Soc. London Ser. B* **258**, 285–313. (doi: 10.1098/rstb.1970.0037)
6. Denton EJ, Land MF. 1971 Mechanism of reflexion in silvery layers of fish and

- cephalopods. *Proc. R. Soc. Lond. B* **178**, 43-61.(doi: 10.1098/rspb.1971.0051)
7. Denton EJ, Herring PJ, Widder EA, Latz MF, Case JF. 1985 The roles of filters in the photophores of oceanic animals and their relation to vision in the oceanic environment. *Proc. R. Soc. Lond. B* **225**, 63-97.(doi: 10.1098/rspb.1985.0051)
 8. Herring PJ. 1994 Reflective systems in aquatic animals. *Comp. Biochem. Physiol.* **109A**, 513-546.(doi: 10.1016/0300-9629(94)90192-9)
 9. Herring PJ, Cope C. 2005 Red bioluminescence in fishes: on the suborbital photophores of *Malacosteus*, *Pachystomias* and *Aristostomias*. *Mar. Biol.* **148**, 383-394.(10.1007/s00227-005-0085-3)
 10. Partridge JC, Douglas RH, Marshall NJ, Chung WS, Jordan TM, Wagner HJ. 2014 Reflecting optics in the diverticular eye of a deep-sea barreleye fish (*Rhynchohyalus natalensis*), *Proc. R. Soc. B* **281**, 3223.(doi: 10.1098/rspb.2013.3223)
 11. Anderson TF, Richards AG. 1942 An electron microscope study of some structural colors of insects. *J. Appl. Phys.* **13**, 748-758.(doi: 10.1063/1.1714827)
 12. Kinoshita S, Yoshioka S, Kawagoe K. 2002 Mechanisms of structural colour in the Morpho butterfly: cooperation of regularity and irregularity in an iridescent scale. *Proc. Royal Soc. B* **269**, 1417-1421.(doi: 10.1098/rspb.2002.2019)
 13. Zhu D, Kinoshita S, Cai DS, Cole JB. 2009 Investigation of structural colors in Morpho butterflies using the nonstandard-finite-difference time-domain method: Effects of alternately stacked shelves and ridge density. *Phys. Rev. E* **80**, 051924.(doi: 10.1007/s11465-012-0347-y)
 14. Berthier S, Charron E, Da Silva A. 2003 Determination of the cuticle index of the scales of the iridescent butterfly *Morpho menelaus*. *Opt. Commun.* **228**, 349-356. (doi: 10.1016/j.optcom.2003.10.032)
 15. Plattner L. 2004 Optical properties of the scales of *Morpho rhetenor* butterflies: theoretical and experimental investigation of the back-scattering of light in the visible spectrum. *J. Royal Soc. Interface* **1**, 49-59.(doi: 10.1098/rsif.2004.0006)
 16. Wickham S, Large MCJ, Poladian L, Jermy LS. 2006 Exaggeration and suppression of iridescence: the evolution of two-dimensional butterfly structural colours. *J. Royal Soc. Interface* **3**, 99-108.(10.1098/rsif.2005.0071)
 17. Teyssier J, Saenko SV, van der Marel D, Milinkovitch MC. 2015 Photonic crystals cause active colour change in chameleons. *Nature Commun.* **6**, 6368.(doi: 10.1038/ncomms7368)
 18. Levy-Lior A, Shimoni E, Schwartz O, Gavish-Regev E, Oron D, Oxford G, Weiner S, Addadi L. 2010 Guanine Based Biogenic Photonic Crystal Arrays in Fish and Spiders. *Adv. Funct. Mater.* **20**, 320-329. (doi: 10.1002/adfm.200901437)
 19. Levy-Lior A, Pokroy B, Levavi-Sivan B, Leiserowitz L, Weiner S, Addadi L. 2008 Biogenic guanine crystals from the skin of fish may be designed to enhance light reflectance. *Cryst. Growth Des.* **8**, 507-511. (doi: 10.1021/cg0704753)
 20. Jordan TM, Partridge JC, Roberts NW. 2012 Non-polarizing broadband multilayer reflectors in fish. *Nature Photon.* **260**, 759-763.(doi: 10.1038/nphoton.2012.260)
 21. Iwasaka M, Asada H. 2018 Floating photonic crystals utilizing magnetically aligned biogenic guanine platelets. *Sci. Rep.* **8**, 16940. (doi: 10.1038/s41598-018-34866-x)
 22. Palmer BA, Taylor GJ, Brumfeld V, Gur D, Shemesh M, Elad N, Osherov A, Oron D, Weiner S, Addadi L. 2017 The image-forming mirror in the eye of the scallop. *Science* **358**, 1172–1175.(doi: 10.1126/science.aam9506)
 23. Chae J, Nishida S. 1994 Integumental ultrastructure and color patterns in the iridescent cope-pods of the family *Sapphirinidae* (Copepoda: *Poecilostomatoida*), *Marine Biology*, **119**, 205–210.(doi: 10.1007/BF00349558)
 24. Gur D, Leshem B, Pierantoni M, Farstey V, Oron D, Weiner S, Addadi L. 2015 Structural Basis for the Brilliant Colors of the *Sapphirinid* Copepods. *J. Am. Chem. Soc.* **137****26**, 8408-

- 8411.(doi: 10.1021/jacs.5b05289)
- ~~Nüsslein-Volhard C, Singh AP. 2017 How fish color their skin: A paradigm for development and evolution of adult patterns Multipotency, plasticity, and cell competition regulate proliferation and spreading of pigment cells in Zebrafish coloration, *Bioessays* **39**, 3, 1600231.(doi: 10.1002/bies.201600231)~~
25. Thayer, A, 1896 The law which underlies protective coloration. *Auk* **13**, 124–129.
26. Rowland, HM, 2009 From Abbott Thayer to the present day: what have we learned about the function of countershading? *Philos. Trans. R. Soc. B-Biol. Sci.* **364**, 519–527. (<https://doi.org/10.1098/rstb.2008.0261>).
27. Fujii R. 2000 Review- The Regulation of Motile Activity in Fish Chromatophores, *Pigment Cell Res.* **13**, 300–319.(doi: 10.1034/j.1600-0749.2000.130502.x)
28. Siebeck UE, Parker AN, Sprenger D, Mäthger LM, Wallis G. 2010 A Species of Reef Fish that Uses Ultraviolet Patterns for Covert Face Recognition, *Current Biology* **20**, 407–410. (doi: F. H.(doi: 10.1016/j.cub.2009.12.047)
29. Levy K, Lerner A, Shashar N. 2014 Mate choice and body pattern variations in the Crown Butterfly fish *Chaetodon paucifasciatus* (Chaetodontidae), *Biology Open* **3**, 1245–1251. (doi:10.1242/bio.20149175)
30. Nakamasu A, Takahashi G, Kanbe A, Kondo S. 2009 Interactions between zebrafish pigment cells responsible for the generation of Turing patterns, *Proc. Natl. Acad. Sci.* **106**, 8429–8434.(doi: 10.1073/pnas)
- ~~Shoji H, Mochizuki A, Iwasa Y, Hirata M, Watanabe T, Hioki S, Kondo S. 2003 Origin of Directionality in the Fish Stripe Pattern, *Developmental Dynamics* **226**, 627–633.(doi: 10.1002/dvdy.10277)~~
- ~~Mahalwar P, Walderich B, Singh AP, Nüsslein-Volhard C. 2014 Local reorganization of xanthophores fine-tunes and colors the striped pattern of zebrafish, *Science* **345**, 1362–1364. (doi: 10.1126/science.1254837)~~
- ~~Walderich B, Singh AP, Mahalwar P, Nüsslein-Volhard C. 2016 Homotypic cell competition regulates proliferation and tiling of zebrafish pigment cells during colour pattern formation, *Nat. Commun.* **7**, 11462. (doi: 10.1038/ncomms11462)~~
- ~~McMenamin SK, Bain EJ, McCann AE, Patterson LB, Eom DS, Waller ZP, Hamill JC, Kuhlman JA, Eisen JS, Parichy DM. 2014 Thyroid hormone-dependent adult pigment cell lineage and pattern in zebrafish, *Science* **345**, 1358–1361.(doi: 10.1126/science.1256251)~~
- ~~Price AC, Weadick CJ, Shim J, Rodd FH. 2008 Pigments, Patterns, and Fish Behavior, *Zebrafish* **5**, 297–307. (doi: 10.1089=zeb.2008.0551)~~
31. Demski LS, 1992 Chromatophore systems in teleosts and cephalopods: a levels oriented analysis of convergent systems, *Brain, Behavior and Evolution.* **40**, 141–156. (doi:10.1159/000113909)
32. Iwasaka M, Mizukawa Y, 2013 Light Reflection Control in Biogenic Micro-Mirror by Diamagnetic Orientation, *Langmuir* **29**, 4328–4334 (doi: 10.1021/la400046a)
33. Iwasaka M. 2020 Flashing spots on the dorsal part of hardyhead silverside fish—Rapid active reflection display on dorsal stripes of *Atherinomorus lacunosus* Dryad Digital Repository. (<https://doi.org/10.5061/dryad.9zw3r22bw>)

Flashing spots on the dorsal part of hardyhead silverside fish

Rapid active reflection display on dorsal stripes of

Atherinomorus lacunosus

-Supporting Information-

Masakazu Iwasaka

Hiroshima University, Kagamiyama 1-4-2, Higashihiroshima, Hiroshima 739-8527, Japan

miwamasa@hiroshima-u.ac.jp

A. Dependence of the iridophore spot brilliance on incident light direction

Figure S1 shows the appearance and disappearance of shining spots when the direction of incident light was changed. In the micrographs of A–C, incident light was supplied from the right side, and changes in the reflection in the spots and skin were compared by changing the incident angle slightly. The radiance in the iridophore spot was kept at the same level although the skin changed its reflection intensity quickly. When the incident light came from the bottom side of the observed area (D, E), the intense radiance in the iridophore disappeared. This kind of behavior, anisotropy of radiance in the reflector, is a typical light reflection property of guanine particles of biogenic fish. The evidence suggests that the brilliant spot occurred not as light emission but by light reflection.

Figure S1. Observation of the reflection in spots by changing the direction of incident light.

~~B. Possible concern of blood stream in the active reflection~~

~~Figure S2. Dynamics of the radiance changes, which were observed in a belt of iridophores.~~

~~Figure S2 shows an example of the dynamic motion of the stream-like brilliance change. It resembled an apparent motion. Therefore, it is possible to conjecture the concern of blood flow existing beneath the skin and iridophore spots. There is a case in which the connected iridophores are located near a peripheral blood vessel. The hypothesis is that the blood stream accompanying refractive index discontinuity can generate an apparent motion of the reflected light under an external light irradiance. This mechanism may not have a primal role for the active reflection but may overlap with the nervous control as the primal system.~~

Supporting Movie 1: S_movieFIG2A: Flashing spots in the dorsal part of hardyhead silverside fish. (This is the real-time movie, that is, the playback speed is the same as real time.)

This recorded video is the original data of Figure 2 (a). The spots are formed in an array along the edge of the belt. Five seconds after starting the playback, the magnified area, which is shown in Fig. 2(a), is presented. A synchronized flashing of the spots can be clearly observed.

Original video format: AVI, Frame rate of original video: 29.97 fps

Converted to MPEG4 format

Playback time 38.75 seconds

Supporting Movie 2: S_movieFIG2B: Flashing spots in the dorsal part of hardyhead silverside fish. (The playback speed is the same as real time.)

This recorded video is the original data of Figure 2 (b). The spots repeat the color changes between blue and green, and periodically, in a long-term cycle, a yellow flash appears.

The average diameter of the spots is estimated to be ~100 micrometers.

Original video format: AVI, Frame rate of original video: 29.97 fps

Converted to MPEG4 format

Playback time 108.41 seconds

Supporting Movie 3: S_movieFIG3A: Movie of the flashing spots analyzed for the changes of the light intensity speed analyses shown in Figure 3 (a). (The playback speed is the same as real time.)

An end of the optical fiber was positioned at the area showing the magnified image.

The width of the full-image was approximately 250 micrometers. The center spot showed that smaller circles existed inside the spot area. The size of these smaller circles was approximately 7–10 micrometers.

The cyclic shifts of the spot seemed to be same as the cyclic motion of the skin of a living fish. However, there were flashes occurring inside the spot which were independent of the skin motion.

Original video format: AVI, Frame rate of original video: 29.97 fps

Converted to MPEG4 format

Playback time 30.38 seconds

Supporting Movie 4: S_movieFIG3B: Movie of the flashing spots analyzed for the changes of the light intensity speed analyses shown in Figure 3 (b). (The playback speed is the same as real time.)

An end of the optical fiber was positioned at the area showing the magnified image.

Therefore, the obtained light intensity was from more than 10 spots at the same time.

The width of the full image was approximately 900 micrometers.

Original video format: AVI, Frame rate of original video: 29.97 fps

Converted to MPEG4 format

Playback time 120.24 seconds

Supporting Movie 5: S_movieFIG4A: Movie of rapidly flashing spots analyzed for the light intensity analyses on a LCD screen shown in Figure 4 (a). (The playback speed is the same as real time.)

From 2 seconds, the magnification and position were changed to the region of interest, but the playback speed was continuously kept as the real time.

Original video format: AVI, Frame rate of original video: 29.97 fps

Converted to MPEG4 format

Playback time 32.37 seconds

Supporting Movie 6: S_movieFIG4B: Movie of rapidly flashing spots analyzed for the light intensity analyses on a LCD screen shown in Figure 4 (b). (The playback speed is the same as real time.)

Original video format: AVI, Frame rate of original video: 29.97 fps

Converted to MPEG4 format

Playback time 63.11 seconds

Supporting Movie 7: S_movieFIG5: Movie of the reflecting particles which were extracted from the dorsal circular spots. This movie was used in the analysis shown in Fig. 5(e, f). (The playback speed is the same as real time.)

Original video format: AVI

Converted to MPEG4 format

Playback time 30.18 seconds

Microscope system: Digital microscope, Hirox, Tokyo Japan

Lens: MXB-2500REZ, Hirox, Tokyo Japan

~~**Movie S1. Dependence of the iridophore spot brilliance on incident light direction.** Movies of the data shown in Figure S1.~~

~~Specification of optics:~~

~~Digital microscope (PC 230, Vixen, Tokyo, Japan)~~

~~White LED light (MLEP-B070W1LRD, MCEP-CW8-2M, Moritex, Tokyo, Japan)~~

~~**Movie S2. Possible concern of blood stream in the active reflection.** Movies of the data shown in Figure S2.~~

~~Specification of optics:~~

~~Digital microscope (PC 230, Vixen, Tokyo, Japan)~~

~~White LED light (MLEP-B070W1LRD, MCEP-CW8-2M, Moritex, Tokyo, Japan)~~

~~**Movie S3. Active reflection blinking of the circular spots in the parietal skin of *Atherinomorus lacunosus*.**~~

~~Specification of optics:~~

~~Digital microscope (PC 230, Vixen, Tokyo, Japan)~~

~~White LED light (MLEP-B070W1LRD, MCEP-CW8-2M, Moritex, Tokyo, Japan)~~

Appendix D

Flashing spots on the dorsal **trunk** part of hardyhead silverside fish

Masakazu Iwasaka

Hiroshima University, Kagamiyama 1-4-2, Higashihiroshima, Hiroshima 739-8527, Japan

Correspondence to: Masakazu Iwasaka, miwamasa@hiroshima-u.ac.jp

A large number of living creatures are able to use **ambient**-environmental light effectively **in** as a biological **signaling** display. ~~The biological structural colors are very attractive not only to members of the same colorful species but also to humans.~~ *Atherinomorus lacunosus*, a teleost fish has alignments of circular spots on its dorsal **trunk** part. The spot consists of **iridophores** whose diameters are approximately 7 - 10 μm . The **iridophore** contains **guanine crystals** ~~reflecting particles~~ with diameters of 1-3 μm , which show light flashing. Here it is found that **more than one spot** ~~a spot~~ with a diameter of approximately 0.1 mm causes a rhythmic flashing of light under **illumination** ~~exposure to an incident light~~. The typical light flash has a pulse width of approximately one second. When a **pulsed train of flashes** appears, the flash repeats at a typical frequency of 0.5–1 Hz. **The observed phenomenon is the first finding of continuously flashing dorsal trunk of hardyhead silverside fish under white light illumination.**

1. Introduction

Studies on cephalopods (squids, cuttlefish and octopus) [1-4] have revealed that a dynamic camouflage effect occurs in the cellular tissue existing in the skin. A similar mechanism for color changes has also been discovered in the tissue structures of fish [5-10], insects, [11-16] and a mammal [11-17]. **In addition, many species of insects have static structural colors** [12-17]. In biological materials that are able to control light, guanine is useful for light reflection and optical interference in iridescent tissues [5-7]. Guanine-based photonic crystal structures have been found in many species of fish [5,6,10,18-21], scallop [22], and animal plankton [23,24].

It is often possible to observe many markings on the surface of a fish body. There is **a long history** of research on fish body colors concerning their purpose in the life, such as countershading [25, 26], **spectral coloration** [1-4, 27,28] and silvering camouflage of fish [20]. The role of patterns on a fish body surface has been suggested to be an active communication tool, and nonlinear mechanism studies have ascribed the stripe pattern in a fish to a Turing pattern, which is generated by an interaction between chromatophores [29-32 27-30]. Chromatophores are groups of cells containing particles that control the color based on the filtering or reflection of light [33 34]. Among the chromatophores, iridophores are known to possess reflecting particles such as guanine platelets [29, 33 27,34].

Concerning the guanine platelets of fish, we have **found** ~~observed~~ a light reflection change in the guanine platelets **by the magnetically induced orientation** ~~of fish upon exposure to a magnetic field.~~ [21,34 32]. We were able to control the orientation of the guanine **platelets,** **and** observed distinct light flashes. If the guanine platelets in fish skin exist in a flexible condition, we can expect a rapid reflection of light by them. The aim of this study is to **find** ~~confirm if a rapid light intensity change can be observed that relates to the opto-mechanical behavior of~~ **concerning** reflecting particles such as guanine platelets in a fish body. ~~It is interesting to explore various kinds of fish from the viewpoint of discovering new optical properties in a fish body.~~ In this study, I investigate the dynamics of light reflection for cells of the dorsal skin of the hardyhead silverside fish, *Atherinomorus lacunosus*.

2. Materials and Methods

2.1 Sample

Specimens of the hardyhead silverside fish, *Atherinomorus lacunosus*, were caught in Okinawa, Japan, by using a net, and then were transported to Hiroshima. The specimens were kept in an **outdoor open** aquarium (**Okinawa, 26–28 °C**) **in Okinawa** and in an indoor aquarium (**Hiroshima 26–27 °C**) **in Hiroshima**. ~~The temperature of the seawater in the aquarium was 26–28 °C (open aquarium) and 26–27 °C (indoor aquarium).~~ The height and weight of the used specimens were 98.9 mm +/- 4.9 mm and 9.45 g +/- 1.43 g, respectively. Nine specimens were used for this study. The sex of the used specimens was not determined in this study. Animal experiments in this study were carried out in accordance with the policies of Hiroshima University Animal Care and Use Committee (approval number: F19-2, Hiroshima University). A photograph of the belt-like pattern in the dorsal **trunk part** of *A. lacunosus* is shown in Figure 1 (a). This belt consisted of a wave-like alignment of circular spots.

2.2 Observation of anesthetized fish in **and** aquarium

Figure 1(b) shows the observation system for the fish in an aquarium, which consisted of a microscope lens and a light source. The fish were anesthetized by exposure to 0.1% 2-phenoxyethanol for 30 s to 1 minute, and then were placed inside of a small box (aquarium) containing

sea water with aeration. Experiments on the anesthetized fish were performed for 5 to 10 minutes. Before the anesthesia wore off, the fish were moved back to the large aquarium.

2.3 Video image recording

The images of dorsal spots were obtained by using two types of high resolution microscope lens, NAVITAR 2.0x1-51473 (Navitar, Rochester, USA) (Figure 1(b)) and Hirox MXB-2500REZ (Hirox, Tokyo Japan). The former lens (NAVITAR) was connected to a CMOS camera (HOZAN L-835, HOZAN, Osaka, Japan) by a C-mount adapter. The swimming fish were recorded by a video camera (FDR-AX45TIC, SONY, Tokyo, Japan) for the macroscopic presentation shown in Figure 1(a). The focal distance of the lens (NAVITAR) was 32 mm. For the digital microscopy recordings using capturing software (Xplovview, VIXEN, Tokyo, Japan), the white balance was manually set to be 4500–4650, and this white balance parameter was fixed during the individual video recording.

2.4 Light source

As a light source for the presented data in this paper, a white LED light (LA-HDF158A, Hayashi Repic Co. Ltd., Tokyo, Japan) was used. A light guide connected to this light source was directed towards the back of the fish, as shown in Fig. 1(b). The size of the diameter outlet of the light guide was approximately 8 mm. This provided a scattered light and the light polarization was not additionally modulated. The incident light was provided from the side, as illustrated in Fig. 1(b), and the scattered or reflected light from the sample was monitored.

2.5 Fiber optic measurements

2.5.1 Measurement of real light from dorsal spots and skin

As shown in Fig. 1(c), the reflected light from the dorsal trunk part of the fish was directly introduced to an optical fiber bundle made of quartz with a total diameter of 10 mm. One end of the optical fiber was connected to a cooled CCD (charge coupled device) spectrometer spectrophotometer (UCP-2000, Unisoku Co. Ltd. Osaka, Japan). The light was collected in the CCD device with an exposure time of 20 ms, sampling time of 40 ms, and recorded time of up to 300 s. The utilized spectrometer having a diffraction grating collected light over the wavelength of 450 nm to 600 nm at once. For the purpose of showing light intensity change (light flash rate), the data of light intensity at three to five wavelengths (450nm, 500nm, 528nm, 574nm, 600nm) was plotted. In the measurement preparation, the intensity of the collected light was monitored and the linearity of detection was confirmed. Because the aim of this experiment is measuring light intensity changes, the obtained raw data in arbitrary unit were plotted versus time without normalization. The other end of the optical fiber was positioned at the focused position on the top of the C-mount adapter of the microscope lens (NAVITAR).

Prior to the fiber optic measurement, adjustment of the region of interest was carried out by using a CMOS camera (Hozan). Exchange Replacement of the CMOS camera with the optical fiber was executed within 10 seconds, then the CCD-spectrophotometer started the light collection. The exchanges of camera and fiber were carried out without screwing. There was nothing lost in terms of data acquisition.

After the light collection was finished, the opposite replacement was carried out smoothly, and any drifting of the region of interest was monitored. Any data thought that were conjectured to have been collected where there was a drifting of the region of interest was were not used for the analysis.

2.5.2 Measurements on region of interest on a computer LCD screen during playback of recorded video

In order to collect data on flash rates, direct measurements on LCD screen was carried out. By using the same fiber optic system except for the lens, measurements on the region of interest on a computer display were carried out. First, the video playback was paused at the time when the region of interest appeared. After one of the ends of the optical fiber was placed close to the region and fixed with by using surgical tape (paper tape) on the LCD screen, light collection by a CCD spectrophotometer was started, then video playback commenced. To assist the fixation of optical fiber, a stand was also utilized. The data obtained from LCD screen measurement was normalized for intensity by using a white standard.

2.6 Additional image analysis

To check one of the fiber optic measurements (Fig. 3(a)(4)), additional image analyses were performed using Image J (NIH) software. After the color image was transformed to a gray scale image, the intensity relative to gray in the image was measured.

Figure 1. Experimental set-up for video recordings and fiber optic measurements on the dorsal trunk part of hardyhead silverside fish. (a) Image of the dorsal trunk part with an array of circular spots. Scale bar is 1 mm. (b) Experimental diagram of the observation of fish in an aquarium with a microscope lens and light source. An anesthetized fish was placed on a bed

made of soft clay in a small aquarium containing sea water with aeration. (c) Real light measurement, a method for collecting light from the observed area through the microscope lens. ~~Exchange Replacement~~ of the CMOS camera with an optical fiber ($\phi 10$ mm) and the opposite replacement were carried out smoothly (within 10 seconds). ~~(d) Collection of light from the region of interest on a LCD screen to which an end of optical fiber was affixed.~~

Figure 2. The observed phenomenon of the flash of light reflection in spots of the dorsal trunk part of hardyhead silverside fish. (a) and (b) Two examples of the phenomenon. Time indicators denote passing time in the recorded video for each of the static images. The static images in (a) and (b) were captured from the recorded videos S_movieFIG2A and 2B, respectively. Scale bar is 100 μ m.

3. Results and Discussion

Figure 2 shows two examples of the flashing spots on the dorsal trunk part of hardyhead silverside fish. The major color of the circular spot in Fig. 2(a) at 7.50 s and 8.73 s was blue or green, but became yellow at 9.30 s. A similar phenomenon is shown in Fig. 2(b), where many circular spots proceeded with the color change process which was reversible. The spots seemed to cause a light reflection flash.

Next, the continuous flashing of the circular spots was investigated, as shown in Figure 3. These data were obtained from two specimens that were different from the specimens used in Fig. 2. The flash frequency of a dorsal spot was analyzed by using two types of fiber optic measurements. In the first specimen (Fig. 3(a)), the flashing of the real light was monitored for 300 s (Fig. 3(a)(1)). A distinct flash at three wavelengths (528 nm, 574 nm, and 600 nm) was obtained because the light source intensity at these wavelengths was higher than at others (450 nm and 500 nm). Magnification of the time course for the light at 528 nm (Fig. 3(a)(3)) suggested that the roughly estimated frequency of this spot was 0.5–1 Hz.

In the first two seconds of S_movieFIG3A, 26 sheets of static color images were captured (25 frame per second) and transformed to gray scale images, then the intensity of gray was plotted versus time, as shown in Fig. 3(ba)(4). The intensity time course exhibited approximately four peaks during the first two seconds. The inset images correspond to the intensity changes.

~~By using fiber optic light collection from a LCD screen, additional analysis could be carried out (Fig. 3(a) (5-7)). The red circle marked on Fig. 3(a)(3) was the position measured by the optical fiber. Fig. 3(a)(5) shows the change in light intensity from the beginning to 30s in the playback of S_movieFIG3A. Fig. 3(a)(6) shows a magnification of the first two seconds. The intensity was the summation of the light intensity at five wavelengths (450 nm, 500 nm, 528 nm, 574 nm, and 600 nm). The roughly estimated frequency of this spot was 0.3–1 Hz, which was similar to the results of the real light shown in Fig. 3(a) (1-2).~~

~~In addition, As a possible artifact, the effect of skin movement was should be tested. In the blue circle marked in Fig. 3(a)(3), an edge of a neighboring spot appeared that showed a vibrational movement during video playback. The change of light intensity arising from this movement was analyzed on the LCD screen by a fiber optic measurement (Fig. 3(a)(7)). The result indicated the existence of a change in light intensity with a frequency of around 2 Hz that resulted from skin motion, and this corresponded to the results of the image-only analysis (Fig. 3(a)(4)).~~

Figure 3(c) shows the light flashes in another specimen. Fig. 3(cb)(1) presents the long-term cycle of the light flash that occurred every fifty seconds, which was repeated at least three times. Magnification of the time course for the light at three wavelengths (528 nm, 574 nm, and 600 nm) (Fig. 3(cb)(3)) indicates that the dorsal spots caused a flash of light reflection at frequencies of 0.5–0.7 Hz. The data shown here were obtained only by real light collection through the microscope lens and optical fiber that was connected to the spectrophotometer. In addition, this example showed a slower frequency at 0.02 Hz.

A more detailed analysis at higher frequencies for the flash of dorsal spots was carried out is needed. The analyses were carried out only using light collection from the LCD screen, as shown in Figure 4. The intensity was the summation of the light intensity at five wavelengths (450 nm, 500 nm, 528 nm, 574 nm, and 600 nm). The data were obtained from an additional two specimens. Both specimens showed flashes that exhibited changes in their intensity at frequencies of 3 Hz to 4 Hz. However, the light intensity changes in a black cell also showed a similar frequency, 3 Hz ~ 4 Hz. The analyzed videos (S_movieFIG4A and 4B) exhibited a period time with rapid flashing. The fish skin vibration at a frequency of 3–4 Hz may be

involved with the flash of the dorsal spots. In addition to the skin motion, other influencing factors, such as blood flow, may be conjectured. In the present study, most of observed dorsal spots existed apart from blood vessels, and no correlation between the flash and blood flow was detected.

Figure 5 shows an analysis of the reflecting particles, presumably guanine crystals in a dorsal spot. Figure 5 (a) and (b) are high resolution microscopy images of circular spots that were captured from a euthanized specimen. The reflecting circular spot was shown to exist on a black cell with dendrites. According to previous studies literatures [29, 33 27,31], this black cell is a melanophore that contains the black pigments, so-called melanin. In both images ((a) and (b)), there were many iridophores with a diameter of 7–10 μm , and these cells formed the observed circular spot. These iridophores were considered to be iridophores containing light reflecting platelets. In addition, the magnified image (Figure 5 (c, d)) indicated that the small iridophores had smaller guanine crystal particles whose diameter was estimated to be 1–3 μm .

To analyze these smaller particles, a piece of dorsal skin with scales was separated from a euthanized specimen, and the particles were extracted from the skin by sticking with a plastic spatula in water. The particles floating in water exhibited Brownian motion and their rotation caused the light flash pattern shown in Figure 5 (e). The size of the particles was 1–3 μm . Larger particles with a size of 2–3 μm exhibited a distinct light reflection change. The light reflection behavior of the particles floating in water resembled the light reflection of guanine platelets of goldfish and Japanese anchovy [21,34 32]. Fiber optic measurements on a reflecting particle appearing in a video display provided a time course of the light intensity, as shown in Figure 5 (f). The pulse width of the light flash generated by this floating particle was less than one second, i.e., a similar width to that observed from a dorsal spot. The frequency of the reflecting particles floating in water appeared looked random. The random reflection of the floating particles was same as the previously reported behavior of guanine crystal platelets floating in water [34]. A mechanism controlling the frequency of flashes may exist inside the iridophores, spots or underlying skin. A detailed analysis of this will be performed in a future study.

4. Conclusion

It was found that circular spots on the dorsal trunk-part of hardyhead silverside fish, *Atherinomorus lacunosus*, cause a light flash under illumination.

Fiber optic measurements revealed that the flash in the spot continues in a style of a pulsed train whose frequency varied from 0.02 Hz to 4 Hz. Normally, an anesthetized living fish exhibited flashes of the dorsal spots at frequencies of 0.5–1 Hz. Higher frequencies at 3–4 Hz were also observed from the vibrational motion of the skin.

The dorsal spot with a typical diameter size of 100 μm consisted of 30–50 iridophores whose sizes were 7–10 μm . Each of the cells contained light reflecting guanine crystal particles with a size of 1–3 μm .

The results indicated that in the dorsal trunk of hardyhead silverside fish, there were iridophores frequently changing their light intensity under white light illumination by using guanine crystal particles.

Figure 3. Analysis of flash frequency of the dorsal spot by using two types of fiber optic measurements, light collection through a microscope lens and from a LCD screen.

(a) First specimen. (1) Time course of real light at five wavelengths through a microscope lens after light reflection from the body surface of the dorsal trunk part. (2) Static image in the beginning of S_movieFIG3A. Scale bar is 100 μ m. The spot existed in the area measured for

the data (1) and (3), which was determined separately from the video recording. (3) Magnification of the time course at 528 nm shown in (1) from 150 s to 180 s.

(5)–(7) Analyses on light collection from the LCD screen. The value of the intensity was a summation of the light intensity at five wavelengths (450 nm, 500 nm, 528 nm, 574 nm, 600 nm). (4) Time course of intensity of gray which was measured from the spot shown in (3).

(b) Image analysis using Image-J was carried out after the color image was transformed to a gray scale image. (5–7) Data obtained with a cooled CCD spectrometer. (5–6) Time course of light intensity in the whole area of the video image. (7) Time course of light collection in the blue circle shown in (3), where an edge of a neighboring spot appears.

(c) Second specimen. (1) Time course of real light at three wavelengths through a microscope lens after light reflection from the body surface of the dorsal trunk part. (2) Static image taken at the beginning of S_movieFIG3B. Scale bar is 100 μ m. The spot existed in the area measured for the data (1) and (3), which was determined separately from the video recording. (3) Magnification of the time course of (1) from 100 s to 120 s.

Figure 4. Analysis of flash frequency of the dorsal spot showing a fast flashing. (a), (b) Two examples of fiber optic measurements from a position on a computer LCD screen displaying the recorded video (S_movieFIG4A, 4B). Red and blue circles indicate the position on a dorsal spot and a black cell to which an end of an optical fiber (ϕ 10 mm) was positioned, respectively.

Start time means the time in the recorded video when the spectrometer started its measurement on the LCD screen. Scale bar is 100 μm .

Figure 5. Analysis of reflecting particles in the iridophores of a dorsal spot. (a), (b) Reflecting iridophores on a black cell with dendrites. Scale bar is 100 μm . (c) Magnified image of the area marked by a red square shown in (b). Scale bar is 15 μm . (d) Magnified image of the area marked by a blue square shown in (b). (e) Reflecting particles extracted from the dorsal circular spots. The particles are floating in water. Scale bar is 15 μm . (f) Fiber optic measurement of flashing of the reflector particles floating in water. The measurement was carried out at a position on a computer LCD screen displaying the recorded video (S_movieFIG5). The end of an optical fiber ($\phi 10 \text{ mm}$) was positioned at the point marked by a red circle shown in (e).

Data accessibility. All data are available from the Dryad Digital Repository: <https://doi.org/10.5061/dryad.9zw3r22bw> [335].

https://datadryad.org/stash/share/eC4kHyKuhOPlwt_wYUp3-QzJtFXwikoVXJGloGFhwKA

Authors' contributions. M.I. performed the experimental design, experiments, measurements and analyses. All part of manuscript and illustrations were prepared by M. I.

Competing interests. Author declares no competing interests.

Funding. This work was supported by JST-CREST “Advanced core technology for creation and practical utilization of innovative properties and functions based upon optics and photonics (Grant number: JPMJCR16N1).”

Acknowledgements. The author thanks Edanz Group for editing a draft of this manuscript.

References

1. Kreit E, Mäthger LM, Hanlon RT, Dennis PB, Naik RR, Forsythe E, Heikenfeld J. 2012 Biological versus electronic adaptive coloration: how can one inform the other? *J. R. Soc Interface* **10**, 0601. (doi: 10.1098/rsif.2012.0601)
2. Allen JJ, Mäthger LM, Barbosa A, Buresch KC, Sogin E, Schwartz J, Chubb C, Hanlon RT. 2010 Cuttlefish dynamic camouflage: responses to substrate choice and integration of multiple visual cues. *Proc. R. Soc. B* **277**, 1031–1039.(doi: 10.1098/rspb.2009.1694)
3. Wardill TJ, Gonzalez-Bellido PT, Crook RJ, Hanlon RT. 2012 Neural control of tuneable skin iridescence in squid. *Proc. R. Soc. B* **279**, 4243–4252.(doi: 10.1098/rspb.2012.1374)
4. Mäthger LM, Bell G, Kuzirian AM, Allen JJ, Hanlon RT. 2012 How does the blue-ringed octopus (*Hapalochlaena lunulata*) flash its blue rings? *J. Exp. Biol.* **215**, 3752-3757.(doi: 10.1242/jeb.076869)
5. Denton EJ. 1970 Review Lecture: On the organization of reflecting surfaces in some marine animals. *Philos. Trans. R. Soc. London Ser. B* **258**, 285-313.(doi: 10.1098/rstb.1970.0037)
6. Denton EJ, Land MF. 1971 Mechanism of reflexion in silvery layers of fish and cephalopods. *Proc. R. Soc. Lond. B* **178**, 43-61.(doi: 10.1098/rspb.1971.0051)
7. Denton EJ, Herring PJ, Widder EA, Latz MF, Case JF. 1985 The roles of filters in the photophores of oceanic animals and their relation to vision in the oceanic environment. *Proc. R. Soc. Lond. B* **225**, 63-97.(doi: 10.1098/rspb.1985.0051)
8. Herring PJ. 1994 Reflective systems in aquatic animals. *Comp. Biochem. Physiol.* **109A**, 513-546.(doi: 10.1016/0300-9629(94)90192-9)
9. Herring PJ, Cope C. 2005 Red bioluminescence in fishes: on the suborbital photophores of *Malacosteus*, *Pachystomias* and *Aristostomias*. *Mar. Biol.* **148**, 383-394.(doi:

- 10.1007/s00227-005-0085-3)
10. Partridge JC, Douglas RH, Marshall NJ, Chung WS, Jordan TM, Wagner HJ. 2014 Reflecting optics in the diverticular eye of a deep-sea barreleye fish (*Rhynchohyalus natalensis*), *Proc. R. Soc. B* **281**, 3223.(doi: 10.1098/rspb.2013.3223)
 11. Teyssier J, Saenko SV, van der Marel D, Milinkovitch MC. 2015 Photonic crystals cause active colour change in chameleons. *Nature Commun.* **6**, 6368.(doi: 10.1038/ncomms7368)
 12. Anderson TF, Richards AG. 1942 An electron microscope study of some structural colors of insects. *J. Appl. Phys.* **13**, 748-758.(doi: 10.1063/1.1714827)
 13. Kinoshita S, Yoshioka S, Kawagoe K. 2002 Mechanisms of structural colour in the Morpho butterfly: cooperation of regularity and irregularity in an iridescent scale. *Proc. Royal Soc. B* **269**, 1417-1421.(doi: 10.1098/rspb.2002.2019)
 14. Zhu D, Kinoshita S, Cai DS, Cole JB. 2009 Investigation of structural colors in Morpho butterflies using the nonstandard-finite-difference time-domain method: Effects of alternately stacked shelves and ridge density. *Phys. Rev. E* **80**, 051924.(doi: 10.1007/s11465-012-0347-y)
 15. Berthier S, Charron E, Da Silva A. 2003 Determination of the cuticle index of the scales of the iridescent butterfly *Morpho menelaus*. *Opt. Commun.* **228**, 349-356. (doi: 10.1016/j.optcom.2003.10.032)
 16. Plattner L. 2004 Optical properties of the scales of *Morpho rhetenor* butterflies: theoretical and experimental investigation of the back-scattering of light in the visible spectrum. *J. Royal Soc. Interface* **1**, 49-59.(doi: 10.1098/rsif.2004.0006)
 17. Wickham S, Large MCJ, Poladian L, Jermy LS. 2006 Exaggeration and suppression of iridescence: the evolution of two-dimensional butterfly structural colours. *J. Royal Soc. Interface* **3**, 99-108.(doi; 10.1098/rsif.2005.0071)
 18. Levy-Lior A, Shimoni E, Schwartz O, Gavish-Regev E, Oron D, Oxford G, Weiner S, Addadi L. 2010 Guanine Based Biogenic Photonic Crystal Arrays in Fish and Spiders. *Adv. Funct. Mater.* **20**, 320-329. (doi: 10.1002/adfm.200901437)
 19. Levy-Lior A, Pokroy B, Levavi-Sivan B, Leiserowitz L, Weiner S, Addadi L. 2008 Biogenic guanine crystals from the skin of fish may be designed to enhance light reflectance. *Cryst. Growth Des.* **8**, 507-511. (doi: 10.1021/cg0704753)
 20. Jordan TM, Partridge JC, Roberts NW. 2012 Non-polarizing broadband multilayer reflectors in fish. *Nature Photon.* **260**, 759-763.(doi: 10.1038/nphoton.2012.260)
 21. Iwasaka M, Asada H. 2018 Floating photonic crystals utilizing magnetically aligned biogenic guanine platelets. *Sci. Rep.* **8**, 16940. (doi: 10.1038/s41598-018-34866-x)
 22. Palmer BA, Taylor GJ, Brumfeld V, Gur D, Shemesh M, Elad N, Osherov A, Oron D, Weiner S, Addadi L. 2017 The image-forming mirror in the eye of the scallop. *Science* **358**, 1172–1175.(doi: 10.1126/science.aam9506)
 23. Chae J, Nishida S. 1994 Integumental ultrastructure and color patterns in the iridescent cope-pods of the family *Sapphirinidae* (Copepoda: *Poecilostomatoida*), *Marine Biology*, **119**, 205–210.(doi: 10.1007/BF00349558)
 24. Gur D, Leshem B, Pierantoni M, Farstey V, Oron D, Weiner S, Addadi L. 2015 Structural Basis for the Brilliant Colors of the *Sapphirinid* Copepods. *J. Am. Chem. Soc.* **137****26**, 8408-8411.(doi: 10.1021/jacs.5b05289)
 25. Thayer, A, 1896 The law which underlies protective coloration. *Auk* **13**, 124–129.
 26. Rowland, HM, 2009 From Abbott Thayer to the present day: what have we learned about the function of countershading? *Philos. Trans. R. Soc. B-Biol. Sci.* **364**, 519–527. (doi: 10.1098/rstb.2008.0261).
 27. Marshall J, 2000 *The Visual Ecology of Reef Fish Colours*, In “Signaling and Signal Design in Animal Communication,” Edited by Y. Espmark, T. Amundsen, and G. Rosenquist, Tapir Academic Press, 83-120. (ISBN 82-519-1545-7)

28. Marshall NJ, Jennings K, McFarland WN, Loew ER, Losey GS, 2003 Visual Biology of Hawaiian Coral Reef Fishes. II. Colors of Hawaiian Coral Reef Fish, *Copeia* **2003**, 455-466. (<https://www.jstor.org/stable/1448696>)
29. Fujii R, 2000 Review- The Regulation of Motile Activity in Fish Chromatophores, *Pigment Cell Res.* **13**, 300–319.(doi: 10.1034/j.1600-0749.2000.130502.x)
30. Siebeck UE, Parker AN, Sprenger D, Mäthger LM, Wallis G. 2010 A Species of Reef Fish that Uses Ultraviolet Patterns for Covert Face Recognition, *Current Biology* **20**, 407–410. (doi: 10.1016/j.cub.2009.12.047)
31. Levy K, Lerner A, Shashar N. 2014 Mate choice and body pattern variations in the Crown Butterfly fish *Chaetodon paucifasciatus* (*Chaetodontidae*), *Biology Open* **3**, 1245–1251. (doi:10.1242/bio.20149175)
32. Nakamasu A, Takahashi G, Kanbe A, Kondo S. 2009 Interactions between zebrafish pigment cells responsible for the generation of Turing patterns, *Proc. Natl. Acad. Sci.* **106**, 8429–8434.(doi: 10.1073/pnas)
33. Demski LS, 1992 Chromatophore systems in teleosts and cephalopods: a levels oriented analysis of convergent systems, *Brain, Behavior and Evolution.* **40**, 141–156. (doi:10.1159/000113909)
34. Iwasaka M, Mizukawa Y, 2013 Light Reflection Control in Biogenic Micro-Mirror by Diamagnetic Orientation, *Langmuir* **29**, 4328–4334 (doi: 10.1021/la400046a)
35. Iwasaka M. 2020 Flashing spots on the dorsal trunk part of hardyhead silverside fish, Dryad Digital Repository. (<https://doi.org/10.5061/dryad.9zw3r22bw>)

Flashing spots on the dorsal **trunk** part of hardyhead silverside fish

-Supporting Information-

Masakazu Iwasaka

Hiroshima University, Kagamiyama 1-4-2, Higashihiroshima, Hiroshima 739-8527, Japan
miwamasa@hiroshima-u.ac.jp

Supporting Movie 1: S_movieFIG2A: Flashing spots in the dorsal **trunk part of hardyhead silverside fish.** (This is the real-time movie, that is, the playback speed is the same as real time.)

This recorded video is the original data of Figure 2 (a). The spots are formed in an array along the edge of the belt. Five seconds after starting the playback, the magnified area, which is shown in Fig. 2(a), is presented. A synchronized flashing of the spots can be clearly observed.

Original video format: AVI, Frame rate of original video: 29.97 fps
Converted to MPEG4 format
Playback time 38.75 seconds

Supporting Movie 2: S_movieFIG2B: Flashing spots in the dorsal **trunk part of hardyhead silverside fish.** (The playback speed is the same as real time.)

This recorded video is the original data of Figure 2 (b). The spots repeat the color changes between blue and green, and periodically, in a long-term cycle, a yellow flash appears. The average diameter of the spots is estimated to be ~100 micrometers.

Original video format: AVI, Frame rate of original video: 29.97 fps
Converted to MPEG4 format
Playback time 108.41 seconds

Supporting Movie 3: S_movieFIG3A: Movie of the flashing spots analyzed for the changes of the light intensity speed analyses shown in Figure 3 (a). (The playback speed is the same as real time.)

An end of the optical fiber was positioned at the area showing the magnified image. The width of the full-image was approximately 250 micrometers. The center spot showed that smaller circles existed inside the spot area. The size of these smaller circles was approximately 7–10 micrometers.

The cyclic shifts of the spot seemed to be same as the cyclic motion of the skin of a living fish. However, there were flashes occurring inside the spot which were independent of the skin motion.

Original video format: AVI, Frame rate of original video: 29.97 fps

Converted to MPEG4 format

Playback time 30.38 seconds

Supporting Movie 4: S_movieFIG3CB: Movie of the flashing spots analyzed for the changes of the light intensity speed analyses shown in Figure 3 (cb). (The playback speed is the same as real time.)

An end of the optical fiber was positioned at the area showing the magnified image.

Therefore, the obtained light intensity was from more than 10 spots at the same time.

The width of the full image was approximately 900 micrometers.

Original video format: AVI, Frame rate of original video: 29.97 fps

Converted to MPEG4 format

Playback time 120.24 seconds

Supporting Movie 5: S_movieFIG4A: Movie of rapidly flashing spots analyzed for the light intensity analyses on a LCD screen shown in Figure 4 (a). (The playback speed is the same as real time.)

From 2 seconds, the magnification and position were changed to the region of interest, but the playback speed was continuously kept as the real time.

Original video format: AVI, Frame rate of original video: 29.97 fps

Converted to MPEG4 format

Playback time 32.37 seconds

Supporting Movie 6: S_movieFIG4B: Movie of rapidly flashing spots analyzed for the light intensity analyses on a LCD screen shown in Figure 4 (b). (The playback speed is the same as real time.)

Original video format: AVI, Frame rate of original video: 29.97 fps

Converted to MPEG4 format

Playback time 63.11 seconds

Supporting Movie 7: S_movieFIG5: Movie of the reflecting particles which were extracted from the dorsal circular spots. This movie was used in the analysis shown in Fig. 5(e, f). (The playback speed is the same as real time.)

Original video format: AVI

Converted to MPEG4 format

Playback time 30.18 seconds

Microscope system: Digital microscope, Hirox, Tokyo Japan

Lens: MXB-2500REZ, Hirox, Tokyo Japan

Appendix E

Dear Reviewers and Editors,

I hope you are keeping well at this difficult time. I appreciate your support of my manuscript.

I have revised the following manuscript according to the three Reviewers' comments.

Manuscript ID: RSOS-201578.R1

Former title of the manuscript: Flashing spots on the dorsal part of hardyhead silverside fish

Revised title: Flashing spots on the dorsal trunk of hardyhead silverside fish

Author: Masakazu Iwasaka

Thank you very much for reading my manuscript and spending time on providing a review. According to the very important comments from all reviewers, I have revised the manuscript. I hope this revision suitably responds to the comments.

Sincerely

Masakazu Iwasaka, Prof.
Research Institute for Nanodevice and Bio Sysrems
Hiroshima University
1-4-2 Kagamiyama, Higashihiroshima
739-8527 Hiroshima, Japan
miwamasa@hiroshima-u.ac.jp

Responses to Reviewer comments:

Responses to Reviewer 1

Reviewer comment:

The paper has been significantly improved upon revision and is an important contribution to the field of structural coloration. One minor comment is that the author has removed any mention of 'iridophores', perhaps due to a misunderstanding of one the reviewers comments. In the revised ms iridophore has been substituted for 'circular cell'. Given their reflective properties and location in the fish skin (and the well-known fish chromatophore system), then I think it would be perfectly acceptable to term the circular cells 'iridophores' or 'iridophore cells' throughout the paper.

Response:

I appreciate the reviewer's important advices. According to the comment, the sentences were revised. The term 'circular cells' was changed to 'iridophores' throughout the paper.

Reviewer comment:

On a similar note, the author has also removed most mention of guanine crystals in the paper in exchange for 'particles'. I appreciate there is no direct spectroscopic or diffraction data proving the particles are guanine in this paper, but, given everything we know, then it would be very fair to assume they are indeed guanine. I think it would be entirely reasonable for the author to discuss the optical phenomenon in terms of guanine crystal movement, rather than just 'particles'. One option would be that in the first occurrence in the results section then the term 'particles' be followed with the parenthesis "(presumably guanine crystals (refs))". Thereafter I think the terms particles/platelets/crystals or even guanine crystals could be used interchangeably.

Response:

According to reviewer's comment, the term 'particles' was changed to 'guanine crystals' or 'guanine crystals platelets.'

I appreciate all of the important comments from the Reviewer-1. Thank you so much for spending your time to review the manuscript.

Responses to Reviewer 2

Reviewer comment:

I reviewed an earlier version of this manuscript and agreed to re-review it. The author has made substantial changes to the manuscript and has addressed most of the comments that were raised earlier. However, in revising the manuscript, its figures, removing data and adding new data to this paper, a few new concerns have come up for me. Most of these may merely require revising the manuscript again, although I do have some substantial concerns that may need the author to take out datasets or think of ways to collect these data differently. More details below.

Response: I appreciate the reviewer's helpful comments.

According to reviewer's comment, the sentences were revised.

Reviewer comment:

Title- I would replace PART with TRUNK Dorsal part could be head as well. Trunk is the back of a fish.

Abstract line 35: I suggested this before... I would stay away from the term DISPLAY because it is a very specific form of SIGNAL. Instead, I would rewrite this sentence to read... "... use ambient light effectively in biological signalling."

Next sentence: whether they are attractive or not, even to members of the same species, depends on the circumstances and the type of signal. There are certainly deterring visual signals out there. I would delete the sentence. It doesn't add anything to the abstract and nothing is lost by taking it out.

Line 38. After this sentence (ends in PART), move the two last sentences here.

Response:

According to reviewer's comment, the sentences were revised.

Reviewer comment:Line 38: more than one spot.

Response: According to reviewer's comment, the sentences were revised.

Reviewer comment: Line 42: what is a flash train?

Response: 'flash train' means pulsed train of flashes. The text was revised using this term.

Reviewer comment: Last two sentence... move up

Finish abstract with a conclusion and/or why this study matters. I currently find this a weak and uninformative abstract.

Response: According to reviewer's comment, the sentences were revised. In the end of Abstract,

following sentence was added.

“The observed phenomenon is the first finding of continuously flashing dorsal trunk of hardyhead silverside fish under white light illumination.”

Reviewer comment: Introduction

Line 8: insects do not have dynamic colour change for camouflage. At least the refs 11-16 cannot be used to that effect. An appearance of changing colour can be affected by movement but this is not the mechanism that the author is trying to introduce here, i.e., dynamic colour changes that originate from a structural/physiological etc. change in the skin.

Response: According to reviewer’s comment, the references were deleted.

Reviewer comment: Line 16: and other examples, e.g., spectral colouration, rather than just silvery. Look at some of the papers from NJ Marshall's lab.

Response: According to reviewer’s comment, other examples were inserted.

Reviewer comment: Line 25: The author said that they removed the magnetic field part of the study. It is still mentioned here.

Response: As I mentioned, new data for this study using magnetic field were removed from Results and Discussion. I am just referring our previous studies for explaining the aim of this study.

The sentences referring previous paper using magnetic orientation were newly added, in order to enhance the motivation of this study. To make this clear, the sentence was revised as follows.

Concerning the guanine platelets of fish, we have found a light reflection change in the guanine platelets by the magnetically induced orientation [21,34].

Reviewer comment: Line 28: The aim of this study.... Has the author truly accomplished to confirm that rapid light intensity changes relate to the optomechanics...? I do not see any more information on that anywhere in the manuscript (and also see my later comment on plates in suspension). To truly answer this question the author would have had to employ much higher res imaging techniques than were used, including also techniques that study the structural component in intact cells.

Response: According to reviewer’s comment, the sentences were revised. In the end of Introduction, following sentence was added.

“The aim of this study is to find a rapid light intensity change concerning reflecting particles such as guanine platelets in a fish body.”

Reviewer comment: Line 31: Sentence that starts with “It is interesting...” delete that sentence unless it can be made stronger. Just because something is interesting isn't enough justification.

Response: According to reviewer's comment, the sentence was revised.

Reviewer comment: Materials and methods

Overall comment... Is the formatting 2.5, 2.5.1 etc strictly necessary? It reads more like a progress report or a thesis than a scientific paper...? Maybe this is just my opinion.

Response: According to reviewer's comment, the sentences were revised.

Reviewer comment: Lines 42-44: combine these two sentences into one. Temperatures right after mentioning the tanks.

Response: According to reviewer's comment, the sentences were revised as follows.

"The specimens were kept in an open aquarium (~~Okinawa~~, 26–28 °C) in Okinawa and in an indoor aquarium (~~Hiroshima~~ 26–27 °C) in Hiroshima."

Reviewer comment: Light source, point 2.4. line 21. SIZE ... OUTLET Does the author mean diameter? Outlet might not be best wording...maybe replace by focused beam or something.

Response: According to reviewer's comment, the sentences were revised as follows.

"The size of the diameter of the light guide was approximately 8 mm."

Reviewer comment: Section on fiber optic measurements

Which wavelengths? How were they measured? Filters? More detail, please. HOWEVER, I would like to add that I do not completely agree with the author about the use of the spectrophotometer in obtaining spectral data. More details below.

Response: According to reviewer's comment, the term 'spectrophotometer' was changed to 'spectrometer.' The utilized apparatus was cooled CCD (charge coupled device) type of spectrometer having a diffraction grating. Without filter, the machine can collect light over the wavelength of 450 nm to 600 nm at once. In the presented data some of wavelengths were selected for showing light intensity changes. The purpose of this study is investigating light flash rate, not the spectral profile. I am showing the data of light intensity at three to five wavelengths (450nm, 500nm, 528nm, 574nm, 600nm). The data is only utilized for the evaluation of the flash rate.

The details were explained as following sentences.

"The utilized spectrometer having a diffraction grating collected light over the wavelength of 450 nm to 600 nm at once. For the purpose of showing light intensity change (light flash rate), the data of light intensity at three to five wavelengths (450nm, 500nm, 528nm, 574nm, 600nm) was plotted. In the measurement preparation, the intensity of the collected light was monitored and the linearity of detection was confirmed. Because the aim of this experiment is measuring light intensity changes, the obtained raw data in arbitrary unit were plotted versus time without normalization."

Reviewer comment: Line 42... data were not was

Response: According to reviewer's comment, the sentences were revised.

Reviewer comment: Section 2.5.2 Measurements on LCD screen. Why were these done? I do not follow the reason. A computer LCD screen is far from replicating the reflective properties of the real animal. LCD colours are based on RGB content. Why would you collect spectral data from an LCD screen? Because of intensity changes? If so, all mention of spectra needs to be removed. All data need to be normalized for intensity, calibrated against a white standard, etc.

Response: Yes, it is because of measuring intensity changes. According to reviewer's comment, the data obtained from LCD screen measurement was normalized for intensity by using a white standard.

Reviewer comment: Fig. 1b, top and side views? Make this obvious.

Response: According to reviewer's comment, the figure was revised.

Reviewer comment: Fig. 1c, why do we see the tank set up twice if the point of this fig is that the camera and the spectrometer were used interchangeably during an experiment? I would suggest to just write at the top of the camera box that this location was for camera and spec.

Response: According to reviewer's comment, the figure was revised.

Reviewer comment: Fig. 1d. This newly added method makes no sense to me. I am guessing that what the author is trying to do is collect data on spectral changes but because the spots are too small to capture with the setup the author employed, they used an LCD screen to obtain these data. For reasons mentioned above, this is not possible. No spectral data are transmitted through an LCD screen. A microscope should have been used here and the fiber optic cable should have been attached to that. There are plenty of papers that document how others have accomplished collecting spectral reflectance data from small reflecting structures. And if the point was to collect data on flash rates, then this needs to be stated clearly and any mention of spectral data needs to be removed.

Response: This study is the latter case. The point was to collect data on flash rates. Any mention of spectral data was removed. Fig. 1d was also removed to avoid misunderstanding.

According to reviewer's comment, following sentences were added.

“In order to collect data on flash rates, direct measurements on LCD screen was carried out. The data obtained from LCD screen measurement was normalized for intensity by using a white standard.”

Reviewer comment: Caption of Fig. 1 last 4 lines. Not replacement... they were exchanged to be able

to record video and spectra. Ideally, to be able to do both, one would use a scope with multiple mounts that are either functional at the same time (because of beam splitters) or one switches back and forth between the ports, rather than having to unscrew the camera, place the spec, unscrew that again and put the camera back in place. The author may want to be transparent about this and state that there was nothing lost in terms of data acquisition.

Response: According to reviewer's comment, the sentences was revised. In addition, following sentence was added to make it clear.

“The exchanges of camera and fiber were carried out without screwing. There was nothing lost in terms of data acquisition.”

Reviewer comment: Results and discussion section, lines 29-35: LCD vs wavelength... see my earlier comment.

Response: According to reviewer's comment, the sentences were revised.

Reviewer comment: Line 37 Author writes SHOULD BE but this needs to be WAS?

Response: According to reviewer's comment, the sentences were revised.

Reviewer comment: Lines 37-44: why does the author call this a possible artifact? Could this be the underlying mechanism? Or is the author suggesting the possibility that there really are no dynamic changes at all but what we see are merely the result of a change in viewing angle that gives this the appearance of colour change when in actual fact, nothing changes at all? Multilayer reflectors do that... they change colour depending on the direction from which they are viewed. This is without any active changes inside the reflector.

Response: According to reviewer's comment, the term 'artifact' was deleted.

“In addition, ~~As a possible artifact,~~ the effect of skin movement was ~~should be~~ tested.”

Line 53: I am more comfortable about the author using an LCD screen to get a better idea about flash rate although those data were already obtained using the video camera. So, I'm not sure why we are seeing different ways of doing the same thing.

Response: I am writing in Results and Discussion of 1st revision about the aim of LCD screen measurement.

“A more detailed analysis at higher frequencies for the flash of dorsal spots was carried out ~~is needed.~~”

In some movies, it looked like there were higher frequencies in flashing. For understandable presentation of figure, LCD screen measurements were carried out.

Reviewer comment: Line 59/60 and following page 3-5. Sentences on blood flow. I would remove

this altogether. Maybe you have more evidence down the line worth a follow up study but this comes without any evidence, so I would suggest removing the sentences.

Response: According to reviewer's comment, the sentences were deleted.

Reviewer comment: Line 22, p7 of manuscript. Section on floating particles. If reflecting plates are placed in suspension, they will do what the author describes because they are moving around relative to the light source. I do not understand why this is important to mention here? Perhaps a little more detail is needed to make this point clear. Is the author using this as evidence that this fish species' reflecting system is based on an array of ordered platelets? I would like to point out that even collagen, and many other cellular structures, will reflect light if teased apart and placed in suspension. It's merely a difference in refractive index that will cause this. Others (especially the older literature from the 1960s and 1970s, namely Denton and others) have used this technique to determine refractive indices of reflective plates but, in my opinion, just placing reflectors inside a liquid does not give us much of anything.

Response: As written in final section of Results and Discussion of 1st revision, "The particles floating in water exhibited Brownian motion and their rotation caused the light flash pattern shown in Figure 5 (e)."

As the reviewer comments, just placing reflectors inside a liquid does not cause anything. In the collagen polymerization, collagen gel changed its turbidity depending on diameter of polymerized fibers.

Important point is making a condition where the guanine particles can cause Brownian motion.

By checking the strong flashing of guanine particles floating in water, we can conjecture that the particles are guanine crystal.

Reviewer comment: Conclusion

Line 34: EXPOSURE... this sounds like the light is causing the flashes, i.e., is there a photosensitive aspect to this? of not, I would merely state "under white light illumination"

General comment about the conclusion... Is there anything else? I think this paper needs a general discussion and subsequent conclusions about why this study is important, how it relates to other studies on fish, cephalopods. See more thoughts at the end of my review.

Response: According to reviewer's comment, the sentences were revised.

Reviewer comment: Fig. 3

Why arbitrary units? Were these not calibrated to give you an intensity relative to a white standard? Sometimes these arbitrary units were a log scale apart...? Data could/should be normalized?

This fig needs a lot more work to be clear. The caption is very confusing an all over the place. The

order of the panels inside this figure are also confusing and given that the author is trying to report data from two specimens I would expect to see the same order of panels for each specimen inside this figure.

Response: As I mentioned before, the point of this study is measuring intensity changes in flashing. Figure 3 is showing raw data of the pulsed flash under white light illumination. Reflectance and reflectivity were not discussed on the data. Frequency of the flashing was discussed. So, the raw data without any normalization are important.

According to reviewer's comment, the same order of panels was presented in the 2nd revision.

Reviewer comment: Fig. 4

As with Fig. 3... arbitrary units that are all over the place. The layout of the figure is also very confusing. images and panels need to be aligned, labelled, etc.

Response: According to reviewer's comment, the arbitrary units were revised. The vertical axis shows the value calibrated by using white standard sample (reflectance standard plate). The order of panels was revised and labeled.

Reviewer comment: Fig. 5.

Label dendrites.

Response: According to reviewer's comment, the figure was revised.

Reviewer comment: A general comment of Figures.

The author is presenting data on a species of fish that shows reflecting flashes. This is the story of this paper. In my opinion, there are way too many Figs to make this point. Given the simplicity of what the author is reporting, the Figures are over-complicating the story, adding data that do not add to the story (e.g., recording spectral data from an LCD screen), even adding the spectrophotometer data on flash rate when the author was already able to obtain these data using the video camera. I would suggest condensing the Figs so that there is one figure showing images, not several, and one Fig that shows flash rates, not several. Also, once the author has revised the method for collecting spectral data, one Fig. that shows these data, not several.

Response: According to reviewer's comment, the figures were revised.

In Fig. 3, the number of panels was reduced. All of data are for explaining the flash rate, not for spectral analysis. I added this explanation in the revised text. In Fig. 1, there are no panel of method for collecting spectral data, there are only for flash rate. To avoid confusing, last panel in Fig. 1 was deleted.

Reviewer comment: FINAL COMMENT:

I continue to like this story, and I do not need convincing that these types of studies are interesting and necessary. However, not everyone agrees on this and I do think a focus on a bigger picture is necessary. There are many examples in the existing literature that report colour changes in fish. How is this study different? Why does a reader need to read this paper? What does the study add to what we know? What are the shortcomings, what are the improvements that could be made? Where does this study lead us in the future? These are only a small number of thoughts that could guide the author in revising the discussion and conclusion sections. I do not suggest adding lengthy sentences because I do like the conciseness of the current manuscript. After all, this is a “brief communication”- type paper and it needs to remain brief. But it is currently lacking important information.

Response: According to reviewer’s comment, Abstract and Conclusion were enhanced to provide the important information.

In Abstract: “The observed phenomenon is the first finding of continuously flashing dorsal trunk of fish under white light illumination.”

In Conclusion: “The results indicated that in the dorsal trunk of hardyhead silverside fish, there were iridophores frequently changing their light intensity under white light illumination by using guanine crystal particles.”

I appreciate all of the important comments from the Reviewer-2. Thank you so much for spending your time to review the manuscript.

Responses to Reviewer 3

Reviewer comment:

The author has addressed my suggestions well. I just have a few wording suggestions, and questions.

P21L14: "...a long history of.."

Response: According to reviewer's comment, the sentence was revised.

Reviewer comment: P21L26: "...platelets, and observed.."

Response: According to reviewer's comment, the sentence was revised.

Reviewer comment: P21L43: What is an open aquarium? Is it outside?

Response: Yes, an outside aquarium in Okinawa, warm place.

Reviewer comment: P22L41: "Any data thought to have been collected.."

Response: According to reviewer's comment, the sentence was revised.

Reviewer comment: P22L50: ..and fixed with.."

Response: According to reviewer's comment, the sentence was revised.

Reviewer comment: P22L53: A stand was used for what purpose?

Response: The stand was used to stabilize the optical fiber collecting the light from LCD screen.

The sentence was revised as follows,

"To assist the fixation of optical fiber, a stand was also utilized."

Reviewer comment: P25L51: "...spots was carried out.."

Response: According to reviewer's comment, the sentence was revised.

Reviewer comment: P25L57:time->period

Response: According to reviewer's comment, the sentence was revised.

Reviewer comment: P26L7: "...spots captured.."

Response: According to reviewer's comment, the sentence was revised.

Reviewer comment: P26L9: literatures->studies

Response: According to reviewer's comment, the sentence was revised.

Reviewer comment: P26L10: "...contains the black pigment melanin"

Response: According to reviewer's comment, the sentence was revised.

Reviewer comment: P26L16L "...a piece of dorsal skin.."

Response: According to reviewer's comment, the sentence was revised.

Reviewer comment: P26L28: "...appeared random.."

Response: According to reviewer's comment, the sentence was revised.

I appreciate all of the important comments from the Reviewer-3. Thank you so much for spending your time to review the manuscript.

Appendix F

Dear Reviewers and Editors,

I hope you are keeping well at this difficult time. I appreciate your support of my manuscript.

I have revised the following manuscript according to the Reviewers' comments.

Manuscript ID: RSOS-201578.R2

Title: Flashing spots on the dorsal trunk of hardyhead silverside fish

Author: Masakazu Iwasaka

Thank you very much for reading my manuscript and spending time on providing a review. According to the very important helpful comments from the reviewer, I have revised the manuscript. I hope this minor revision suitably responds to the comments.

Sincerely

Masakazu Iwasaka, Prof.
Research Institute for Nanodevice and Bio Systems
Hiroshima University
1-4-2 Kagamiyama, Higashihiroshima
739-8527 Hiroshima, Japan
miwamasa@hiroshima-u.ac.jp

Responses to Reviewer comments:

Responses to Reviewer 2

Reviewer comment: Abstract Line 37 of iridophores, whose (comma)

Response: I appreciate the reviewer's helpful comment. According to reviewer's comment, the sentence was revised.

Reviewer comment: Line 39 delete WHICH SHOWS LIGHT FLASHING because you mention it in the following sentence.

Response: According to reviewer's comment, the sentence was revised.

Reviewer comment: Line 43 UNDER ILLUMINATION... I would replace with WHEN VIEWED UNDER WHITE LIGHT

Response: According to reviewer's comment, the sentence was revised.

Reviewer comment: Last sentence: I would revise this sentence more along the lines of why the study is important and what we can learn from this study...

Incidentally: A paper by Mathger et al 2003 J Exp Biol 206: 3607-3613 reported fast reflective changes (continuous flashing) in a tropical teleost, so the phenomenon you are reporting is not completely unique hardyheads. However, your study adds to a growing body of literature reporting colour changes in an animal group that was (and still is) largely considered to have static colouration. I would finish the sentence with something along these lines.

Response: According to reviewer's comment, the sentences were revised as follows.

"The observed phenomenon is one of the evidences for the existence of rapid colour changing teleost fish."

The references were revised by adding the key literature, "Mäthger LM, Land, MF, Siebeck, UE, Marshall, NJ. 2003 Rapid colour changes in multilayer reflecting stripes in the *Pentapodus paradiseus*, *J. Exp. Biol.* **206**, 3507-3613". In addition, I recognized that a literature reporting rapid color changes of the chameleon sand tile fish was reported in 2017. Let me add this literature as well.

I made a referencing of these literature in Introduction and in the last part of discussion.

Reviewer comment: Introduction Line 6 The author claims colour changes in a mammal and uses a chameleon paper as a reference? (Reptile...)

Response: According to reviewer's comment, the sentence was revised.

Reviewer comment: Materials and methods

Section on Fiber optic measurements

Line 32 Rewrite sentence along these lines: The spectrometer was equipped with a diffraction grating that simultaneously collected light ranging from 450 nm to 650 nm.

Response: According to reviewer's comment, the sentence was revised.

Reviewer comment: Line 38 This comment is regarding the arbitrary units (I commented on this previously). As a seasoned spectrometer user myself, I suspect that the author's "arbitrary" units are relative to a white standard. Assuming that the set-up was calibrated before each measurement and specimens were not moved, magnification not changed, total reflective area inside field of view remained steady, these measurements can be presented on a scale from 0-1, or 0-100%, whichever the author prefers. The units are therefore not arbitrary. Of course, you would not normalize all of these data. You would not want to cancel out the intensity changes you are trying to demonstrate, but I would suggest getting away from using "arbitrary".

Response: According to reviewer's comment, I deleted "arbitrary units" in the text. As well as the data in Fig. 4, the data shown in Fig,3 and Fig.5 were calibrated relative to white standard. The y-axis of the graph of these figures is changed to 'Light intensity relative to white standard.'

Reviewer comment: Line 45 WITHOUT SCREWING.... Maybe find a better term here... Or just take it out. All that matters is that it was a quick equipment change, and that you didn't miss anything. Next sentence: Maybe re-write sentence? E.g., No crucial data were missed during spectrometer/camera exchange.

Response: According to reviewer's comment, I deleted the sentence. The next sentence was re-written.

Reviewer comment: Line 46 Exchange, rather than REPLACEMENT

Response: According to reviewer's comment, the term was revised.

Reviewer comment: Line 47 Drifting? Explain, please. Specimen moved outside of field of view etc.

Response: According to reviewer's comment, the sentence was revised.

Reviewer comment: Line 49 Re-write perhaps? ... thought to have been collected while the region of interest drifted ...?

Response: According to reviewer's comment, the sentence was revised.

Reviewer comment: Important information missing here: Calibration... what kind of reflectance standard was used? Diameter of fiber optic cable?

Response: I added following explanation for the calibration in the text.

“The obtained data were given relative to a white standard by using a reflectance standard plate (SRS-99-010, Labsphere Inc., New Hampshire, USA) with the same experimental system mentioned above.”
The inner diameter of the utilized optical fiber bundle was 5mm (outer diameter of fiber cable was 10mm).

Reviewer comment: Section on measurements off LCD screen

Line 54 Make clearer WHY this was done... To magnify reflecting areas sufficiently for recording...
The first two sentences could also be combined, they are a little bit repetitive.

Response: According to reviewer's comment, the sentence was revised.

Reviewer comment: Last sentence: Did you not use a white standard before? I would reword this and state that data WERE (not WAS) given RELATIVE TO a white standard (not normalized)

Response: I used white standard. All of the figures showing fiber optic measurement results were revised by using the suggested term, 'Light intensity relative to white standard.'

Reviewer comment: Fig. 1C I would say EXCHANGE, rather than replacement. No need for both terms

Response: According to reviewer's comment, the term was revised.

Reviewer comment: Fig. 4 Why is the start time in (a) 25 sec but the x-axis starts at zero? In (b), start time is 0 sec and the x-axis starts at zero too? Just a minor comment but it gets a reader confused. In reality, you can probably get rid of the two start times above the images.

Response: To avoid a reader confused, the two start times above the images were deleted.

Reviewer comment: Now the y-axis is normalized? How come? This is inconsistent compared to Fig. 3. See my comments above about the arbitrary units. These are presumably relative to a white standard.

Response: The value of y-axis, light intensities were calibrated by using white standard. The term 'normalized' was changed to 'Light intensity relative to white standard.'

Reviewer comment:

Fig. 5 Maybe make all images the same size and align them? Fig looks a tad careless.

Response: According to reviewer's comment, alignment of panels of Fig. 5 was rearranged.

Reviewer comment:

Final comment: The author has improved the manuscript tremendously. I only have one last suggestion, which I raised previously and still feel needs to be addressed. How does this study compare to others in the field? How is it different? Is the flashing faster, slower, spans wider wavelength, etc., than what other studies have reported? As it stands, this is a very brief report that lacks a bigger picture.

Response: I appreciate the reviewer teaching me about the key literature “Mähger LM, Land, MF, Siebeck, UE, Marshall, NJ. 2003 Rapid colour changes in multilayer reflecting stripes in the *Pentapodus paradiseus*, *J. Exp. Biol.* 206, 3507-3613”.

I made a simple discussion about the difference between the pioneer works and my present report, as follows.

“The pioneer works on the rapid colour change of iridophore of teleost fish, *Pentapodus paradiseus* [11] and *Hoplolatilus chlupatyi* [12] showed a distinct colour change from blue to red. The hardyhead silverside fish, *Atherinomorus lacunosus* did not show a change to red colour, but the fish could repeat the flashing faster.”

I appreciate all of the important and helpful comments from the Reviewer.
Thank you so much for spending the time to review this manuscript.